# A Strategy-Agnostic Framework for Partial Participation in Federated Learning

## Abstract

Partial participation (PP) is a fundamental paradigm in federated learning, where only a fraction of clients can be involved in each communication round. In recent years, a wide range of mechanisms for partial participation have been proposed. However, the effectiveness of a particular technique strongly depends on problem-specific characteristics, e.g. local data distributions. Consequently, achieving better performance requires a comprehensive search across a number of strategies. This observation highlights the necessity of a unified framework. In this paper, we address this challenge by introducing a general scheme that can be combined with almost any client selection strategy. We provide a unified theoretical analysis of our approach without relying on properties specific to individual heuristics. Furthermore, we extend it to settings with unstable client-server connections, thereby covering real-world scenarios in federated learning. We present empirical validation of our framework across a range of PP strategies on image classification tasks, employing modern architectures, such as FasterViT.

## 1 Introduction

Optimization is a cornerstone of training machine learning and neural network models. In a nutshell, almost every AI-based solution aims to minimize an empirical risk (Shalev-Shwartz et al., 2010), which evaluates how well the data is approximated. This process involves adjusting parameters to reduce the discrepancy between predicted outputs and ground truth labels, thereby improving generalization performance. Formally, the problem can be expressed as

$$\min_{x \in \mathbb{R}^d} \left[ \frac{1}{n} \sum_{i=1}^{n} \ell(g(x, a_i), b_i) \right], \tag{1}$$

where $x$ denotes the trainable parameters of the model $g$, $(a_i, b_i)$ is the $i$-th sample from the dataset with size $n$, and $\ell$ is the loss function. Nowadays, there is a variety of methods developed to efficiently solve equation 1 (Robbins and Monro, 1951; Nesterov, 1983; Kingma and Ba, 2014; Defazio and Mishchenko, 2023). The current successes of machine/deep learning owe much to the development of powerful numerical techniques that enable training on a huge amount of samples. Large-scale data processing became possible with the advancement of distributed optimization (Verbraeken et al., 2020). Instead of solving the problem on a single machine, samples are shared among $M$ nodes/devices/clients/machines connected via a server. Hence, the problem equation 1 transforms into

$$\min_{x \in \mathbb{R}^d} \left[ f(x) = \frac{1}{M} \sum_{m=1}^{M} f_m(x) = \frac{1}{M} \sum_{m=1}^{M} \frac{1}{n_m} \sum_{i_m=1}^{n_m} \ell(g(x, a_{i_m}), b_{i_m}) \right], \tag{2}$$

where $n_m$ is the size of the dataset, stored on $m$-th device.

### 1.1 Client Weighting

Parallel data processing helps to reduce computational time significantly (Zinkevich et al., 2010; Abadi et al., 2016; Jouppi et al., 2017). However, contemporary applications present new challenges. Training samples are often accumulated locally by each specific machine, rather than being collected and distributed manually. This paradigm with data remaining on edge devices is called federated learning (Konečný et al., 2016; McMahan et al., 2017; Bonawitz et al., 2019). In such a setup, local datasets are typically heterogeneous – they vary in size, distribution, and quality. For instance, one device may hold unique objects that are poorly represented across the rest of the network, but are

crucial for capturing more dependencies. This leads to the conclusion that some clients may be more useful than others. Modern approaches usually assign dynamic weights $\{\pi_m\}_{m=1}^M$ and use

$$f(x) = \sum_{m=1}^M \pi_m f_m(x), \;\; \text{s.t.} \;\; \pi_m > 0, \sum_{m=1}^M \pi_m = 1 \tag{3}$$

to calculate statistics. If the devices are considered to be equivalent, this corresponds to the case where $\pi_1 = \ldots = \pi_M = 1/M$. As a result, more important nodes contribute more significantly to the global loss. There are many strategies to prioritize the clients known in the literature.

**Weighting Based on Data Quality/Quantity.** The most straightforward way to cope with data imbalance is to consider a number of local samples. McMahan et al. (2017) suggested setting each coefficient as the constant $\pi_m = n_m/n$. Since then, many modifications of this approach have been proposed, including federated averaging schemes with momentum (Wang et al., 2019; Reddi et al., 2020), variance reduction (Liang et al., 2019; Karimireddy et al., 2020) and proximal updates (Li et al., 2020). However, this type of weighting ignores heterogeneity in terms of data quality, leading to bias, e.g. if some client holds an enormous amount of objects with the same labels. To support the diversity of training samples, Yurochkin et al. (2019) proposed to match the neurons of client neural networks before averaging. Building on the foundations laid by this work, subsequent works have explored more efficient approaches extensively (Wang et al., 2020a; Zhang et al., 2022; Yang et al., 2023; Wu et al., 2023; Kafshgari et al., 2023).

**Learned Weighting Strategies.** It is also common to learn weighting strategies instead of using fixed heuristics. Mohri et al. (2019) were among the first to present results in this direction. They proposed solving the saddle-point problem $\min_{x \in \mathbb{R}^d} \max_{\pi \in \triangle_1^M} \sum_{m=1}^M \pi_m f_m(x)$ to give small weights to well-trained devices. The idea of optimizing agnostic empirical loss was then generalized by Li et al. (2019a). Their `q-FedAvg` can be reduced to agnostic optimization as one of the special cases. However, in practice, it is hard to search for appropriate saddle-points (Daskalakis and Panageas, 2018; Jin et al., 2020), especially in federated learning (Sharma et al., 2023). As a result, the community has shifted towards softer adaptive approaches based on local losses (Zhang et al., 2020; Gao et al., 2022) and gradients (Wang et al., 2020b; Luo et al., 2024).

**Robust Weighting.** The idea of assigning weights to the devices found its application in robust optimization, where malicious clients can disrupt the learning process (Baruch et al., 2019; Xie et al., 2020; Fang et al., 2020). To combat such attacks, advanced schemes usually compute $\{\pi_m\}_{m=1}^M$, as the trust scores of the devices based on their objectives decrease (Xie et al., 2019), local gradients (Cao et al., 2020; Yan et al., 2023), and the number of local samples (Cao and Lai, 2019). Recently, researchers came up with the idea of using a Bayesian approach (Yang et al., 2024).

## 1.2 CLIENT SAMPLING

Another significant issue of federated learning, on par with heterogeneity, is the communication bottleneck (Tang et al., 2020; Shi et al., 2020). Sharing information between machines is costly and can limit the positive effect of parallelism, which is especially tangible when clients send messages to the server (Kairouz et al., 2021). This issue is magnified in federated learning, where edge devices may have unstable network connectivity, and transmitting large updates may be prohibitively slow. Many techniques exist to reduce communication (Seide et al., 2014; Alistarh et al., 2017; Stich, 2018). Partial participation is a special one among them (Li et al., 2019b; Yang et al., 2021). In each communication round, only a random subset of clients participates in training, while the rest remain inactive. This approach offloads the server by decreasing the number of updates that need to be aggregated. Moreover, it provides significant advantages in edge computing, where communication channels are not equivalent, or some of them may be unavailable. Nowadays, there is a wide range of heuristics, which allows to choose subset of clients efficiently.

**Data-Based Sampling Strategies.** Methods from this class rely on zero- and first-order information of local functions. `Importance Sampling FedAvg` (Rizk et al., 2021) was one of the first such approaches. The authors suggested evaluating the relevance of a device by how large its gradient is relative to the others. Indeed, a small gradient makes a weak contribution to the step. Consequently, communication with this node can be neglected. Nguyen et al. (2020) proposed an orthogonal approach. Their `FOLB` measures the angle between local and average gradient. If it is negative, then such a device is useless at the current moment. This idea was then developed extensively in (Wu

and Wang, 2022; Zhou et al., 2022). In addition, techniques based on the norms of updates (Chen et al., 2020) and local loss decrease (Cho et al., 2022) were proposed. There are also a number of approaches that dynamically exploit data heterogeneity to maintain balance (Zhang et al., 2023) or support diversity (Chen and Vikalo, 2024).

**System-Based Sampling Strategies.** Another approach is to use information about the network itself. `FedCS` (Nishio and Yonetani, 2019) categorizes clients into groups based on their computational power. This strategy saves wall-clock time by avoiding frequent selection of weak devices. Another class of techniques optimizes energy consumption (Xu and Wang, 2020). Most modern system heterogeneity techniques also incorporate local data considerations (Lai et al., 2021; Li et al., 2022). `F3AST` (Ribero et al., 2022) learns an availability-dependent client selection strategy to minimize the impact of variance on the global model's convergence.

Thus, the community came up with various techniques for weighting and sampling to make partial participation as efficient as possible. The development of each new scheme was challenging in terms of algorithm design and convergence proof. Consequently, a number of papers appeared attempting to propose a theory without utilizing the properties of any particular strategy.

### 1.3 UNIFICATION OF SAMPLING STRATEGIES

Existing papers in this area of research are built around the federated averaging scheme (McMahan et al., 2017). Li et al. (2019b) proposed an analysis for strongly convex objectives, obtaining a sublinear convergence rate $\mathcal{O}\left(\kappa^2/K\right)$, where $\kappa$ is the condition number. However, they modeled the partial participation environment via unbiased sampling. Cho et al. (2022) were the first to study the unified case with biased devices selection. They derived $\mathcal{O}\left(\kappa^2/K + \kappa Q\right)$, where $Q$ is a non-vanishing term that becomes zero solely in the absence of sampling bias. Thus, the authors recovered the results of Li et al. (2019b), but failed to extend the theory to weaker assumptions. The first success in this direction was achieved in (Luo et al., 2022). This work resolved key questions regarding biased sampling in the strongly convex case. However, the non-convex analysis holds greater significance for applications. For this setting, Wang and Ji (2022) obtained $\mathcal{O}\left(\sqrt{L}/\sqrt{K} + \delta\right)$, where $L$ is the smoothness constant and $\delta$ is the uniform bound on the difference between local gradients. This result contains the non-vanishing term and does not match the lower bound $\Omega\left(L/K\right)$ (Carmon et al., 2020).

Thus, current works in this field rely on `FedAvg`. Consequently, their analysis requires boundedness of gradients (Li et al., 2019b; Cho et al., 2022; Luo et al., 2022) or their differences (Wang and Ji, 2022) even in the non-stochastic case. Therefore, there is still no flawless unified theory of partial participation.

### 1.4 OUR CONTRIBUTION

In contrast to prior works, where partial participation analysis was built upon `FedAvg`, we introduce our own scheme to leverage client sampling. While existing techniques ignore the information from inactive clients, our approach utilizes it for benefits. Namely, devices accumulate gradient surrogates locally, and the server accounts for them after the full aggregation round. The proposed approach allows weighting and sampling clients according to a variety of strategies, including biased ones. The convergence of our scheme can be proven in both strongly convex and non-convex cases without introducing unnatural assumptions. The obtained rates do not contain non-vanishing terms. To validate the theory, we conduct experiments with RESNET-18 and VIT.

## 2 SETUP

We begin presenting our results with assumptions necessary to prove convergence. First of all, the objective is assumed to be smooth. This requirement is well-established in optimization.

**Assumption 2.1.** The function $f$ is $L$-smooth, i.e. for all $x, y \in \mathbb{R}^d$ it satisfies
$$\|\nabla f(x) - \nabla f(y)\| \leqslant L\|x - y\|.$$

Neural networks tend to have a complex loss landscape (Cybenko, 1989; Nguyen and Hein, 2018). Since we are motivated by real-world scenarios, our main goal is to prove convergence in the non-convex case. For completeness, we also derive results under stronger assumptions.

**Assumption 2.2.** The function $f$ is:

(a) **non-convex** with at least one global minimum:

$$\text{there exists may be not unique, } x^* \text{ s.t. } f(x^*) = \inf_{x \in \mathbb{R}^d} f(x) > -\infty.$$

(b) $\mu$**-strongly convex**, i.e. for all $x, y \in \mathbb{R}^d$ it satisfies

$$f(y) \geqslant f(x) + \langle \nabla f(x), y - x \rangle + \frac{\mu}{2} \|y - x\|^2.$$

Federated learning methods usually require a bound on data heterogeneity to provide convergence guarantees (Khaled et al., 2020; Karimireddy et al., 2020). In our work, we quantify it via gradients (Tang et al., 2018; Stich, 2020).

**Assumption 2.3.** Each gradient $\nabla f_m$ is similar to the full gradient $\nabla f$, i.e. for all $x \in \mathbb{R}^d$ it satisfies

$$\frac{1}{M} \sum_{m=1}^{M} \|\nabla f_m(x) - \nabla f(x)\|^2 \leqslant \delta_1 \|\nabla f(x)\|^2 + \delta_2.$$

This assumption is not too strict, since we do not require uniform boundedness ($\delta_1 = 0$). The following one is imposed to derive convergence of our algorithm with local stochasticity. If one removes it, our theory still holds.

**Assumption 2.4.** Each worker has access to a stochastic gradient $\nabla f_m(x, \xi_m)$. This is an unbiased random variable with bounded variance, i.e. for all $x \in \mathbb{R}^d$ it satisfies

$$\mathbb{E}_{\xi_m} \left[ \nabla f_m(x, \xi_m) \right] = \nabla f_m(x),$$

$$\mathbb{E}_{\xi_m} \left[ \|\nabla f_m(x, \xi_m) - \nabla f_m(x)\|^2 \right] \leqslant \sigma^2.$$

This assumption appears in different forms in a number of classic papers (Stich, 2018; Gower et al., 2019; Gorbunov et al., 2020). Next, we consider that weights $\{\pi_m\}_{m=1}^{M}$ from equation 3 lie on the regularized simplex. Namely, $\pi \in \Delta_1^M \cap \left( \bigcap_{m=1}^{M} \left\{ \pi : e_m^\top \pi + \frac{\alpha}{M} \geqslant 0 \right\} \right)$, where $1 \leqslant \alpha \leqslant M$ is the regularization parameter and $e$ is the unit basis. This technique is useful for solving a wide range of tasks (Mehta et al., 2024).

# 3 ALGORITHMS AND ANALYSIS

## 3.1 MOTIVATION

Existing papers on the unification of client sampling consider `FedAvg` without any modifications. Section 1.3 suggests that this approach is not promising due to poor results even under strong assumptions. A potential direction for future research could be to find a more suitable scheme. Below we propose an intuition that helps to address this issue.

To understand biased sampling, Cho et al. (2022) introduced the definition of selection skew and utilized it in the analysis. This is exactly the cause of the non-vanishing term in their rate. Indeed, there is no convergence if, for example, some devices are never selected for communication. However, we propose that the problem could be solved if we could somehow account for the error accumulated due to bias. To develop this idea, we formalize the sampling strategy as follows. First, we assign weights $\pi_m$ to devices, as described in equation 3. Next, we define the selection rule of the server as a stochastic operator $\mathcal{R} : \mathbb{R}^M \to \mathbb{R}^M$ that zeros some entries of the input vector while retaining the others. Applying this operator to the introduced vector of weights, it can be seen that the wide variety of strategies described in Section 1.2 fits this formalism. This applies not only to simple cases of selecting clients with the highest weights but also to non-trivial ones, such as zeroing the weights of unavailable nodes.

Viewing partial participation as weight vector sparsification reveals connections to well-studied techniques (Beznosikov et al., 2023). A state-of-the-art approach to handle it efficiently is error feedback (Stich and Karimireddy, 2020; Richtárik et al., 2021). Since sampling rules are represented as compressors, we believe that this idea may be extremely useful in our setting as well. However, we cannot apply the error feedback framework directly. The reason is that the sampling rules are non-contractive compressors, as they zero out certain local gradients. Formally, there does not exist $\beta < \infty$ such that $\|x - \mathcal{C}(x)\|^2 \leqslant \left(1 - \frac{1}{\beta}\right) \|x\|^2$ for $\mathcal{C}(x) = 0, x \in \mathbb{R}^d$.

Thus, we have to address the challenge of designing a scheme that can handle non-contractive compression before proceeding to a unified analysis of partial participation.

## 3.2 PARTIAL PARTICIPATION WITHOUT UNAVAILABLE DEVICES

To develop the idea proposed in Section 3.1, we present the **P**artial **P**articipation with **B**ias **C**orrection framework (PPBC, see Algorithm 1) that supports a wide class of weighting and sampling approaches. Since computing full-batch gradients is often impractical in modern applications, we also account for local stochasticity.

---

**Algorithm 1** PPBC

---

1: **Input:** Start point $x^{-1,H^{-1}} \in \mathbb{R}^d$, $g^{-1,H^{-1}} \in \mathbb{R}^d$, epochs number $K$, number of devices $M$
2: **Parameters:** Stepsize $\gamma > 0$, momentum $0 < \theta < 1$, regularization $1 \leqslant \alpha \leqslant M$
3: **for** epochs $k = 0, \ldots, K - 1$ **do**
4:     Initialize $\pi^k$ *// Server weighs clients using any procedure*
5:     $\hat{\pi}^k = \widehat{\mathcal{R}}^k(\pi^k)$ *// Server selects clients to communicate through epoch using any rule $\widehat{\mathcal{R}}$*
6:     $g_m^{k,0} = 0$ *// Each client initializes the gradient surrogate*
7:     $x^{k,0} = x^{k-1,H^{k-1}} - \gamma g^{k-1,H^{k-1}}$ *// Server initializes the initial point of the epoch*
8:     Generate $H^k \sim \text{Geom}(p)$ *// Server generates number of iterations of k-th epoch*
9:     **for** iterations $h = 0, \ldots, H^k - 1$ **do**
10:         $\widetilde{\pi}^{k,h} = \widetilde{\mathcal{R}}^{k,h}(\hat{\pi}^k)$ *// Server selects clients to communicate at the current round using rule $\widetilde{\mathcal{R}}$*
11:         **for** devices $m = 1 \ldots M$ in parallel **do**
12:             $g_m^{k,h+1} = g_m^{k,h} + (1 - \theta)\left(\frac{1}{M} - \widetilde{\pi}_m^{k,h}\right)\nabla f_m(x^{k,h}, \xi_m^{k,h})$ *// Update the gradient surrogate*
13:         **end for**
14:         **for** each device $m : \widetilde{\pi}_m^{k,h} \neq 0$ **do**
15:             Send $\nabla f_m(x^{k,h}, \xi_m^{k,h})$ to the server
16:         **end for**
17:         $x^{k,h+1} = x^{k,h} - \gamma\left[(1 - \theta)\sum\limits_{m=1}^{M}\widetilde{\pi}_m^{k,h}\nabla f_m(x^{k,h}, \xi_m^{k,h}) + \theta g^{k-1,H^{k-1}}\right]$ *// Server updates parameters*
18:     **end for**
19:     **for** devices $m = 1 \ldots M$ in parallel **do**
20:         Send $g_m^{k,H^k}$ to the server
21:     **end for**
22:     $g^{k,H^k} = \sum\limits_{m=1}^{M} g_m^{k,H^k}$ *// Server aggregates gradient surrogates*
23: **end for**

---

**Description of Algorithm 1.** In Algorithm 1, the weights $\pi^k = (\pi_1^k, \ldots, \pi_M^k)^\top$ are computed according to any of the mentioned strategies at the beginning of each epoch (Line 4). Next, the rule $\widehat{\mathcal{R}}$ is applied to determine the participating machines (Line 5). Its output $\hat{\pi}^k$ contains zeros at positions corresponding to nodes that are not chosen to communicate with the server. Note that $\widehat{\mathcal{R}}$ is not necessarily constant. There are no theoretical restrictions to change it during the execution. For example, one can vary the number of participating devices. We also allow additional client sampling at each iteration of the epoch by introducing a rule $\widetilde{\mathcal{R}}$ (Line 10). We propose to aggregate local gradient surrogates during the epoch (Line 12). To provide intuition beyond this update, we give a toy example where each $\pi_m$ is equal to $1/M$. In this way, all inactive devices collect their gradients, while all active ones retain the vector $g_m$ from the previous iteration. In the practical case with various weights, each device accounts for its deviation from the uniform distribution $\pi_u = \{1/M\}_{m=1}^M$. Next, we use the accumulated vectors during the following epoch (Line 17). To handle the magnitude imbalance between the gradient and its surrogate, we employ a smoothing scheme with a small parameter $\theta$. We provide an ablation studies regarding $\theta$ and $p$ in Appendix B.

**Analysis of Algorithm 1.** We utilize virtual sequences to derive convergence rates of PPBC. The idea is to introduce an additional vector

$$\widetilde{x}^{k,h} = x^{k,h} - \gamma \sum_{m=1}^{M} g_m^{k,h}$$

and use it to prove convergence. Substituting Lines 10, 17 in this definition, we obtain

$$\widetilde{x}^{k,h+1} = \widetilde{x}^{k,h} - \gamma\left[(1 - \theta)\frac{1}{M}\sum_{m=1}^{M}\nabla f_m(x^{k,h}, \xi_m^{k,h}) + \theta g^{k-1,H^{k-1}}\right].$$

This is an important technique for our method, since the sequence $\widetilde{x}$ is updated with the average of gradients from all devices, contrary to the original $x$. However, the virtual update also contains a combination of accumulated gradients from the previous epoch. We emphasize that handling $g^{k-1,H^{k-1}}$ is one of the main theoretical challenges we address. We set the epoch size $H^k$ as a geometrically distributed random variable and provide the following lemma.

**Lemma 3.1.** *Suppose Assumptions 2.3, 2.4 hold. We consider the epoch size $H^k \sim Geom(p)$ and $1 \leqslant \alpha \leqslant M$. Then for Algorithm 1 it implies*

$$
\mathbb{E}_{H^k} \mathbb{E}_{\xi_m^{k,0}} \dots \mathbb{E}_{\xi_m^{k,H^k-1}} \left\| g^{k,H^k} \right\|^2 \leqslant \frac{24(1-\theta)^2 \alpha(\delta_1+1)}{p^2} \mathbb{E}_{H^k} \left\| \nabla f(x^{k,H^k}) \right\|^2 + \frac{48(1-\theta)^2 \alpha \delta_2}{p^2}
$$
$$
+ \frac{24(1-\theta)^2 \alpha \sigma^2}{Mp^2}.
$$

Assumption 2.4 is required only to handle local stochasticity. If the devices are able to compute exact gradients, Lemma 3.1 holds with $\sigma = 0$. For the details, see Appendix D. As a result, we obtain the convergence theorem.

**Theorem 3.2.** *Suppose Assumptions 2.1, 2.2(a), 2.3, 2.4 hold. Then for Algorithm 1 with $\theta \leqslant \frac{\gamma L p^2}{2}$ and $\gamma \leqslant \frac{p}{384L\alpha(\delta_1+1)}$ it implies that*

$$
\frac{1}{K} \sum_{k=0}^{K-1} \mathbb{E} \left\| \nabla f(x^{k,0}) \right\|^2 \leqslant \frac{16 \left( f(x^{0,0}) - f(x^*) \right)}{\gamma K} + \frac{768\gamma L\alpha\delta_2}{p} + \frac{384\gamma^2 L^2 \alpha \delta_2}{p^3}
$$
$$
+ \frac{400\gamma L\alpha\sigma^2}{Mp} + \frac{192\gamma^2 L^2 \alpha \sigma^2}{Mp^3}.
$$

The main obstacle in proving Theorem 3.2 is the terms $\|g^{k,H^k}\|^2$ and $\|g^{k-1,H^{k-1}}\|^2$ that appear in the analysis. Using Lemma 3.1, they can be screwed to $\|\nabla f(x^{k,H^k})\|^2$ and $\|\nabla f(x^{k-1,H^{k-1}})\|^2$, respectively. The first norm is easy to analyze. Classically, it serves as a convergence criterion. Eliminating the second one turns out to be challenging. To cope with it, we incorporate the surrogate into the starting point of the epoch (Line 7). For the details, see Appendix D.1. With such an estimate, there is a technique to choose the stepsize $\gamma$ appropriately to obtain convergence (Stich, 2019).

**Corollary 3.3.** *Under conditions of Theorem 3.2 Algorithm 1 with fixed rules $\widehat{\mathcal{R}}^k \equiv \widetilde{\mathcal{R}}^{k,h} \equiv \mathcal{R}$ needs*

$$
\mathcal{O} \left( M \frac{M}{C} \left( \frac{\Delta L\alpha\delta_1}{\varepsilon^2} + \frac{\Delta L\alpha\delta_2}{\varepsilon^4} + \frac{\Delta L\alpha\sigma^2}{M\varepsilon^4} \right) \right)
$$

*number of devices communications to reach $\varepsilon$-accuracy, where $\varepsilon^2 = \frac{1}{K} \sum_{k=0}^{K-1} \mathbb{E} \left\| \nabla f(x^{k,0}) \right\|^2$, $\Delta = f(x^{0,0}) - f(x^*)$ and $C$ is the number of devices participating in each epoch.*

We also consider varying sampling rules $\widehat{\mathcal{R}}^k$ and $\widetilde{\mathcal{R}}^{k,h}$ to study corollaries of Theorem 3.2.

**Corollary 3.4.** *Under conditions of Theorem 3.2 Algorithm 1 needs*

$$
\mathcal{O} \left( \frac{M}{\min_{k,h} C^{k,h}} \left( \frac{\Delta L\alpha\delta_1}{\varepsilon^2} + \frac{\Delta L\alpha\delta_2}{\varepsilon^4} + \frac{\Delta L\alpha\sigma^2}{M\varepsilon^4} \right) \right) \text{ epochs and}
$$

$$
\mathcal{O} \left( M \left( \frac{M}{\min_{k,h} C^{k,h}} \right)^2 \left( \frac{\Delta L\alpha\delta_1}{\varepsilon^2} + \frac{\Delta L\alpha\delta_2}{\varepsilon^4} + \frac{\Delta L\alpha\sigma^2}{M\varepsilon^4} \right) \right) \text{ number of devices communications}
$$

*to reach $\varepsilon$-accuracy, where $\varepsilon^2 = \frac{1}{K} \sum_{k=0}^{K-1} \mathbb{E} \left\| \nabla f(x^{k,0}) \right\|^2$, $\Delta = f(x^{0,0}) - f(x^*)$ and $C^{k,h}$ is the number of devices participating in $k$-th iteration in $h$-th epoch.*

In our work, the analysis is extended to the strongly convex case.

**Theorem 3.5.** *Suppose Assumptions 2.1, 2.2(b), 2.3, 2.4 hold. Then for Algorithm 1 with $\theta \leqslant \frac{p\gamma\mu}{4}$ and $\gamma \leqslant \frac{p^2}{96L\alpha(\delta_1+1)}$ it implies that*

$$\mathbb{E}\left\|x^{K,0} - x^*\right\|^2 \leqslant \left(1 - \frac{\gamma\mu}{8}\right)^K \left\|x^{0,0} - x^*\right\|^2 + \frac{8\gamma\alpha}{\mu p^3}\left(144\delta_2 + \frac{74\sigma^2}{M}\right).$$

As well as for the non-convex objective, suitable $\gamma$ can be chosen in Theorem 3.5.

**Corollary 3.6.** *Under conditions of Theorem 3.5 Algorithm 1 with fixed rules $\widehat{\mathcal{R}}^{k,h} \equiv \widetilde{\mathcal{R}}^{k,h} \equiv \mathcal{R}$ needs*

$$\widetilde{\mathcal{O}}\left(M\left(\frac{M}{C}\right)^2 \left(\frac{L}{\mu}\alpha\delta_1 \log\left(\frac{1}{\varepsilon}\right) + \frac{M}{C}\frac{\alpha\delta_2}{\mu^2\varepsilon} + \frac{\alpha\sigma^2}{\mu^2 C\varepsilon}\right)\right)$$

*number of devices communications to reach $\varepsilon$-accuracy, where $\varepsilon^2 = \mathbb{E}\left\|x^{K,0} - x^*\right\|^2$ and $C$ is the number of devices participating in each epoch.*

**Corollary 3.7.** *Under conditions of Theorem 3.5 Algorithm 1 needs*

$$\widetilde{\mathcal{O}}\left(\left(\frac{M}{\min\limits_{k,h}C^{k,h}}\right)^2 \left(\frac{L}{\mu}\alpha\delta_1 \log\left(\frac{1}{\varepsilon}\right) + \frac{M}{\min\limits_{k,h}C^{k,h}}\frac{\alpha\delta_2}{\mu^2\varepsilon} + \frac{\alpha\sigma^2}{\mu^2\min\limits_{k,h}C^{k,h}\varepsilon}\right)\right) \quad \text{epochs and}$$

$$\widetilde{\mathcal{O}}\left(M\left(\frac{M}{\min\limits_{k,h}C^{k,h}}\right)^3 \left(\frac{L}{\mu}\alpha\delta_1 \log\left(\frac{1}{\varepsilon}\right) + \frac{M}{\min\limits_{k,h}C^{k,h}}\frac{\alpha\delta_2}{\mu^2\varepsilon} + \frac{\alpha\sigma^2}{\mu^2\min\limits_{k,h}C^{k,h}\varepsilon}\right)\right)$$

*number of devices communications to reach $\varepsilon$-accuracy, where $\varepsilon^2 = \mathbb{E}\left\|x^{K,0} - x^*\right\|^2$ and $C^{k,h}$ is the number of devices participating in $k$-th iteration in $h$-th epoch.*

### 3.3 PARTIAL PARTICIPATION WITH UNAVAILABLE DEVICES

The previous section addresses partial participation when all devices are available to communicate with the server. Indeed, in Algorithm 1 each node receives the current parameters at the end of the iteration, but does not send its gradient. This is motivated by the fact that forwarding a message from the client to the server is much more expensive than the other way around (Kairouz et al., 2021). However, in practice, some devices can become inactive periodically (Li et al., 2019b; Yang et al., 2021). Namely, these machines not only refrain from transmitting information but also do not perform local computations. In this section, we extend our theory to cover the case where the actual parameters are sent to only a fraction of the clients.

**Description of Algorithm 2.** In this section we present the part of Algorithm 2 (see Appendix A) that reflects key differences from Algorithm 1. To design it, we refuse using the biased sampling rule $\widetilde{\mathcal{R}}$ during the epoch. Instead, we simulate outage probability of the $m$-th device as a Bernoulli random variable $\eta_m^{k,h} \sim \mathrm{Be}(q_m)$ (Chung, 2000) (Line 11). To describe client disconnection formally, $\eta_m^{k,h}$ is used to update the gradient surrogates (Line 12) and to perform the step (Line 17). Thus, in practice, it is not necessary for an inactive device to know the actual parameters. We also normalize the computed gradients by factors $\{q_m\}_{m=1}^M$ to balance their magnitudes.

---

11: Generate $\eta^{k,h}$

12: $g_m^{k,h+1} = g_m^{k,h} + (1-\theta)\frac{\eta_m^{k,h}}{q_m}\left(\frac{1}{M} - \hat{\pi}_m^{k,h}\right)\nabla f_m(x^{k,h}, \xi_m^{k,h})$

17: $x^{k,h+1} = x^{k,h} - \gamma\left[(1-\theta)\sum\limits_{m=1}^{M}\frac{\eta_m^{k,h}}{q_m}\hat{\pi}_m^{k,h}\nabla f_m(x^{k,h}, \xi_m^{k,h}) + \theta g^{k-1,H^{k-1}}\right]$

---

**Analysis of Algorithm 2.** We formulate the results for both non-convex and strongly-convex cases.

**Corollary 3.8.** *Suppose Assumptions 2.1, 2.2(a), 2.3, 2.4 hold. Algorithm 2 with fixed rules $\widehat{\mathcal{R}}^k \equiv \widetilde{\mathcal{R}}^{k,h} \equiv \mathcal{R}$ needs*

$$\mathcal{O}\left(M\frac{M}{C}\frac{1}{\min\limits_{1\leqslant m\leqslant M}q_m}\left(\frac{\Delta L\alpha\delta_1}{\varepsilon^2} + \frac{\Delta L\alpha\delta_2}{\varepsilon^4} + \frac{\Delta L\alpha\sigma^2}{\varepsilon^4}\right)\right)$$

*number of devices communications to reach $\varepsilon$-accuracy, where $\varepsilon^2 = \frac{1}{K} \sum\limits_{k=0}^{K-1} \mathbb{E} \left\| \nabla f(x^{k,0}) \right\|^2$, $\Delta =$*

*$f(x^{0,0}) - f(x^*)$ and $C$ is the number of devices participating in each epoch.*

**Corollary 3.9.** *Under conditions of Theorem E.2 Algorithm 2 needs*

$$\mathcal{O} \left( \frac{M}{\min\limits_{k,h} C^{k,h}} \frac{1}{\min\limits_{1 \leqslant m \leqslant M} q_m} \left( \frac{\Delta L \alpha \delta_1}{\varepsilon^2} + \frac{\Delta L \alpha \delta_2}{\varepsilon^4} + \frac{\Delta L \alpha \sigma^2}{\varepsilon^4} \right) \right) \quad \textit{epochs and}$$

$$\mathcal{O} \left( M \left( \frac{M}{\min\limits_{k,h} C^{k,h}} \right)^2 \frac{1}{\min\limits_{1 \leqslant m \leqslant M} q_m} \left( \frac{\Delta L \alpha \delta_1}{\varepsilon^2} + \frac{\Delta L \alpha \delta_2}{\varepsilon^4} + \frac{\Delta L \alpha \sigma^2}{\varepsilon^4} \right) \right)$$

*number of devices communications to reach $\varepsilon$-accuracy, where $\varepsilon^2 = \frac{1}{K} \sum\limits_{k=0}^{K-1} \mathbb{E} \left\| \nabla f(x^{k,0}) \right\|^2$, $\Delta =$*

*$f(x^{0,0}) - f(x^*)$ and $C^{k,h}$ is the number of devices participating in $k$-th iteration in $h$-th epoch.*

**Corollary 3.10.** *Suppose Assumptions 2.1, 2.2(b), 2.3, 2.4 hold. Algorithm 2 with fixed rules $\widehat{\mathcal{R}}^k \equiv \widetilde{\mathcal{R}}^{k,h} \equiv \mathcal{R}$ needs*

$$\widetilde{\mathcal{O}} \left( M \left( \frac{M}{C} \right)^2 \frac{1}{\min\limits_{1 \leqslant m \leqslant M} q_m} \left( \frac{L}{\mu} \alpha \delta_1 \log \left( \frac{1}{\varepsilon} \right) + \frac{M}{C} \frac{\alpha \delta_2}{\mu^2 \varepsilon} + \frac{M}{C} \frac{\alpha \sigma^2}{\mu^2 \varepsilon} \right) \right)$$

*number of devices communications to reach $\varepsilon$-accuracy, where $\varepsilon^2 = \mathbb{E} \left\| x^{K,0} - x^* \right\|^2$ and $C$ is the number of devices participating in each epoch.*

**Corollary 3.11.** *Under conditions of Theorem E.6 Algorithm 2 needs*

$$\widetilde{\mathcal{O}} \left( \left( \frac{M}{\min\limits_{k,h} C^{k,h}} \right)^2 \frac{1}{\min\limits_{1 \leqslant m \leqslant M} q_m} \left( \frac{L}{\mu} \alpha \delta_1 \log \left( \frac{1}{\varepsilon} \right) + \frac{M}{\min\limits_{k,h} C^{k,h}} \frac{\alpha \delta_2}{\mu^2 \varepsilon} + \frac{M}{\min\limits_{k,h} C^{k,h}} \frac{\alpha \sigma^2}{\mu^2 \varepsilon} \right) \right)$$

*epochs or*

$$\widetilde{\mathcal{O}} \left( M \left( \frac{M}{\min\limits_{k,h} C^{k,h}} \right)^3 \frac{1}{\min\limits_{1 \leqslant m \leqslant M} q_m} \left( \frac{L}{\mu} \alpha \delta_1 \log \left( \frac{1}{\varepsilon} \right) + \frac{M}{\min\limits_{k,h} C^{k,h}} \frac{\alpha \delta_2}{\mu^2 \varepsilon} + \frac{M}{\min\limits_{k,h} C^{k,h}} \frac{\alpha \sigma^2}{\mu^2 \varepsilon} \right) \right)$$

*communications*

*to reach $\varepsilon$-accuracy, where $\varepsilon^2 = \mathbb{E} \left\| x^{K,0} - x^* \right\|^2$ and $C^{k,h}$ is the number of devices participating in $k$-th iteration in $h$-th epoch.*

For more details, see Appendix E. Note that $\min_{1 \leq m \leq M} q_m$ is a constant lying in the interval $(0, 1]$. Thus, the rates of Algorithm 2 do not differ significantly from those for Algorithm 1. The only deterioration occurs in the variance term associated with local stochasticity. Thus, if each device has an access to its exact gradient, there is no asymptotical difference compared to Corollaries 3.3 and 3.6.

## 3.4 DISCUSSION

We analyzed a wide class of sampling and weighting techniques and proposed algorithms for different network scenarios. Their rates asymptotically coincide with the optimal ones for SGD-like approaches (Stich, 2019). Due to considering biased strategies, we obtained an additional factor $M/C$. Again analogizing to compression, this multiplier signifies compression power. It is a well-known fact that there is no theoretical improvement for methods built upon error-feedback (Richtárik et al., 2021; Beznosikov et al., 2023). However, we recover the convergence of SGD in the case of full participation. Comparing our non-convex rate regarding the main term $\mathcal{O} \left( 1/\varepsilon^2 \right)$ with prior works, we note that it surpasses that in (Wang and Ji, 2022) $\left( \mathcal{O} \left( 1/\varepsilon^4 + \delta_2 \right) \right)$ both asymptotically and by the absence of the non-vanishing term. Next, comparing strongly-convex rates $\left( \mathcal{O} \left( \kappa \log 1/\varepsilon \right) \right)$, we are superior to (Cho et al., 2022) $\left( \mathcal{O} \left( \kappa^2/\varepsilon + \kappa \delta_2 \right) \right)$ and (Luo et al., 2022) $\left( \mathcal{O} \left( \kappa/\varepsilon \right) \right)$. Moreover, both of these works lack non-convex analysis. We highlight that we soften assumptions from all aforementioned works.

## 4 EXPERIMENTS

To validate our theoretical findings, we conduct a systematic empirical comparison of six optimization frameworks – FedAvg (Reddi et al., 2020), SCAFFOLD (Karimireddy et al., 2020), FedDyn (Chen et al., 2023), Moon (Li et al., 2021), and PPBC (Algorithm 1) — evaluated under full client participation (FCP), along with two additional frameworks – F3AST (Ribero et al., 2022) and PPBC+ (Algorithm 2) – specifically designed for and evaluated under partial client participation (PCP). Crucially, we fix the sampling strategy across all frameworks to isolate how each optimizer interacts with it, thereby decoupling the sampling mechanism from core algorithmic innovations for FCP experiments. All methods are compared under identical experimental conditions: same model architectures, benchmark datasets, and hardware configurations. The following section details the experimental setup, including architectures, datasets, and infrastructure.

**Experimental Setup.** We evaluate sampling strategies under three distinct data distribution settings: (**distr-1**) homogeneous (i.i.d.), (**distr-2**) heterogeneous (client-specific class sets), and (**distr-3**) strongly heterogeneous (varying data volumes and class skew). In this section we will present results for the most challenging setup with **distr-3**, full version of experiments is in Appendix B along with other details. Experiments use CIFAR-10 (Krizhevsky et al., 2009) with RESNET-18 (Meng et al., 2019) for image classification and FOOD101 Bossard et al. (2014) with FASTERVIT (Hatamizadeh et al., 2023) for fine-tuning, providing a controlled benchmark for comparing Algorithm 1. Importantly, each plot compares frameworks – not strategies – by fixing the underlying strategy and varying the framework. This correspondence is formalized in Algorithm 1, where the gradient surrogate term vanishes, recovering the conventional update rule. Further implementation details (partitioning, architecture, datasets) appear in Appendix B.

### 4.1 FULL CLIENT PARTICIPATION

**Client Selection Rule.** Notably, not all strategies included in our comparative analysis inherently incorporate a client selection mechanism. To ensure a fair and consistent evaluation, we uniformly applied the following selection rule across all methods:

$$\widehat{\mathcal{R}}^k = \text{Top}_C\left(\pi^k\right),$$

where $\text{Top}_C$ denotes taking $C > 0$ clients with the highest weights $\pi^k$. Consequently, the remainder of our experiments will focus exclusively on the formulation and analysis of weight update rules, while treating the client selection process itself as a fixed component of the experimental framework.

**Client Sampling.** We evaluate four established client sampling strategies, each designed to improve convergence or robustness by prioritizing clients based on different criteria. PoC (Cho et al., 2022) selects clients proportionally to their local loss values, favoring those with higher empirical risk to accelerate optimization. BANT (Xie et al., 2019) employs a trust-based mechanism, dynamically scoring clients by their historical alignment with server-side validation performance, thereby promoting reliability over time. FOLB (Nguyen et al., 2020) samples clients based on the projected utility of their updates – specifically, the inner product between local gradients and the server's global descent direction – to maximize progress per round. Finally, GNS (Wang et al., 2020b) prioritizes clients with larger gradient norms, under the intuition that clients exhibiting stronger local signals contribute more meaningfully to global updates.

Full algorithmic descriptions and implementation details for all strategies are provided in Appendix B.

**Results.** The comparative results are summarized in Table 1, with primary evaluation based on final test loss and accuracy metric. Figure 1 complements this by visualizing the training dynamics of our PPBC framework against the strongest baselines. For FedAvg and SCAFFOLD, we report their best-performing variant per sampling strat-

Table 1: Frameworks and strategies comparison on CIFAR-10 & RESNET-18.

| Method + Strategy | distr-3 | |
| --- | --- | --- |
| | Loss ($\downarrow$) | Acc ($\uparrow$) |
| FedAvg + PoC | 0.898±0.021 | 65.3±0.20 |
| FedAvg + FOLB | 0.674±0.020 | 71.42±0.19 |
| FedAvg + BANT | 2.324±0.023 | 11.32±0.25 |
| FedAvg + GNS | 0.657±0.019 | 71.15±0.19 |
| SCAFFOLD + PoC | 0.788±0.020 | 69.81±0.19 |
| SCAFFOLD + FOLB | 0.663±0.016 | 71.80±0.20 |
| SCAFFOLD + BANT | 0.698±0.017 | 71.31±0.18 |
| SCAFFOLD + GNS | 0.689±0.020 | 71.75±0.19 |
| FedDyn | 0.652±0.016 | 76.71±0.14 |
| Moon | 0.627±0.014 | 75.21±0.15 |
| PPBC + PoC | 0.367±0.019 | 88.87±0.16 |
| PPBC + FOLB | 0.362±0.016 | 88.91±0.14 |
| PPBC + BANT | 0.357±0.015 | 88.96±0.15 |
| PPBC + GNS | 0.364±0.016 | 88.90±0.15 |

*Notation:* All values averaged over 3 seeds. Arrows indicate optimization direction: $\downarrow$ minimize loss, $\uparrow$ maximize accuracy. Green color represents our algorithms.

egy, ensuring a fair and strategy-aware comparison. This allows us to isolate the impact of the optimization framework itself, independent of sampling-induced variance.

### 4.2 PARTIAL CLIENT PARTICIPATION

**Client Sampling and Partial Participation.** To simulate real-world scenarios, we model client presence at each round via independent Bernoulli trials with participation probability $q_m$. We evaluate performance across a spectrum of participation regimes, ranging from full availability ($q_m = 1$) to highly sparse communication ($q_m = 0.3$), reflecting scenarios with frequent dropouts or intermittent connectivity. To contextualize our framework's robustness under such conditions, we include comparative experiments against F3AST, an algorithm specifically designed to handle client outages and non-uniform participation.

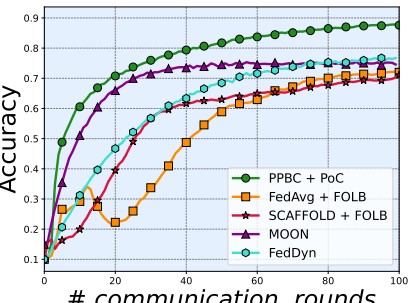

Figure 1: Comparison graphs on **distr-3** for best runs.

For PPBC+, we set server strategy $\widehat{\mathcal{R}}^k$ with the FOLB strategy and employ PoC as the client sampling mechanism $\widetilde{\mathcal{R}}^k$.

**Results.** Similarly to the previous section, results are summarized in Table 2, with the primary evaluation based on the final test loss and accuracy metrics. Figure 2 represents accuracy graphs of our PPBC+ framework (Algorithm 2) with $q_m = 0.3$ against F3AST with $q_m = 1, 0.7, 0.5$ and FedAvg with $q_m = 1$. This plot clearly demonstrates the superiority of our method over F3AST. Moreover, we highlight that even under the most challenging communication conditions ($q_m = 0.3$), our approach consistently converges to substantially higher accuracy than all competing baselines.

**Discussion.** We provided experimental validation of the theoretical convergence estimates for the proposed algorithms across a range of practical federated learning tasks. Our evaluation included large-scale models, such as the FASTERVIT architecture with 270M parameters, demonstrating the scalability and effectiveness of our approach in realistic learning scenarios. Results demonstrate a substantial performance gap between conventional approaches (FedAvg, SCAFFOLD, FedDyn, Moon) and Algorithm 1 Additionally, we analyzed the behavior of the PPBC+ (Algorithm 2) under varying client sampling conditions, confirming the robustness and consistency of its performance across different parameter $q_m$ values.

To further support our theoretical findings, we present Figure 3, which illustrates that the algorithms introduced in this work maintain comparable convergence rates across all considered configurations. These results affirm that our methods preserve efficiency and stability even when applied to heterogeneous data distributions and complex model architectures.

Table 2: Frameworks and strategies comparison on FASTERVIT & FOOD101.

| Method | distr-3 | |
|---|---|---|
| | Loss ($\downarrow$) | Acc ($\uparrow$) |
| FedAvg ($q_m = 1$) | 1.896±0.021 | 56.74±0.13 |
| F3AST ($q_m = 1$) | 1.692±0.022 | 68.31±0.11 |
| F3AST ($q_m = 0.7$) | 1.754±0.020 | 65.52±0.12 |
| F3AST ($q_m = 0.5$) | 1.812±0.018 | 61.30±0.13 |
| PPBC+ ($q_m = 1$) | 0.930±0.017 | 76.11±0.09 |
| PPBC+ ($q_m = 0.7$) | 0.937±0.018 | 76.04±0.12 |
| PPBC+ ($q_m = 0.5$) | 0.961±0.018 | 75.07±0.10 |
| PPBC+ ($q_m = 0.3$) | 0.996±0.020 | 74.68±0.11 |

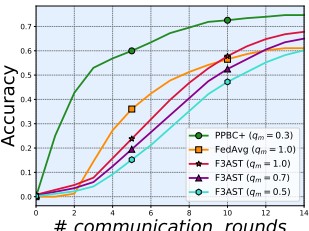

Figure 2: Comparison graphs on **distr-3** for best runs.

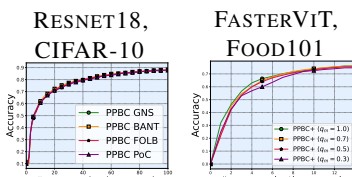

Figure 3: Test accuracy of PPBC/PPBC+ for image classification with RESNET18 on CIFAR-10 and FASTERVIT fine-tuning on FOOD101.

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

## APPENDIX

## CONTENTS

## A  PARTIAL PARTICIPATION WITH UNAVAILABLE DEVICES

In this section, we present Algorithm 2, which is the complete version of algorithm from Section 3.3. This method can be applied to environments where devices do not perform local computations periodically.

---

**Algorithm 2** PPBC+

---

1: **Input:** Start point $x^{-1,H^{-1}} \in \mathbb{R}^d$, $g^{-1,H^{-1}} \in \mathbb{R}^d$, epochs number $K$, number of devices $M$
2: **Parameters:** Stepsize $\gamma > 0$, momentum $0 < \theta < 1$, regularization $1 \leqslant \alpha \leqslant M$
3: **for** epochs $k = 0, \ldots, K - 1$ **do**
4:     Initialize $\pi^k$ // *Server weighs clients using any procedure*
5:     $\hat{\pi}^k = \widehat{\mathcal{R}}^k(\pi^k)$ // *Server selects clients to communicate through epoch using any rule $\hat{\mathcal{R}}$*
6:     $g_m^{k,0} = 0$ // *Each client initializes the gradient surrogate*
7:     $x^{k,0} = x^{k-1,H^{k-1}} - \gamma g^{k-1,H^{k-1}}$ // *Server initializes the initial point of the epoch*
8:     Generate $H^k \sim \text{Geom}(p)$ // *Server generates number of iterations of k-th epoch*
9:     **for** iterations $h = 0, \ldots, H^k - 1$ **do**
10:         **for** devices $m = 1 \ldots M$ in parallel **do**
11:             Generate $\eta_m^{k,h} \sim \mathcal{B}(q_m)$ // *Device generates its state: available / unavailable*
12:             $g_m^{k,h+1} = g_m^{k,h} + (1-\theta) \frac{\eta_m^{k,h}}{q_m} \left(\frac{1}{M} - \hat{\pi}_m^{k,h}\right) \nabla f_m(x^{k,h}, \xi_m^{k,h})$ // *Update the gradient surrogate*
13:         **end for**
14:         **for** each device $m : \eta_m^{k,h} \neq 0$ and $\hat{\pi}_m^k \neq 0$ **do**
15:             Send $\frac{\eta_m^{k,h}}{q_m} \nabla f_m(x^{k,h}, \xi_m^{k,h})$ to the server
16:         **end for**
17:         $x^{k,h+1} = x^{k,h} - \gamma \left[ (1-\theta) \sum_{m=1}^{M} \frac{\eta_m^{k,h}}{q_m} \hat{\pi}_m^{k,h} \nabla f_m(x^{k,h}, \xi_m^{k,h}) + \theta g^{k-1,H^{k-1}} \right]$ // *Server updates parameters*
18:     **end for**
19:     **for** devices $m = 1 \ldots M$ in parallel **do**
20:         Send $g_m^{k,H^k}$ to the server
21:     **end for**
22:     $g^{k,H^k} = \sum_{m=1}^{M} g_m^{k,H^k}$ // *Server aggregates gradient surrogates*
23: **end for**

---

## B  ADDITIONAL EXPERIMENTS AND DETAILS

Our code is available at `https://anonymous.4open.science/r/EF25_ICLR/`.

**Hardware Details.**   The experiments were conducted using Python with the PyTorch deep learning framework (Paszke et al., 2017). The computational hardware consisted of a server equipped with an Intel Xeon Gold 6342 CPU and two NVIDIA A100 40GB GPUs. The total runtime for all experimental evaluations amounted to approximately 80 hours. To simulate a federated learning environment, data was distributed across clients based on a heterogeneity parameter.

**Data Distribution.**   In our study, we employed 10 clients for both the RESNET-18 on CIFAR-10 setup and the FASTERVIT fine-tuning on the FOOD101 dataset. This client count was carefully chosen to enable comprehensive evaluation across the diverse data distribution scenarios proposed in our work, while maintaining computational feasibility for thorough experimentation. Below, we provide a detailed summary of the data distribution characteristics for each experimental setup.

Homogeneous data distribution (**distr-1**) – each client has the same number of data samples, and class labels are uniformly distributed across clients.

*Example (CIFAR-10):* Each client has 500 training samples per class, resulting in 5,000 samples per client in total.

Heterogeneous data distribution (**distr-2**) – each client has the same total number of samples, but class labels are distributed in a non-IID manner.

*Example (CIFAR-10):* We split the 10 classes into two disjoint groups (e.g., classes 0-4 and 5-9), and assign clients to one of the two groups. Clients in each group receive data only from their assigned classes. Additionally, the number of samples per class varies across clients.

Pathological data distribution (**distr-3**) – clients possess different amounts of data. The distribution of sample proportions across clients is as follows:

Table 3: Client-wise data sample proportions in **distr-3**.

| Client no. | Proportion |
|---|---|
| 1 | 10.6% |
| 2 | 7.4% |
| 3 | 12.0% |
| 4 | 11.4% |
| 5 | 8.8% |
| 6 | 14.6% |
| 7 | 10.0% |
| 8 | 5.4% |
| 9 | 10.2% |
| 10 | 9.2% |

Within each client, class labels are sampled according to a Dirichlet distribution with concentration parameter $\alpha = 0.5$, resulting in highly non-IID label distributions.

Next, we provide a detailed overview of the client sampling strategies and present comparative results for `FedAvg`, `SCAFFOLD`, and Algorithm 1. We exclude `FedDyn` and `Moon` from this analysis, as their designs incorporate fixed strategies that cannot be decoupled from their core update rules.

**Loss-aware Client Sampling.** Building upon previous work, Cho et al. (2022) introduced the POWER-OF-CHOICE (PoC) strategy, which employs a weighted client sampling mechanism based on local loss values. Formally, the weight update rule can be expressed as:

1. The server assigns to all clients the probabilities proportional to the data size fractions

$$p_m = \frac{n_m}{\left( \sum_{m'=1}^{M} n_{m'} \right)}.$$

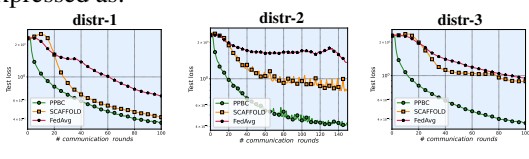

(a) Convergence comparison.

2. The global model is sent by the server to the selected $C$ clients, which compute and return their local loss values based on their datasets. Subsequently, the weights are updated:

$$\pi^k = \left( \left[ \frac{1}{n_m} \sum_{i_m=1}^{n_m} \ell(g(x, a_{i_m}), b_{i_m}) \right] \right)_{m=1}^{M}.$$

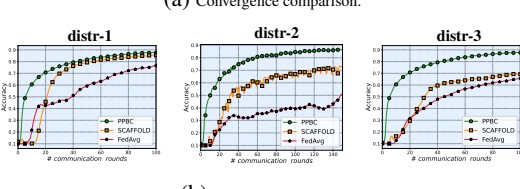

(b) Metrics comparison.

Figure 4: Performance comparison for `PoC` strategy with different data distributions.

**Trust-Score Sampling.** The study by Xie et al. (2019) introduces the `BANT`, which implements a trust-based sampling mechanism. This approach assigns dynamic trust scores to clients based on historical performance metrics. Thus, weight update rule can be described as:

1. The server assigns trust scores $\text{TS}_m^k$ to each client $m$ based on the alignment of their model updates with the performance on server-held ground truth data $\mathcal{V}$:

$$\text{TS}_m^k = \exp \left[ -\frac{1}{|\mathcal{V}|} \sum_{\xi \in \mathcal{V}} f_m(x^k, \xi) \right].$$

2. The weights are updated with a probability proportional to trust scores:

$$\pi^k = \left( \frac{\text{TS}_m^k}{\sum_{m'=1}^{M} \text{TS}_{m'}^k} \right)_{m=1}^{M}.$$

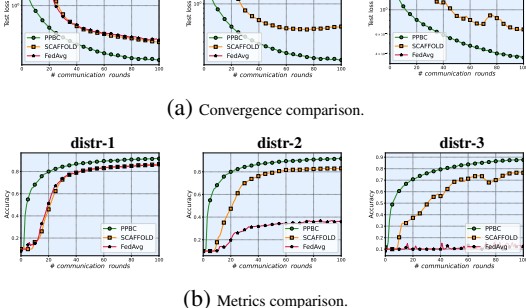

(a) Convergence comparison.

(b) Metrics comparison.

Figure 5: Performance comparison for `BANT` strategy with different data distributions.

**Importance Sampling.** Nguyen et al. (2020) introduced FOLB, a theoretically grounded client selection framework for federated learning that optimizes convergence by sampling clients proportionally to the expected utility of their local updates. The core selection mechanism operates as follows:

1. Each client is assigned an importance score $\text{IS}_m^k$ proportional to the inner product between its gradient $\nabla f_m(x^k, \xi_m^k)$ and the direction of the server model improvement (previous gradient $d^k$):

$$\text{IS}_m^k = \left| \left\langle \nabla f_m(x^k, \xi_m^k), d^k \right\rangle \right|.$$

2. The weights are updated with a probability proportional to the trust scores for each client:

$$\pi^k = \left( \frac{\text{IS}_m^k}{\sum\limits_{m'=1}^{M} \text{IS}_{m'}^k} \right)_{m=1}^{M}.$$

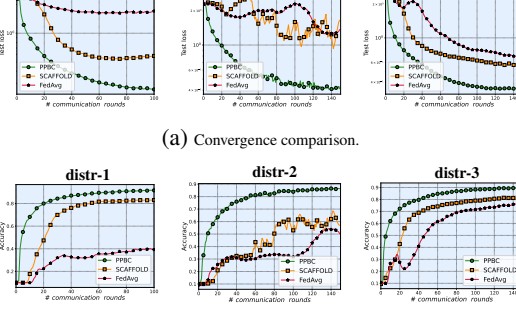

(a) Convergence comparison.

(b) Metrics comparison.

Figure 6: Performance comparison for FOLB strategy with different data distributions.

**Gradient-Norm-Based Sampling.** For the image classification problem on CIFAR-10 dataset, we introduce an alternative client sampling strategy based on gradient norm sampling GNS Wang et al. (2020b), which prioritizes clients whose local updates exhibit larger magnitudes. In particular:

1. At each communication round $k$, the server estimates the relative importance of each client $m$ using the norm of its reported gradient $\nabla f_m(w^k, \xi_m^k)$:

$$p_m^k = \frac{\left\| \nabla f_m(w^k, \xi_m^k) \right\|_2}{\sum_{m'=1}^{M} \left\| \nabla f_{m'}(w^k, \xi_{m'}^k) \right\|_2}.$$

2. Clients are then sampled with probabilities proportional to $\{p_m^k\}_{m=1}^{M}$, ensuring that those with larger gradient norms are selected more frequently:

$$\pi^k = \left( p_m^k \right)_{m=1}^{M}.$$

The obtained comparison results are presented in Figures 4, 6, 5, and 7.

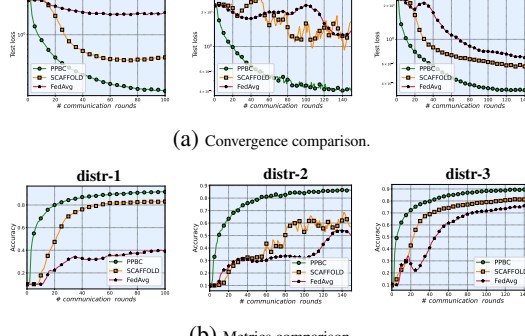

(a) Convergence comparison.

(b) Metrics comparison.

Figure 7: Performance comparison for GNS strategy with different data distributions.

**ViT Fine-tuning.** To further assess the generalization and adaptability of our method, we conduct additional experiments involving the fine-tuning of a state-of-the-art Vision Transformer architecture FASTERVIT (Hatamizadeh et al., 2023). The model, pre-trained on the large-scale IMAGENET21K dataset (Ridnik et al., 2021), comprises approximately 270M parameters and integrates hybrid hierarchical-attention mechanisms for efficient multi-scale feature learning. We fine-tune this model on the FOOD101 dataset (Bossard et al., 2014), a challenging benchmark consisting of 101,000 images across 101 fine-grained food categories. This dataset presents significant visual complexity due to high class variation and subtle inter-class distinctions, making it particularly suitable for evaluating the scalability of our method.

Table 4: Summary of training strategies used in additional experiments. Top and Rand denote the client selection rules, where the number indicates how many clients were selected for training.

| Epoch Strategy | Round Strategy |
|---|---|
| GNS (Top 3) | PoC (Top 1) |
| FOLB (Top 3) | PoC (Top 1) |
| PoC (Top 3) | Rand 1 |
| FOLB (Top 3) | Rand 1 |

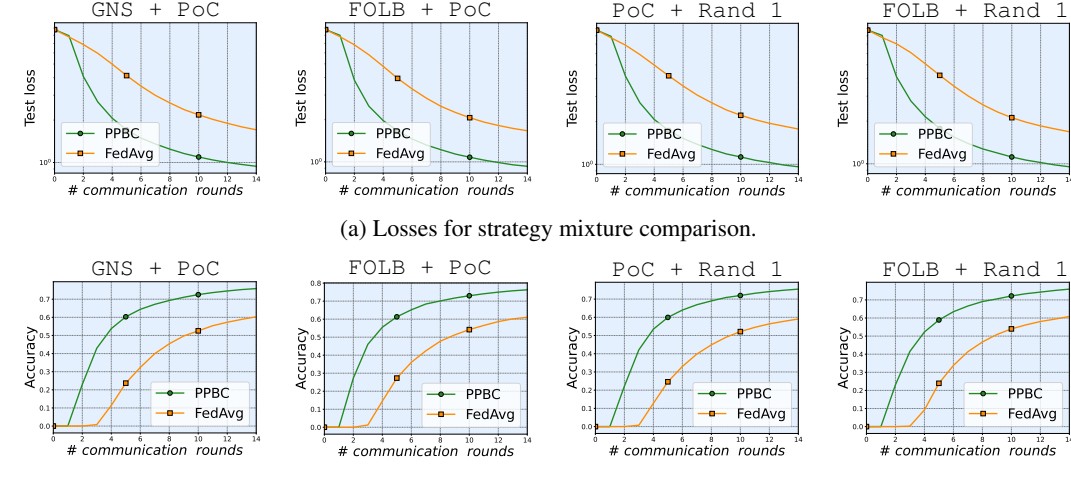

(a) Losses for strategy mixture comparison.

(b) Metrics (Accuracy@1) for strategy mixture comparison.

Figure 8: Performance comparison for combination of strategies on FASTERVIT fine-tuning.

**Strategy Mixture.** In the preceding experimental setups, we restricted our evaluation to a fixed, server-based client sampling strategy. However, as demonstrated in our theoretical analysis, Algorithm 1 is flexible enough to accommodate a broader class of sampling mechanisms, potentially varying across communication rounds. To validate this flexibility empirically, we conduct additional experiments for FASTERVIT fine-tuning on **distr-3** data distribution. We consider this setup to be the most challenging one, because strong heterogeneity with different amount of samples and classes per client and various strategies makes the FedAvg and SCAFFOLD algorithms behave similarly. Therefore, our further experimental comparisons will only include FedAvg. We allow the sampling rule $\widetilde{\mathcal{R}}^{k,h}$ to change dynamically at each communication round $k$. The combinations of strategies are presented in Table 4. The performance validation results for each strategy mixture can be observed in Figure 8.

**Ablation Study on Hyperparameters.** Our framework admits a unifying interpretation: by setting $\theta = 0$ and disabling the client weighting mechanism, we recover the original baseline methods (FedAvg + any client sampling strategy). Consequently, by varying $\theta$ we can obtain various performance of Algorithm 1. Our method also utilizes another hyperparameter: the duration between global aggregations (length of the local epochs) $H^k$, modeled as a geometrically distributed random variable with parameter $p$. Our theoretical analysis imposes no constraints on $p$; convergence guarantees hold for any choice, with rates explicitly dependent on this hyperparameter (see Theorems 3.5, 3.2, E.2, E.6). Next, we conduct an ablation study on both hyperparameters $\theta, p$ to quantify their impact on performance. Moreover, we demonstrate the empirical connection between $\theta$ and $p$, which correlates with our theoretical findings.

Firstly, we provide ablation study on $\theta$. We fix $p = 0.2$ (yielding $H^k = 5$) and vary $\theta$ under the GNS client selection rule. Results are shown in Table 5.

Table 5: Ablation on $\theta$ with $H^k = 5$.

| $\theta$ | Accuracy | Loss |
|---|---|---|
| 0.05 | 0.88 | 0.35 |
| 0.10 | 0.90 | 0.31 |
| 0.15 | **0.93** | **0.21** |
| 0.20 | 0.89 | 0.32 |

We confirm our theoretical expectations: excessively small values of $\theta$ do not allow for effectively accounting for the clients' history ($\theta = 0$ corresponds to FedAvg), while large values disproportionately increases the contribution of gradient surrogates that become outdated after an epoch. However, there exists a wide interval within which the method do not lose much quality compared to optimal $\theta$ value.

Next, we fix $\theta = 0.2$ and vary $p$ (i.e., the expected epoch size $H^k = 1/p$), with results in Table 6.

Table 6: Ablation on local epoch size with $\theta = 0.2$.

| $H^k$ | Accuracy | Loss |
|---|---|---|
| 1 | 0.81 | 0.38 |
| 3 | **0.91** | **0.23** |
| 5 | 0.89 | 0.32 |
| 7 | 0.82 | 0.39 |

For $\theta = 0.15$, the optimal local epoch size is $H^k = 5$ (see Table 6), while for $\theta = 0.2$, the optimal value decreases to $H^k = 3$. This finding is in complete agreement with theoretical expectations: bigger values of $\theta$ require fewer number of local steps to achieve optimal convergence.

**Ablation Study on Convergence.** In this paragraph, we emphasize that the proposed Algorithm 1 maintains similar convergence behavior across all combinations of the considered strategies (see Figure 9). This result is obtained by gradient compensation technique incorporated in our method. Thus, a biases that appear due to applying client sampling strategies are equally mitigated by our algorithm.

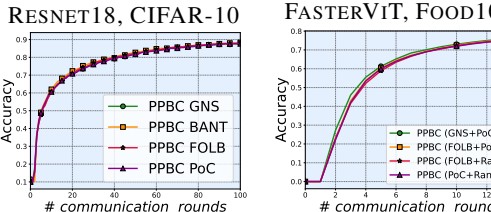

Figure 9: Test accuracy of PPBC for image classification with RESNET18 on CIFAR-10 and FASTERVIT fine-tuning on FOOD101.

**PPBC does not require a fixed aggregation round.** Our algorithms PPBC and PPBC+ have one limitation: they require transmitting all accumulated surrogates once per epoch. For this reason, we conduct an experimental study (PP$\hat{\text{B}}$C) in which we remove the requirement that all devices must send their information every fixed number of iterations.

We introduced an additional mechanism: at the moment of full aggregation, a client may choose not to send the surrogate it accumulated during the epoch. This is modeled similarly to PPBC+, using a Bernoulli random variable with a new hyperparameter $q_e$. In other words, any client may fail to provide its surrogate during the full aggregation step. Consequently, line 20 of the Algorithm 2 is modified to the following block:

$$\text{generate } \eta_m^k \sim \mathcal{Q}_e, \qquad \text{if } \eta_m^k = 1: \text{ send } g_m^{k,H_k} \text{ to the server}$$

We conducted experiments (see Figure 10) for different values of both $q_m$ and $q_e$, and compared our results with standard PPBC+ (Algorithm 2) using $q_m = 0.3$, as well as with the baseline FedAvg under pathological data heterogeneity. As expected, the new algorithm performs worse than PPBC+ with full aggregation, yet it still consistently outperforms FedAvg.

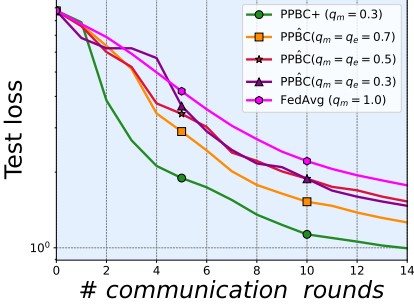
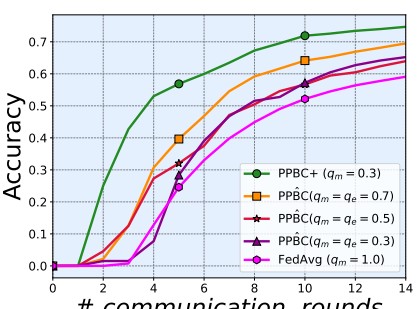

Figure 10: Test loss and test accuracy of PP$\hat{\text{B}}$C, PPBC+, and FedAvg on FASTERVIT fine-tuning on FOOD101.

## C GENERAL STATEMENTS

**Notation.** In the work we use the following notation. $x^{k,h} \in \mathbb{R}^d$ is the vector of model's parameters in $h$-th iteration in $k$-th epoch, $\nabla f_m(x) \in \mathbb{R}^d$ represents the gradient of function $f_m$ at the point $x \in \mathbb{R}^d$, $\nabla f_m(x, \xi) \in \mathbb{R}^d$ denotes the stochastic gradient at the point $x \in \mathbb{R}^d$ with respect to stochastic realization $\xi$.

For a random vector $x \in \mathbb{R}^d$ and stochasticity $\xi$ we denote $\mathbb{E}[x]$ is the expected value of $x$ and $\mathbb{E}_\xi[x]$ as the conditioned expected value with the respect to $\xi$.

We use $\|x\| = \sqrt{\sum_{i=1}^{d} x_i^2}$ as $l_2$-norm of the vector $x \in \mathbb{R}^d$ and $\langle x, y \rangle = \sum_{i=1}^{d}$ represents the scalar product of vectors $x, y \in \mathbb{R}^d$.

We use *number of devices communications* (device to server communications) as the metric. This choice arises from the recognition that the number of rounds of communication is insufficient to adequately compare distributed methods. For example, this limitation becomes evident when the nodes operate asynchronously. In this case, the more appropriate metric is the total number of communications rather than the number of rounds.

**General inequalities.** Suppose $x, y, \{a_i\}_{i=1}^n \in \mathbb{R}^d, \{\omega_i\}_{i=1}^n \in \mathbb{R}$, $f(\cdot)$ inherent to Assumptions 2.1, 2.2(b), $\varphi(\cdot)$ is under Assumption 2.2(b). Then,

$$\|\nabla f(x) - \nabla f(y)\|^2 \leqslant 2L\left(f(x) - f(y) - \langle \nabla f(y), x - y \rangle\right), \tag{Lip}$$

$$\langle x, y \rangle \leqslant \frac{\beta}{2}\|x\|^2 + \frac{1}{2\beta}\|y\|^2, \tag{Fen}$$

$$\left\|\sum_{i=1}^{n} a_i\right\|^2 \leqslant n\sum_{i=1}^{n}\|a_i\|^2, \tag{CS}$$

$$\varphi\left(\frac{\sum_{i=1}^{n} w_i a_i}{\sum_{i=1}^{n} a_i}\right) \leqslant \frac{\sum_{i=1}^{n} a_i \varphi(x_i)}{\sum_{i=1}^{n} a_i}. \tag{Jen}$$

**Lemma C.1** ((Allen-Zhu, 2018)). *Given sequence $D_0, D_1, \ldots D_N \in \mathbb{R}$, where $N \in Geom(p)$. Then,*

$$\mathbb{E}_N[D_{N-1}] = pD_0 + (1-p)\mathbb{E}_N[D_N].$$

## D PROOFS FOR ALGORITHM 1

**Lemma D.1 (Lemma 3.1).** *Suppose Assumptions 2.3, 2.4 hold. Then for Algorithm 1 it implies that*

$$\mathbb{E}_{H^k}\mathbb{E}_{\xi_m^{k,0}}\ldots\mathbb{E}_{\xi_m^{k,H^k-1}}\left\|g^{k,H^k}\right\|^2 \leqslant \frac{24(1-\theta)^2\alpha(\delta_1+1)}{p^2}\mathbb{E}_{H^k}\left\|\nabla f(x^{k,H^k})\right\|^2 + \frac{48(1-\theta)^2\alpha\delta_2}{p^2}$$
$$+ \frac{24(1-\theta)^2\alpha\sigma^2}{Mp^2}.$$

*Proof.* Let us start with the following estimate:

$$\left\|g^{k,h+1}\right\|^2 = \left\|g^{k,h} + (1-\theta)\sum_{m=1}^{M}\left(\frac{1}{M} - \widetilde{\pi}_m^{k,h}\right)\nabla f_m(x^{k,h}, \xi_m^{k,h})\right\|^2$$

$$\overset{(Fen)}{\leqslant} (1+c)\left\|g^{k,h}\right\|^2$$

$$+ \left(1 + \frac{1}{c}\right)(1-\theta)^2\left\|\sum_{m=1}^{M}\left(\frac{1}{M} - \widetilde{\pi}_m^{k,h}\right)\nabla f_m(x^{k,h}, \xi_m^{k,h})\right\|^2, \tag{4}$$

where $c$ is defined below. Let us estimate the last term and obtain

$$\left\|\sum_{m=1}^{M}\left(\frac{1}{M} - \widetilde{\pi}_m^{k,h}\right)\nabla f_m(x^{k,h}, \xi_m^{k,h})\right\|^2$$

$$\overset{(CS)}{\leqslant} 2 \left\| \sum_{m=1}^{M} \left( \frac{1}{M} - \widetilde{\pi}_m^{k,h} \right) \nabla f_m(x^{k,h}) \right\|^2$$

$$+ 2 \left\| \sum_{m=1}^{M} \left( \frac{1}{M} - \widetilde{\pi}_m^{k,h} \right) \left[ \nabla f_m(x^{k,h}, \xi_m^{k,h}) - \nabla f_m(x^{k,h}) \right] \right\|^2$$

$$\overset{(i)}{=} 2 \left\| \sum_{m=1}^{M} \left( \frac{1}{M} - \widetilde{\pi}_m^{k,h} \right) \nabla f_m(x^{k,h}) - \sum_{m=1}^{M} \left( \frac{1}{M} - \pi_m^k \right) \nabla f(x^{k,h}) \right\|^2$$

$$+ 2 \left\| \sum_{m=1}^{M} \left( \frac{1}{M} - \widetilde{\pi}_m^{k,h} \right) \left[ \nabla f_m(x^{k,h}, \xi_m^{k,h}) - \nabla f_m(x^{k,h}) \right] \right\|^2.$$

Adding and subtracting $\sum_{m=1}^{M} \widetilde{\pi}_m^{k,h} \nabla f(x^{k,h})$ in the first term yields

$$\left\| \sum_{m=1}^{M} \left( \frac{1}{M} - \widetilde{\pi}_m^{k,h} \right) \nabla f_m(x^{k,h}, \xi_m^{k,h}) \right\|^2$$

$$\leqslant 2 \left\| \sum_{m=1}^{M} \left( \frac{1}{M} - \widetilde{\pi}_m^{k,h} \right) \left[ \nabla f_m(x^{k,h}) - \nabla f(x^{k,h}) \right] - \sum_{m=1}^{M} \left( \widetilde{\pi}_m^{k,h} - \pi_m^k \right) \nabla f(x^{k,h}) \right\|^2$$

$$+ 2 \left\| \sum_{m=1}^{M} \left( \frac{1}{M} - \widetilde{\pi}_m^{k,h} \right) \left[ \nabla f_m(x^{k,h}, \xi_m^{k,h}) - \nabla f_m(x^{k,h}) \right] \right\|^2$$

$$\overset{(CS)}{\leqslant} 4 \left\| \sum_{m=1}^{M} \left( \frac{1}{M} - \widetilde{\pi}_m^{k,h} \right) \left[ \nabla f_m(x^{k,h}) - \nabla f(x^{k,h}) \right] \right\|^2$$

$$+ 4 \left\| \sum_{m=1}^{M} \left( \widetilde{\pi}_m^{k,h} - \pi_m^k \right) \nabla f(x^{k,h}) \right\|^2$$

$$+ 2 \left\| \sum_{m=1}^{M} \left( \frac{1}{M} - \widetilde{\pi}_m^{k,h} \right) \left[ \nabla f_m(x^{k,h}, \xi_m^{k,h}) - \nabla f_m(x^{k,h}) \right] \right\|^2.$$

We apply equation CS to the first term and identically transform the second and third terms:

$$\left\| \sum_{m=1}^{M} \left( \frac{1}{M} - \widetilde{\pi}_m^{k,h} \right) \nabla f_m(x^{k,h}, \xi_m^{k,h}) \right\|^2$$

$$\leqslant 4 \sum_{m=1}^{M} \left( \frac{1}{M} - \widetilde{\pi}_m^{k,h} \right)^2 \sum_{m=1}^{M} \left\| \nabla f_m(x^{k,h}) - \nabla f(x^{k,h}) \right\|^2$$

$$+ 4 \left( \sum_{m=1}^{M} \left( \widetilde{\pi}_m^{k,h} - \pi_m^k \right) \right)^2 \left\| \nabla f(x^{k,h}) \right\|^2$$

$$+ 2 \sum_{m=1}^{M} \left( \frac{1}{M} - \widetilde{\pi}_m^{k,h} \right)^2 \left\| \nabla f_m(x^{k,h}, \xi_m^{k,h}) - \nabla f_m(x^{k,h}) \right\|^2$$

$$+ 4 \sum_{i \neq j} \left( \frac{1}{M} - \widetilde{\pi}_i^{k,h} \right) \left( \frac{1}{M} - \widetilde{\pi}_j^{k,h} \right)$$

$$\cdot \left\langle \nabla f_i(x^{k,h}, \xi_i^{k,h}) - \nabla f_i(x^{k,h}), \nabla f_j(x^{k,h}, \xi_j^{k,h}) - \nabla f_j(x^{k,h}) \right\rangle$$

$$\overset{\substack{\text{As. 2.3} \\ (ii)}}{\leqslant} 4M \left( \delta_1 \left\| \nabla f(x^{k,h}) \right\|^2 + \delta_2 \right) \sum_{m=1}^{M} \left( \frac{1}{M} - \widetilde{\pi}_m^{k,h} \right)^2 + 4 \left\| \nabla f(x^{k,h}) \right\|^2$$

$$+2 \sum_{m=1}^{M} \left( \frac{1}{M} - \widetilde{\pi}_m^{k,h} \right)^2 \left\| \nabla f_m(x^{k,h}, \xi_m^{k,h}) - \nabla f_m(x^{k,h}) \right\|^2$$

$$+4 \sum_{i \neq j} \left( \frac{1}{M} - \widetilde{\pi}_i^{k,h} \right) \left( \frac{1}{M} - \widetilde{\pi}_j^{k,h} \right) \cdot$$

$$\cdot \left\langle \nabla f_i(x^{k,h}, \xi_i^{k,h}) - \nabla f_i(x^{k,h}), \nabla f_j(x^{k,h}, \xi_j^{k,h}) - \nabla f_j(x^{k,h}) \right\rangle,$$

where $(i)$ was made due to $\sum_{m=1}^{M} \left( \frac{1}{M} - \pi_m^k \right) = 1 - 1 = 0$, $(ii)$ with respect to $\sum_{m=1}^{M} \left( \widetilde{\pi}_m^{k,h} - \pi_m^k \right) \leqslant 1$. Taking expectation on $\xi_m^{k,h}$ and using Assumption 2.4, we have

$$\mathbb{E}_{\xi_m^{k,h}} \left\| \sum_{m=1}^{M} \left( \frac{1}{M} - \widetilde{\pi}_m^{k,h} \right) \nabla f_m(x^{k,h}, \xi_m^{k,h}) \right\|^2 \leqslant 4M\delta_1 \left\| \nabla f(x^{k,h}) \right\|^2 \sum_{m=1}^{M} \left( \frac{1}{M} - \widetilde{\pi}_m^{k,h} \right)^2$$

$$+4 \left\| \nabla f(x^{k,h}) \right\|^2$$

$$+4M\delta_2 \sum_{m=1}^{M} \left( \frac{1}{M} - \widetilde{\pi}_m^{k,h} \right)^2$$

$$+2\sigma^2 \sum_{m=1}^{M} \left( \frac{1}{M} - \widetilde{\pi}_m^{k,h} \right)^2, \quad (5)$$

since $\xi_i^{k,h}$ and $\xi_j^{k,h}$ are independent random variables and, consequently, the scalar product equals to zero.

We use $\pi^k \in \Delta_1^M \cap \left( \bigcap_{m=1}^{M} \left\{ \pi : e_m^\top \pi + \frac{\alpha}{M} \geqslant 0 \right\} \right)$, where $1 \leqslant \alpha \leqslant M$ and $\{e_m\}_{m=1}^{M}$ is the unit basis. In this way, worst case in terms of average distance from $\frac{1}{M}$ is realization, where $\lfloor \frac{M}{\alpha} \rfloor$ weights are $\frac{\alpha}{M}$ and the rest are zero. In such a case, we can estimate

$$\sum_{m=1}^{M} \left( \frac{1}{M} - \widetilde{\pi}_m^{k,h} \right)^2 \leqslant \left\lfloor \frac{M}{\alpha} \right\rfloor \frac{(\alpha-1)^2}{M^2} + \left( M - \left\lfloor \frac{M}{\alpha} \right\rfloor \right) \frac{1}{M^2}$$

$$\leqslant \frac{M}{\alpha} \frac{(\alpha-1)^2}{M^2} + \left( M - \frac{M}{\alpha} + 1 \right) \frac{1}{M^2}$$

$$= \frac{\alpha-1}{M} + \frac{1}{M^2} \leqslant \frac{\alpha}{M}. \quad (6)$$

We can transform equation 5 into

$$\mathbb{E}_{\xi_m^{k,h}} \left\| \sum_{m=1}^{M} \left( \frac{1}{M} - \widetilde{\pi}_m^{k,h} \right) \nabla f_m(x^{k,h}) \right\|^2 \leqslant 4\alpha(\delta_1+1) \left\| \nabla f(x^{k,h}) \right\|^2 + 4\alpha\delta_2 + \frac{2\alpha\sigma^2}{M}. \quad (7)$$

Substituting equation 7 into equation 4, we have

$$\mathbb{E}_{\xi_m^{k,h}} \left\| g^{k,h+1} \right\|^2 \leqslant (1+c) \left\| g^{k,h} \right\|^2 + 4 \left( 1 + \frac{1}{c} \right) (1-\theta)^2 \alpha(\delta_1+1) \left\| \nabla f(x^{k,h}) \right\|^2$$

$$+4 \left( 1 + \frac{1}{c} \right) (1-\theta)^2 \alpha\delta_2$$

$$+2 \left( 1 + \frac{1}{c} \right) (1-\theta)^2 \frac{\alpha}{M} \sigma^2.$$

Enrolling a recursion, we get

$$\mathbb{E}_{\xi_m^{k,0}} \dots \mathbb{E}_{\xi_m^{k,h}} \left\| g^{k,h+1} \right\|^2 \leqslant 4 \left( 1 + \frac{1}{c} \right) (1-\theta)^2 \alpha(\delta_1+1) \sum_{i=0}^{h} (1+c)^{h-i} \left\| \nabla f(x^{k,i}) \right\|^2$$

$$+4\left(1+\frac{1}{c}\right)(1-\theta)^2\alpha\delta_2\sum_{i=0}^{h}(1+c)^{h-i}$$

$$+2\left(1+\frac{1}{c}\right)(1-\theta)^2\frac{\alpha}{M}\sigma^2\sum_{i=0}^{h}(1+c)^{h-i}. \tag{8}$$

Now we use that $H^k \sim \mathrm{Geom}(p)$:

$$\mathbb{E}_{H^k}\mathbb{E}_{\xi_m^{k,0}}\ldots\mathbb{E}_{\xi_m^{k,H^k-1}}\left\|g^{k,H^k}\right\|^2 = \sum_{j\geqslant 0}p(1-p)^j\mathbb{E}_{\xi_m^{k,0}}\ldots\mathbb{E}_{\xi_m^{k,j-1}}\left\|g^{k,j}\right\|^2$$

$$\overset{(8)}{\leqslant} 4\left(1+\frac{1}{c}\right)(1-\theta)^2\alpha(\delta_1+1)\cdot$$

$$\cdot\sum_{j\geqslant 0}p(1-p)^j\sum_{i=0}^{j-1}(1+c)^{j-i-1}\left\|\nabla f(x^{k,i})\right\|^2$$

$$+2\left(1+\frac{1}{c}\right)(1-\theta)^2\frac{\alpha}{M}\left(\sigma^2+2M\delta_2\right)\cdot$$

$$\cdot\sum_{j\geqslant 0}p(1-p)^j\sum_{i=0}^{j-1}(1+c)^{j-i-1}. \tag{9}$$

Let us choose $c=\frac{p}{2}$ and consider the following term individually:

$$\sum_{j\geqslant 0}p(1-p)^j\sum_{i=0}^{j-1}(1+c)^{j-i-1}\left\|\nabla f(x^{k,i})\right\|^2 = p\Big[(1-p)\left\|\nabla f(x^{k,0})\right\|^2$$

$$+(1-p)^2\left\{(1+c)\left\|\nabla f(x^{k,0})\right\|^2+\left\|\nabla f(x^{k,1})\right\|^2\right\}+\ldots\Big]$$

$$= p(1-p)\Big[(1-p)^0(1+c)^0+(1-p)(1+c)+\ldots\Big]\left\|\nabla f(x^{k,0})\right\|^2$$

$$+p(1-p)^2\Big[(1-p)^0(1+c)^0+(1-p)(1+c)+\ldots\Big]\left\|\nabla f(x^{k,1})\right\|^2+\ldots$$

$$\leqslant \sum_{l\geqslant 0}(1-p)^l(1+\frac{p}{2})^l\sum_{j\geqslant 0}p(1-p)^{j+1}\left\|\nabla f(x^{k,j})\right\|^2$$

$$\leqslant \frac{1}{1-(1-p)(1+\frac{p}{2})}\sum_{j\geqslant 0}p(1-p)^j\left\|\nabla f(x^{k,j})\right\|^2 = \frac{2}{p(p+1)}\mathbb{E}_{H^k}\left\|\nabla f(x^{k,H^k})\right\|^2$$

$$\leqslant \frac{2}{p}\mathbb{E}_{H^k}\left\|\nabla f(x^{k,H^k})\right\|^2. \tag{10}$$

Additionally, we have

$$\sum_{j\geqslant 0}p(1-p)^j\sum_{i=0}^{j-1}(1+c)^{j-i-1} \leqslant p\sum_{j\geqslant 0}(1-p)^j j(1+\frac{p}{2})^j \leqslant p\sum_{j\geqslant 0}j\left(1-\frac{p}{2}\right)^j$$

$$= p\frac{1-\frac{p}{2}}{\left(1-\left(1-\frac{p}{2}\right)\right)^2} \leqslant \frac{4}{p}. \tag{11}$$

Combining this estimates with equation 9 we obtain the result of the lemma:

$$\mathbb{E}_{H^k}\mathbb{E}_{\xi_m^{k,0}}\ldots\mathbb{E}_{\xi_m^{k,H^k-1}}\left\|g^{k,H^k}\right\|^2 \leqslant \frac{24(1-\theta)^2\alpha(\delta_1+1)}{p^2}\mathbb{E}_{H^k}\left\|\nabla f(x^{k,H^k})\right\|^2 + \frac{48(1-\theta)^2\alpha\delta_2}{p^2}$$

$$+ \frac{24(1-\theta)^2 \alpha \sigma^2}{Mp^2}.$$

$\square$

## D.1 PROOF FOR NON-CONVEX CASE

**Theorem D.2** (**Theorem 3.2**). *Suppose Assumptions 2.1, 2.2(a), 2.3, 2.4 hold. Then for Algorithm 1 with $\theta \leqslant \frac{\gamma L p^2}{2}$ and $\gamma \leqslant \frac{p}{384 L \alpha (\delta_1 + 1)}$ it implies that*

$$\frac{1}{K} \sum_{k=0}^{K-1} \mathbb{E} \left\| \nabla f(x^{k,0}) \right\|^2 \quad \leqslant \quad \frac{16 \left( f(x^{0,0}) - f(x^*) \right)}{\gamma K} + \frac{768 \gamma L \alpha \delta_2}{p} + \frac{384 \gamma^2 L^2 \alpha \delta_2}{p^3}$$

$$+ \frac{400 \gamma L \alpha \sigma^2}{Mp} + \frac{192 \gamma^2 L^2 \alpha \sigma^2}{Mp^3}.$$

*Proof.* We start with the definition of virtual sequence:

$$\widetilde{x}^{k,h} = x^{k,h} - \gamma \sum_{m=1}^{M} g_m^{k,h} = x^{k,h} - \gamma g^{k,h}. \tag{12}$$

It is followed by

$$\widetilde{x}^{k,h+1} = x^{k,h+1} - \gamma \sum_{m=1}^{M} g_m^{k,h+1} = x^{k,h} - \gamma \left[ (1-\theta) \sum_{m=1}^{M} \widetilde{\pi}_m^{k,h} \nabla f_m(x^{k,h}, \xi_m^{k,h}) + \theta g^{k-1,H^{k-1}} \right]$$

$$- \gamma \sum_{m=1}^{M} g_m^{k,h} - \gamma (1-\theta) \sum_{m=1}^{M} \left( \frac{1}{M} - \widetilde{\pi}_m^{k,h} \right) \nabla f_m(x^{k,h}, \xi_m^{k,h})$$

$$= \widetilde{x}^{k,h} - \gamma \left[ (1-\theta) \frac{1}{M} \sum_{m=1}^{M} \nabla f_m(x^{k,h}, \xi_m^{k,h}) + \theta g^{k-1,H^{k-1}} \right]. \tag{13}$$

Assumption 2.1 implies

$$f(\widetilde{x}^{k,h+1}) \quad \leqslant \quad f(\widetilde{x}^{k,h}) + \left\langle \nabla f(\widetilde{x}^{k,h}), \widetilde{x}^{k,h+1} - \widetilde{x}^{k,h} \right\rangle + \frac{L}{2} \left\| \widetilde{x}^{k,h+1} - \widetilde{x}^{k,h} \right\|^2$$

$$\overset{\substack{(13) \\ (Jen)}}{\leqslant} \quad f(\widetilde{x}^{k,h}) - \gamma \theta \left\langle \nabla f(\widetilde{x}^{k,h}), g^{k-1,H^{k-1}} \right\rangle$$

$$- \gamma(1-\theta) \left\langle \nabla f(\widetilde{x}^{k,h}), \frac{1}{M} \sum_{m=1}^{M} \nabla f_m(x^{k,h}, \xi_m^{k,h}) \right\rangle$$

$$+ \frac{\gamma^2 L (1-\theta)}{2} \left\| \frac{1}{M} \sum_{m=1}^{M} \nabla f_m(x^{k,h}, \xi_m^{k,h}) \right\|^2 + \frac{\gamma^2 L \theta}{2} \left\| g^{k-1,H^{k-1}} \right\|^2.$$

Taking expectation over $\xi_m^{k,h}$, we have

$$\mathbb{E}_{\xi_m^{k,h}} \left[ f(\widetilde{x}^{k,h+1}) \right] \quad \leqslant \quad \mathbb{E}_{\xi_m^{k,h}} \left[ f(\widetilde{x}^{k,h}) \right] - \gamma \theta \mathbb{E}_{\xi_m^{k,h}} \left\langle \nabla f(\widetilde{x}^{k,h}), g^{k-1,H^{k-1}} \right\rangle$$

$$- \gamma(1-\theta) \mathbb{E}_{\xi_m^{k,h}} \left\langle \nabla f(\widetilde{x}^{k,h}), \frac{1}{M} \sum_{m=1}^{M} \nabla f_m(x^{k,h}, \xi_m^{k,h}) \right\rangle$$

$$+ \frac{\gamma^2 L (1-\theta)}{2} \mathbb{E}_{\xi_m^{k,h}} \left\| \frac{1}{M} \sum_{m=1}^{M} \nabla f_m(x^{k,h}, \xi_m^{k,h}) \right\|^2$$

$$+ \frac{\gamma^2 L \theta}{2} E_{\xi_m^{k,h}} \left\| g^{k-1,H^{k-1}} \right\|^2. \tag{14}$$

Note that

$$\widetilde{x}^{k,h} \quad \overset{(12)}{=} \quad x^{k,h} - \gamma g^{k,h}$$

$$\overset{\text{Line 12}}{=} \quad x^{k,h} - \gamma \left( g^{k,h-1} + (1-\theta) \sum_{m=1}^{M} \left( \frac{1}{M} - \tilde{\pi}_m^{k,h-1} \right) \nabla f(x^{k,h-1}, \xi_m^{k,h-1}) \right).$$

Thus, $\tilde{x}^{k,h}$ and $\xi_m^{k,h}$ are independent. Analogously, $g^{k-1,H^{k-1}}$ and $\xi_m^{k,h}$ are independent. In this way, equation 14 transforms into

$$
\begin{aligned}
\mathbb{E}_{\xi_m^{k,h}} \left[ f(\tilde{x}^{k,h+1}) \right] \quad \leqslant \quad & f(\tilde{x}^{k,h}) - \gamma\theta \left\langle \nabla f(\tilde{x}^{k,h}), g^{k-1,H^{k-1}} \right\rangle \\
& -\gamma(1-\theta) \left\langle \nabla f(\tilde{x}^{k,h}), \nabla f(x^{k,h}) \right\rangle \\
& + \frac{\gamma^2 L(1-\theta)}{2} \mathbb{E}_{\xi_m^{k,h}} \left\| \frac{1}{M} \sum_{m=1}^{M} \nabla f_m(x^{k,h}, \xi_m^{k,h}) \right\|^2 \\
& + \frac{\gamma^2 L\theta}{2} \left\| g^{k-1,H^{k-1}} \right\|^2 \\
\overset{(CS)}{\leqslant} \quad & f(\tilde{x}^{k,h}) - \gamma\theta \left\langle \nabla f(\tilde{x}^{k,h}), g^{k-1,H^{k-1}} \right\rangle \\
& -\gamma(1-\theta) \left\langle \nabla f(\tilde{x}^{k,h}), \nabla f(x^{k,h}) \right\rangle \\
& + \gamma^2 L(1-\theta) \mathbb{E}_{\xi_m^{k,h}} \left\| \frac{1}{M} \sum_{m=1}^{M} \left( \nabla f_m(x^{k,h}, \xi_m^{k,h}) - \nabla f_m(x^{k,h}) \right) \right\|^2 \\
& + \gamma^2 L(1-\theta) \left\| \nabla f(x^{k,h}) \right\|^2 \\
& + \frac{\gamma^2 L\theta}{2} \left\| g^{k-1,H^{k-1}} \right\|^2.
\end{aligned}
\tag{15}
$$

Now we pay attention to the following term:

$$
\begin{aligned}
\mathbb{E}_{\xi_m^{k,h}} & \left\| \frac{1}{M} \sum_{m=1}^{M} \left( \nabla f_m(x^{k,h}, \xi_m^{k,h}) - \nabla f_m(x^{k,h}) \right) \right\|^2 \\
\overset{(i)}{=} \quad & \frac{1}{M^2} \sum_{m=1}^{M} \mathbb{E}_{\xi_m^{k,h}} \left\| \nabla f_m(x^{k,h}, \xi_m^{k,h}) - \nabla f_m(x^{k,h}) \right\|^2 \\
& + \frac{2}{M^2} \sum_{i \neq j} \left\langle \mathbb{E}_{\xi_i^{k,h}} \left[ \nabla f_i(x^{k,h}, \xi_i^{k,h}) - \nabla f_i(x^{k,h}) \right], \mathbb{E}_{\xi_j^{k,h}} \left[ \nabla f_j(x^{k,h}, \xi_j^{k,h}) - \nabla f_j(x^{k,h}) \right] \right\rangle \\
\overset{\text{As. 2.4}}{\leqslant} \quad & \frac{1}{M} \sigma^2,
\end{aligned}
$$

where $(i)$ is correct, since $\xi_i^{k,h}$ and $\xi_j^{k,h}$ are independent. Substituting this estimate into equation 15, we have

$$
\begin{aligned}
\mathbb{E}_{\xi_m^{k,h}} \left[ f(\tilde{x}^{k,h+1}) \right] \quad \leqslant \quad & f(\tilde{x}^{k,h}) - \gamma\theta \left\langle \nabla f(\tilde{x}^{k,h}), g^{k-1,H^{k-1}} \right\rangle \\
& -\gamma(1-\theta) \left\langle \nabla f(\tilde{x}^{k,h}), \nabla f(x^{k,h}) \right\rangle \\
& + \gamma^2 L(1-\theta) \left\| \nabla f(x^{k,h}) \right\|^2 + \frac{\gamma^2 L\theta}{2} \left\| g^{k-1,H^{k-1}} \right\|^2 \\
& + \frac{\gamma^2 L(1-\theta)\sigma^2}{M}.
\end{aligned}
\tag{16}
$$

Let us estimate the scalar products separately.

$$
\begin{aligned}
-\gamma(1-\theta) \left\langle \nabla f(\tilde{x}^{k,h}), \nabla f(x^{k,h}) \right\rangle \quad = \quad & -\frac{\gamma(1-\theta)}{2} \left\| \nabla f(\tilde{x}^{k,h}) \right\|^2 - \frac{\gamma(1-\theta)}{2} \left\| \nabla f(x^{k,h}) \right\|^2 \\
& + \frac{\gamma(1-\theta)}{2} \left\| \nabla f(\tilde{x}^{k,h}) - \nabla f(x^{k,h}) \right\|^2
\end{aligned}
$$

$$\overset{As.\ 2.1}{\leqslant}\quad -\frac{\gamma(1-\theta)}{2}\left\|\nabla f(\widetilde{x}^{k,h})\right\|^2 - \frac{\gamma(1-\theta)}{2}\left\|\nabla f(x^{k,h})\right\|^2$$

$$+\frac{\gamma L^2(1-\theta)}{2}\left\|\widetilde{x}^{k,h}-x^{k,h}\right\|^2$$

$$\overset{(12)}{=}\quad -\frac{\gamma(1-\theta)}{2}\left\|\nabla f(\widetilde{x}^{k,h})\right\|^2 - \frac{\gamma(1-\theta)}{2}\left\|\nabla f(x^{k,h})\right\|^2$$

$$+\frac{\gamma^3 L^2(1-\theta)}{2}\left\|g^{k,h}\right\|^2,$$

$$-\gamma\theta\left\langle\nabla f(\widetilde{x}^{k,h}),g^{k-1,H^{k-1}}\right\rangle\quad\overset{(Fen)}{\leqslant}\quad\frac{\gamma\theta}{2}\left\|\nabla f(\widetilde{x}^{k,h})\right\|^2+\frac{\gamma\theta}{2}\left\|g^{k-1,H^{k-1}}\right\|^2.$$

Combining it with equation 16, we have

$$\mathbb{E}_{\xi_m^{k,h}}\left[f(\widetilde{x}^{k,h+1})\right]\quad\leqslant\quad f(\widetilde{x}^{k,h})-\frac{\gamma(1-\theta)}{2}\left(1-2\gamma L\right)\left\|\nabla f(x^{k,h})\right\|^2-\frac{\gamma(1-2\theta)}{2}\left\|\nabla f(\widetilde{x}^{k,h})\right\|^2$$

$$+\frac{\gamma^3 L^2(1-\theta)}{2}\left\|g^{k,h}\right\|^2+\frac{\gamma\theta(\gamma L+1)}{2}\left\|g^{k-1,H^{k-1}}\right\|^2+\frac{\gamma^2 L(1-\theta)\sigma^2}{M}.$$

Now we put $h=H^k-1$ and take additional expectations.

$$\mathbb{E}_{\xi_m^{k-1,0}}\ldots\mathbb{E}_{\xi_m^{k,H^k-1}}\left[f(\widetilde{x}^{k,H^k})\right]$$

$$\leqslant\mathbb{E}_{\xi_m^{k-1,0}}\ldots\mathbb{E}_{\xi_m^{k,H^k-1}}\left[f(\widetilde{x}^{k,H^k-1})\right]$$

$$-\frac{\gamma(1-\theta)}{2}\left(1-2\gamma L\right)\mathbb{E}_{\xi_m^{k-1,0}}\ldots\mathbb{E}_{\xi_m^{k,H^k-1}}\left\|\nabla f(x^{k,H^k-1})\right\|^2$$

$$-\frac{\gamma(1-2\theta)}{2}\mathbb{E}_{\xi_m^{k-1,0}}\ldots\mathbb{E}_{\xi_m^{k,H^k-1}}\left\|\nabla f(\widetilde{x}^{k,H^k-1})\right\|^2$$

$$+\frac{\gamma^3 L^2(1-\theta)}{2}\mathbb{E}_{\xi_m^{k-1,0}}\ldots\mathbb{E}_{\xi_m^{k,H^k-1}}\left\|g^{k,H^k-1}\right\|^2$$

$$+\frac{\gamma\theta(\gamma L+1)}{2}\mathbb{E}_{\xi_m^{k-1,0}}\ldots\mathbb{E}_{\xi_m^{k-1,H^{k-1}-1}}\left\|g^{k-1,H^{k-1}}\right\|^2$$

$$+\frac{\gamma^2 L(1-\theta)\sigma^2}{M}.$$

We take expectation with respect to $H^{k-1}$ and $H^k$, and apply Lemma C.1:

$$\mathbb{E}_{H^{k-1}}\mathbb{E}_{H^k}\mathbb{E}_{\xi_m^{k-1,0}}\ldots\mathbb{E}_{\xi_m^{k,H^k-1}}\left[f(\widetilde{x}^{k,H^k})\right]$$

$$\leqslant(1-p)\mathbb{E}_{H^{k-1}}\mathbb{E}_{H^k}\mathbb{E}_{\xi_m^{k-1,0}}\ldots\mathbb{E}_{\xi_m^{k,H^k-1}}\left[f(\widetilde{x}^{k,H^k})\right]$$

$$+p\mathbb{E}_{H^{k-1}}\mathbb{E}_{\xi_m^{k-1,0}}\ldots\mathbb{E}_{\xi_m^{k-1,H^{k-1}-1}}\left[f(\widetilde{x}^{k,0})\right]$$

$$-\frac{\gamma(1-\theta)p}{2}\left(1-2\gamma L\right)\mathbb{E}_{H^{k-1}}\mathbb{E}_{\xi_m^{k-1,0}}\ldots\mathbb{E}_{\xi_m^{k-1,H^{k-1}-1}}\left\|\nabla f(x^{k,0})\right\|^2$$

$$-\frac{\gamma(1-\theta)(1-p)}{2}\left(1-2\gamma L\right)\mathbb{E}_{H^{k-1}}\mathbb{E}_{H^k}\mathbb{E}_{\xi_m^{k-1,0}}\ldots\mathbb{E}_{\xi_m^{k,H^k-1}}\left\|\nabla f(x^{k,H^k})\right\|^2$$

$$-\frac{\gamma(1-2\theta)p}{2}\mathbb{E}_{H^{k-1}}\mathbb{E}_{\xi_m^{k-1,0}}\ldots\mathbb{E}_{\xi_m^{k-1,H^{k-1}-1}}\left\|\nabla f(\widetilde{x}^{k,0})\right\|^2$$

$$-\frac{\gamma(1-2\theta)(1-p)}{2}\mathbb{E}_{H^{k-1}}\mathbb{E}_{H^k}\mathbb{E}_{\xi_m^{k-1,0}}\ldots\mathbb{E}_{\xi_m^{k,H^k-1}}\left\|\nabla f(\widetilde{x}^{k,H^k})\right\|^2$$

$$+\frac{\gamma^3 L^2(1-\theta)p}{2}\mathbb{E}_{H^{k-1}}\mathbb{E}_{\xi_m^{k-1,0}}\ldots\mathbb{E}_{\xi_m^{k-1,H^{k-1}-1}}\underbrace{\left\|g^{k,0}\right\|^2}_{=0}$$

$$+\frac{\gamma^3 L^2(1-\theta)(1-p)}{2}\mathbb{E}_{H^{k-1}}\mathbb{E}_{H^k}\mathbb{E}_{\xi_m^{k-1,0}}\ldots\mathbb{E}_{\xi_m^{k,H^k-1}}\left\|g^{k,H^k}\right\|^2$$

$$+ \frac{\gamma\theta(\gamma L + 1)}{2} \mathbb{E}_{H^{k-1}} \mathbb{E}_{\xi_m^{k-1,0}} \ldots \mathbb{E}_{\xi_m^{k-1,H^{k-1}-1}} \left\| g^{k-1,H^{k-1}} \right\|^2$$

$$+ \frac{\gamma^2 L(1-\theta)\sigma^2}{M}.$$

Next, we put $\gamma \leqslant \frac{1}{4L}$ and $\theta \leqslant \frac{1}{2}$. Moreover, we use that $H^k$ and $\left\{ \xi_m^{k-1,h} \right\}_{h=0}^{H^{k-1}-1}$ are independent stochastic values.

$$\mathbb{E}_{H^{k-1}} \mathbb{E}_{H^k} \mathbb{E}_{\xi_m^{k-1,0}} \ldots \mathbb{E}_{\xi_m^{k,H^k-1}} \left[ f(\widetilde{x}^{k,H^k}) \right]$$

$$\leqslant (1-p) \mathbb{E}_{H^{k-1}} \mathbb{E}_{H^k} \mathbb{E}_{\xi_m^{k-1,0}} \ldots \mathbb{E}_{\xi_m^{k,H^k-1}} \left[ f(\widetilde{x}^{k,H^k}) \right]$$

$$+ p \mathbb{E}_{H^{k-1}} \mathbb{E}_{\xi_m^{k-1,0}} \ldots \mathbb{E}_{\xi_m^{k-1,H^{k-1}-1}} \left[ f(\widetilde{x}^{k,0}) \right]$$

$$- \frac{\gamma(1-\theta)p}{4} \mathbb{E}_{H^{k-1}} \mathbb{E}_{\xi_m^{k-1,0}} \ldots \mathbb{E}_{\xi_m^{k-1,H^{k-1}-1}} \left\| \nabla f(x^{k,0}) \right\|^2$$

$$- \frac{\gamma(1-\theta)(1-p)}{4} \mathbb{E}_{H^{k-1}} \mathbb{E}_{H^k} \mathbb{E}_{\xi_m^{k-1,0}} \ldots \mathbb{E}_{\xi_m^{k,H^k-1}} \left\| \nabla f(x^{k,H^k}) \right\|^2$$

$$+ \frac{\gamma^3 L^2(1-\theta)(1-p)}{2} \mathbb{E}_{H^{k-1}} \mathbb{E}_{\xi_m^{k-1,0}} \ldots \mathbb{E}_{\xi_m^{k-1,H^{k-1}-1}} \mathbb{E}_{H^k} \mathbb{E}_{\xi_m^{k,0}} \ldots \mathbb{E}_{\xi_m^{k,H^k-1}} \left\| g^{k,H^k} \right\|^2$$

$$+ \gamma\theta \mathbb{E}_{H^{k-1}} \mathbb{E}_{\xi_m^{k-1,0}} \ldots \mathbb{E}_{\xi_m^{k-1,H^{k-1}-1}} \left\| g^{k-1,H^{k-1}} \right\|^2$$

$$+ \frac{\gamma^2 L(1-\theta)\sigma^2}{M}. \tag{17}$$

We use Lemma 3.1 to estimate $\left\| g^{k,H^k} \right\|^2$ and $\left\| g^{k-1,H^{k-1}} \right\|^2$. We obtain

$$\mathbb{E}_{H^k} \mathbb{E}_{\xi_m^{k,0}} \ldots \mathbb{E}_{\xi_m^{k,H^k-1}} \left\| g^{k,H^k} \right\|^2 \leqslant \frac{24(1-\theta)^2 \alpha(\delta_1+1)}{p^2} \mathbb{E}_{H^k} \left\| \nabla f(x^{k,H^k}) \right\|^2 + \frac{48(1-\theta)^2 \alpha\delta_2}{p^2}$$

$$+ \frac{24(1-\theta)^2 \alpha\sigma^2}{Mp^2}. \tag{18}$$

As for $\left\| g^{k-1,H^{k-1}} \right\|^2$, we have

$$\mathbb{E}_{H^{k-1}} \mathbb{E}_{\xi_m^{k-1,0}} \ldots \mathbb{E}_{\xi_m^{k-1,H^{k-1}-1}} \left\| g^{k-1,H^{k-1}} \right\|^2$$

$$\leqslant \frac{24(1-\theta)^2 \alpha(\delta_1+1)}{p^2} \mathbb{E}_{H^{k-1}} \left\| \nabla f(x^{k-1,H^{k-1}}) \right\|^2 + \frac{48(1-\theta)^2 \alpha\delta_2}{p^2} + \frac{24(1-\theta)^2 \alpha\sigma^2}{Mp^2}$$

$$\overset{(CS)}{\leqslant} \frac{48(1-\theta)\alpha(\delta_1+1)}{p^2} \mathbb{E}_{H^{k-1}} \left\| \nabla f(x^{k-1,H^{k-1}}) - \nabla f(\widetilde{x}^{k-1,H^{k-1}}) \right\|^2$$

$$+ \frac{48(1-\theta)^2 \alpha(\delta_1+1)}{p^2} \mathbb{E}_{H^{k-1}} \left\| \nabla f(\widetilde{x}^{k-1,H^{k-1}}) \right\|^2 + \frac{48(1-\theta)^2 \alpha\delta_2}{p^2} + \frac{24(1-\theta)^2 \alpha\sigma^2}{Mp^2}$$

$$\overset{\text{As. 2.1}}{\leqslant} \frac{48L^2(1-\theta)^2 \alpha(\delta_1+1)}{p^2} \mathbb{E}_{H^{k-1}} \left\| x^{k-1,H^{k-1}} - \widetilde{x}^{k-1,H^{k-1}} \right\|^2$$

$$+ \frac{48(1-\theta)^2 \alpha(\delta_1+1)}{p^2} \left\| \nabla f(x^{k,0}) \right\|^2 + \frac{48(1-\theta)^2 \alpha\delta_2}{p^2} + \frac{24(1-\theta)^2 \alpha\sigma^2}{Mp^2}$$

$$\overset{(12)}{=} \frac{48\gamma^2 L^2(1-\theta)^2 \alpha(\delta_1+1)}{p^2} \mathbb{E}_{H^{k-1}} \left\| g^{k-1,H^{k-1}} \right\|^2$$

$$+ \frac{48(1-\theta)^2 \alpha(\delta_1+1)}{p^2} \left\| \nabla f(x^{k,0}) \right\|^2 + \frac{48(1-\theta)^2 \alpha\delta_2}{p^2} + \frac{24(1-\theta)^2 \alpha\sigma^2}{Mp^2}.$$

We choose $\gamma \leqslant \frac{p}{96L\sqrt{\alpha}\sqrt{\delta_1+1}}$. Moreover, we take additional expectations and again use that $H^{k-1}$ and $\left\{ \xi_m^{k-1,h} \right\}_{h=0}^{H^{k-1}-1}$ are independent stochastic values:

$$\mathbb{E}_{H^{k-1}}\mathbb{E}_{\xi_m^{k-1,0}}\ldots\mathbb{E}_{\xi_m^{k-1,H^{k-1}-1}}\left\|g^{k-1,H^{k-1}}\right\|^2$$

$$\leqslant \frac{96(1-\theta)^2\alpha(\delta_1+1)}{p^2}\mathbb{E}_{\xi_m^{k-1,0}}\ldots\mathbb{E}_{\xi_m^{k-1,H^{k-1}-1}}\left\|\nabla f(x^{k,0})\right\|^2 + \frac{96(1-\theta)^2\alpha\delta_2}{p^2}$$

$$+ \frac{48(1-\theta)^2\alpha\sigma^2}{Mp^2}. \tag{19}$$

Now we substitute equation 18 and equation 19 into equation 17:

$$p\mathbb{E}_{H^{k-1}}\mathbb{E}_{H^k}\mathbb{E}_{\xi_m^{k-1,0}}\ldots\mathbb{E}_{\xi_m^{k,H^k-1}}\left[f(\widetilde{x}^{k,H^k})\right]$$

$$\leqslant p\mathbb{E}_{H^{k-1}}\mathbb{E}_{\xi_m^{k-1,0}}\ldots\mathbb{E}_{\xi_m^{k-1,H^{k-1}-1}}\left[f(\widetilde{x}^{k,0})\right]$$

$$- \frac{\gamma(1-\theta)p}{4}\mathbb{E}_{H^{k-1}}\mathbb{E}_{\xi_m^{k-1,0}}\ldots\mathbb{E}_{\xi_m^{k-1,H^{k-1}-1}}\left\|\nabla f(x^{k,0})\right\|^2$$

$$- \frac{\gamma(1-\theta)(1-p)}{4}\mathbb{E}_{H^{k-1}}\mathbb{E}_{H^k}\mathbb{E}_{\xi_m^{k-1,0}}\ldots\mathbb{E}_{\xi_m^{k,H^k-1}}\left\|\nabla f(x^{k,H^k})\right\|^2$$

$$+ \frac{12\gamma^3L^2(1-\theta)^3(1-p)\alpha(\delta_1+1)}{p^2}\mathbb{E}_{H^{k-1}}\mathbb{E}_{\xi_m^{k-1,0}}\ldots\mathbb{E}_{\xi_m^{k-1,H^{k-1}-1}}\mathbb{E}_{H^k}\left\|\nabla f(x^{k,H^k})\right\|^2$$

$$+ \frac{24\gamma^3L^2(1-\theta)^3(1-p)\alpha\delta_2}{p^2} + \frac{12\gamma^3L^2(1-\theta)^3(1-p)\alpha\sigma^2}{Mp^2}$$

$$+ \frac{96\gamma\theta(1-\theta)^2\alpha(\delta_1+1)}{p^2}\mathbb{E}_{\xi_m^{k-1,0}}\ldots\mathbb{E}_{\xi_m^{k-1,H^{k-1}-1}}\left\|\nabla f(x^{k,0})\right\|^2$$

$$+ \frac{96\gamma\theta(1-\theta)^2\alpha\delta_2}{p^2} + \frac{48\gamma\theta(1-\theta)^2\alpha\sigma^2}{Mp^2} + \frac{\gamma^2L(1-\theta)\sigma^2}{M}.$$

We take the full expectation, then use a law of expectation and rearrange terms:

$$p\mathbb{E}\left[f(\widetilde{x}^{k,H^k})\right] \leqslant p\mathbb{E}\left[f(\widetilde{x}^{k,0})\right]$$

$$- \frac{\gamma(1-\theta)(1-p)}{4}\left(1 - \frac{48\gamma^2L^2(1-\theta)^2\alpha(\delta_1+1)}{p^2}\right)\mathbb{E}\left\|\nabla f(x^{k,H^k})\right\|^2$$

$$- \frac{\gamma(1-\theta)p}{4}\left(1 - \frac{384\theta(1-\theta)\alpha(\delta_1+1)}{p^3}\right)\mathbb{E}\left\|\nabla f(x^{k,0})\right\|^2$$

$$+ \frac{24\gamma^3L^2(1-\theta)^3(1-p)\alpha\delta_2}{p^2} + \frac{96\gamma\theta(1-\theta)^2\alpha\delta_2}{p^2}$$

$$+ \frac{\gamma^2L(1-\theta)\sigma^2}{M} + \frac{12\gamma^3L^2(1-\theta)^3(1-p)\alpha\sigma^2}{Mp^2} + \frac{48\gamma\theta(1-\theta)^2\alpha\sigma^2}{Mp^2}.$$

We choose $\theta \leqslant \frac{\gamma Lp^2}{2}$ and $\gamma \leqslant \frac{p}{384L\alpha(\delta_1+1)}$. Note that all previous transitions hold even with larger choice of $\theta$ and $\gamma$, consequently this choice is correct. In that way, we obtain

$$\frac{\gamma(1-\theta)p}{8}\mathbb{E}\left\|\nabla f(x^{k,0})\right\|^2 \leqslant p\mathbb{E}\left[f(\widetilde{x}^{k,0}) - f(\widetilde{x}^{k,H^k})\right]$$

$$+ \frac{24\gamma^3L^2\alpha\delta_2}{p^2} + 48\gamma^2L\alpha\delta_2$$

$$+ \frac{\gamma^2L\sigma^2}{M} + \frac{12\gamma^3L^2\alpha\sigma^2}{Mp^2} + \frac{24\gamma^2L\alpha\sigma^2}{M}.$$

Note that $\widetilde{x}^{k,H^k} = x^{k,H^k} - \gamma g^{k,H^k} = x^{k+1,0}$ and $\widetilde{x}^{k,0} = x^{k,0}$. Thus,

$$\frac{\gamma(1-\theta)}{8}\mathbb{E}\left\|\nabla f(x^{k,0})\right\|^2 \leqslant \mathbb{E}\left[f(x^{k,0}) - f(x^{k+1,0})\right] + \frac{48\gamma L\alpha\delta_2}{p} + \frac{24\gamma^2L^2\alpha\delta_2}{p^3}$$

$$+\frac{25\gamma^2 L\alpha\sigma^2}{Mp} + \frac{12\gamma^3 L^2\alpha\sigma^2}{Mp^3}.$$

Averaging over all epochs, we obtain the result of the theorem:

$$
\begin{aligned}
\frac{1}{K}\sum_{k=0}^{K-1}\mathbb{E}\left\|\nabla f(x^{k,0})\right\|^2 \;\leqslant\;\;& \frac{8\left(f(x^{0,0}) - \mathbb{E}\left[f(x^{K,0})\right]\right)}{\gamma(1-\theta)K} + \frac{384\gamma L\alpha\delta_2}{p(1-\theta)} + \frac{192\gamma^2 L^2\alpha\delta_2}{p^3(1-\theta)} \\
& + \frac{200\gamma L\alpha\sigma^2}{Mp(1-\theta)} + \frac{96\gamma^2 L^2\alpha\sigma^2}{Mp^3(1-\theta)} \\
\leqslant\;\;& \frac{16\left(f(x^{0,0}) - f(x^*)\right)}{\gamma K} + \frac{768\gamma L\alpha\delta_2}{p} + \frac{384\gamma^2 L^2\alpha\delta_2}{p^3} \\
& + \frac{400\gamma L\alpha\sigma^2}{Mp} + \frac{192\gamma^2 L^2\alpha\sigma^2}{Mp^3}.
\end{aligned}
$$

$\square$

**Corollary D.3** (**Corollary 3.3**). *Under conditions of Theorem 3.2 Algorithm 1 with fixed rules* $\widehat{\mathcal{R}}^k \equiv \widetilde{\mathcal{R}}^{k,h} \equiv \mathcal{R}$ *needs*

$$\mathcal{O}\left(\frac{M}{C}\left(\frac{\Delta L\alpha\delta_1}{\varepsilon^2} + \frac{\Delta L\alpha\delta_2}{\varepsilon^4} + \frac{\Delta L\alpha\sigma^2}{M\varepsilon^4}\right)\right) \;\; epochs \; and$$

$$\mathcal{O}\left(M\frac{M}{C}\left(\frac{\Delta L\alpha\delta_1}{\varepsilon^2} + \frac{\Delta L\alpha\delta_2}{\varepsilon^4} + \frac{\Delta L\alpha\sigma^2}{M\varepsilon^4}\right)\right) \;\; number \; of \; devices \; communications$$

*to reach* $\varepsilon$-*accuracy, where* $\varepsilon^2 = \frac{1}{K}\sum_{k=0}^{K-1}\mathbb{E}\left\|\nabla f(x^{k,0})\right\|^2$, $\Delta = f(x^{0,0}) - f(x^*)$ *and* $C$ *is the number of devices participating in each epoch.*

*Proof.* Using the result of Theorem 3.2, we choose

$$
\gamma \leqslant \min\left\{ \frac{p}{384L\alpha(\delta_1 + 1)}, \; \frac{\sqrt{(f(x^{0,0}) - f(x^*))p}}{4\sqrt{3L\alpha\delta_2 K}}, \; \frac{\sqrt[3]{(f(x^{0,0}) - f(x^*))p}}{2\sqrt[3]{3L^2\alpha\delta_2 K}}, \right.
$$
$$
\left. \frac{\sqrt{(f(x^{0,0}) - f(x^*))Mp}}{5\sqrt{L\alpha\sigma^2 K}}, \; \frac{\sqrt[3]{(f(x^{0,0}) - f(x^*))Mp}}{\sqrt[3]{12L^2\alpha\sigma^2 K}} \right\}.
$$

Thus, we need

$$
\begin{aligned}
\mathcal{O}\left( \frac{\left(f(x^{0,0}) - f(x^*)\right)L\alpha\delta_1}{p\varepsilon^2} \right. & + \frac{\left(f(x^{0,0}) - f(x^*)\right)L\alpha\delta_2}{p\varepsilon^4} + \frac{\left(f(x^{0,0}) - f(x^*)\right)L\alpha\sigma^2}{Mp\varepsilon^4} \\
& \left. + \frac{\left(f(x^{0,0}) - f(x^*)\right)L\sqrt{\alpha\delta_2}}{p^{\frac{3}{2}}\varepsilon^3} + \frac{\left(f(x^{0,0}) - f(x^*)\right)L\sqrt{\alpha}\sigma}{\sqrt{M}p^{\frac{3}{2}}\varepsilon^3} \right)
\end{aligned}
$$

epochs to reach $\varepsilon$-accuracy, where $\varepsilon^2 = \frac{1}{K}\sum_{k=0}^{K-1}\mathbb{E}\left\|\nabla f(x^{k,0})\right\|^2$. Since the last two terms in the estimate in a magnitude smaller, than the second and third accordingly, we can ignore them. The length of the epoch $H \in \text{Geom}(p)$, Algorithm 1 requires $\mathcal{O}\left( \frac{\left(f(x^{0,0}) - f(x^*)\right)L\alpha\delta_1}{p^2\varepsilon^2} + \frac{\left(f(x^{0,0}) - f(x^*)\right)L\alpha\delta_2}{p^2\varepsilon^4} + \frac{\left(f(x^{0,0}) - f(x^*)\right)L\alpha\sigma^2}{Mp^2\varepsilon^4} \right)$ communication rounds. Next we mention that at each communication round we communicate with $C$ devices, thus, number of communications is $\mathcal{O}\left( C\frac{\left(f(x^{0,0}) - f(x^*)\right)L\alpha\delta_1}{p^2\varepsilon^2} + C\frac{\left(f(x^{0,0}) - f(x^*)\right)L\alpha\delta_2}{p^2\varepsilon^4} + C\frac{\left(f(x^{0,0}) - f(x^*)\right)L\alpha\sigma^2}{Mp^2\varepsilon^4} \right)$.

Taking $p = \frac{C}{M}$, we have the result of the corollary. The choice of $p$ is motivated by the fact that we perform $\frac{1}{p}C + M$ communications per-epoch, and established $p$ is the minimal, which delivers $\mathcal{O}(M)$ communications at each epoch. This is also the reason for the additional factor $M$ in the estimate on communications. $\square$

**Corollary D.4.** *Under conditions of Theorem 3.2 Algorithm 1 needs*

$$\mathcal{O}\left(\frac{M}{\min\limits_{k,h} C^{k,h}}\left(\frac{\Delta L\alpha\delta_1}{\varepsilon^2} + \frac{\Delta L\alpha\delta_2}{\varepsilon^4} + \frac{\Delta L\alpha\sigma^2}{M\varepsilon^4}\right)\right) \quad \textit{epochs and}$$

$$\mathcal{O}\left(M\left(\frac{M}{\min\limits_{k,h} C^{k,h}}\right)^2\left(\frac{\Delta L\alpha\delta_1}{\varepsilon^2} + \frac{\Delta L\alpha\delta_2}{\varepsilon^4} + \frac{\Delta L\alpha\sigma^2}{M\varepsilon^4}\right)\right) \quad \textit{number of devices communications}$$

*to reach $\varepsilon$-accuracy, where $\varepsilon^2 = \frac{1}{K}\sum\limits_{k=0}^{K-1}\mathbb{E}\left\|\nabla f(x^{k,0})\right\|^2$, $\Delta = f(x^{0,0}) - f(x^*)$ and $C^{k,h}$ is the number of devices participating in $k$-th iteration in $h$-th epoch.*

*Proof.* Using the result of Theorem 3.2, we choose

$$\gamma \leqslant \min\left\{\frac{p}{384L\alpha(\delta_1+1)}, \frac{\sqrt{(f(x^{0,0}) - f(x^*))\,p}}{4\sqrt{3L\alpha\delta_2 K}}, \frac{\sqrt[3]{(f(x^{0,0}) - f(x^*))p}}{2\sqrt[3]{3L^2\alpha\delta_2 K}},\right.$$

$$\left.\frac{\sqrt{(f(x^{0,0}) - f(x^*))\,Mp}}{5\sqrt{L\alpha\sigma^2 K}}, \frac{\sqrt[3]{(f(x^{0,0}) - f(x^*))\,Mp}}{\sqrt[3]{12L^2\alpha\sigma^2 K}}\right\}.$$

Thus, we need

$$\mathcal{O}\left(\frac{\left(f(x^{0,0}) - f(x^*)\right)L\alpha\delta_1}{p\varepsilon^2} + \frac{\left(f(x^{0,0}) - f(x^*)\right)L\alpha\delta_2}{p\varepsilon^4} + \frac{\left(f(x^{0,0}) - f(x^*)\right)L\alpha\sigma^2}{Mp\varepsilon^4}\right.$$

$$\left. + \frac{\left(f(x^{0,0}) - f(x^*)\right)L\sqrt{\alpha\delta_2}}{p^{\frac{3}{2}}\varepsilon^3} + \frac{\left(f(x^{0,0}) - f(x^*)\right)L\sqrt{\alpha}\sigma}{\sqrt{M}p^{\frac{3}{2}}\varepsilon^3}\right)$$

epochs to reach $\varepsilon$-accuracy, where $\varepsilon^2 = \frac{1}{K}\sum\limits_{k=0}^{K-1}\mathbb{E}\left\|\nabla f(x^{k,0})\right\|^2$. Since the last two terms in the estimate in a magnitude smaller, than the second and third accordingly, we can ignore them. The length of the epoch $H \in \text{Geom}(p)$, Algorithm 1 requires $\mathcal{O}\left(\frac{\left(f(x^{0,0})-f(x^*)\right)L\alpha\delta_1}{p^2\varepsilon^2} + \frac{\left(f(x^{0,0})-f(x^*)\right)L\alpha\delta_2}{p^2\varepsilon^4} + \frac{\left(f(x^{0,0})-f(x^*)\right)L\alpha\sigma^2}{Mp^2\varepsilon^4}\right)$ communication rounds. Next we mention that at each communication round we communicate with $C^{k,h}$ devices, thus, number of communications is $\mathcal{O}\left(\max\limits_{k,h} C^{k,h}\left(\frac{\left(f(x^{0,0})-f(x^*)\right)L\alpha\delta_1}{p^2\varepsilon^2} + \frac{\left(f(x^{0,0})-f(x^*)\right)L\alpha\delta_2}{p^2\varepsilon^4} + \frac{\left(f(x^{0,0})-f(x^*)\right)L\alpha\sigma^2}{Mp^2\varepsilon^4}\right)\right)$. Taking, $p = \frac{\min\limits_{k,h} C^{k,h}}{M}$, we have the result of the corollary. The choice of $p$ is motivated by the fact that we perform $\frac{1}{p}\max\limits_{k,h} C^{k,h} + M$ communications per-epoch, and established $p$ is the minimal, which delivers $\mathcal{O}\left(M\frac{M}{\min\limits_{k,h} C^{k,h}}\right)$ communications at each epoch while guarantee the epoch is executed (if we take $p = \frac{\max\limits_{k,h} C^{k,h}}{M}$, we can meet $p = 1$). This is also the reason for the additional factor $M\frac{M}{\min\limits_{k,h} C^{k,h}}$ in the estimate on communications. $\qquad\square$

*Remark* D.5. Considering fixed rules $\widehat{\mathcal{R}} \equiv \widetilde{\mathcal{R}} \equiv \mathcal{R}$, we have $\mathcal{O}\left(M\frac{M}{C}\left(\frac{\Delta L\delta_1}{\varepsilon^2} + \frac{\Delta L\delta_2}{\varepsilon^4} + \frac{\Delta L\sigma^2}{M\varepsilon^4}\right)\right)$ and $\mathcal{O}\left(M^2\frac{M}{C}\left(\frac{\Delta L\delta_1}{\varepsilon^2} + \frac{\Delta L\delta_2}{\varepsilon^4} + \frac{\Delta L\sigma^2}{M\varepsilon^4}\right)\right)$ number of devices communications with regularizing parameter $\alpha = 1$ and $\alpha = M$ respectively. Considering various rules, best case with regularizing coefficient $\alpha = 1$ gives us $\mathcal{O}\left(M\left(\frac{M}{\min\limits_{k,h} C^{k,h}}\right)^2\left(\frac{\Delta L\delta_1}{\varepsilon^2} + \frac{\Delta L\delta_2}{\varepsilon^4} + \frac{\Delta L\sigma^2}{M\varepsilon^4}\right)\right)$ and worst case $\alpha = M$ gives us $\mathcal{O}\left(M^2\left(\frac{M}{\max\limits_{k,h} C^{k,h}}\right)^2\left(\frac{\Delta L\delta_1}{\varepsilon^2} + \frac{\Delta L\delta_2}{\varepsilon^4} + \frac{\Delta L\sigma^2}{M\varepsilon^4}\right)\right)$ number of devices communications.

### D.2 PROOF FOR STRONGLY-CONVEX CASE

**Theorem D.6** (**Theorem 3.5**). *Suppose Assumptions 2.1, 2.2(b), 2.3, 2.4 hold. Then for Algorithm 1 with $\theta \leqslant \frac{p\gamma\mu}{4}$ and $\gamma \leqslant \frac{p^2}{96L\alpha(\delta_1+1)}$ it implies that*

$$\mathbb{E}\left\|x^{K,0} - x^*\right\|^2 \;\leqslant\; \left(1 - \frac{\gamma\mu}{8}\right)^K \left\|x^{0,0} - x^*\right\|^2 + \frac{8\gamma\alpha}{\mu p^3}\left(144\delta_2 + \frac{74\sigma^2}{M}\right).$$

*Proof.* We start with the definition of virtual sequence:

$$\widetilde{x}^{k,h} = x^{k,h} - \gamma\sum_{m=1}^{M} g_m^{k,h}. \tag{20}$$

It is followed by

$$
\begin{aligned}
\widetilde{x}^{k,h+1} &= x^{k,h+1} - \gamma\sum_{m=1}^{M} g_m^{k,h+1} \\
&= x^{k,h} - \gamma\left[(1-\theta)\sum_{m=1}^{M}\widetilde{\pi}_m^{k,h}\nabla f_m(x^{k,h},\xi_m^{k,h}) + \theta g^{k-1,H^{k-1}}\right] \\
&\quad -\gamma\sum_{m=1}^{M} g_m^{k,h} - \gamma(1-\theta)\sum_{m=1}^{M}\left(\frac{1}{M} - \widetilde{\pi}_m^{k,h}\right)\nabla f_m(x^{k,h},\xi_m^{k,h}) \\
&= \widetilde{x}^{k,h} - \gamma\left[(1-\theta)\frac{1}{M}\sum_{m=1}^{M}\nabla f_m(x^{k,h},\xi_m^{k,h}) + \theta g^{k-1,H^{k-1}}\right]. \tag{21}
\end{aligned}
$$

We use this to write a descent:

$$
\begin{aligned}
\left\|\widetilde{x}^{k,h+1} - x^*\right\|^2 &= \left\|\widetilde{x}^{k,h} - x^*\right\|^2 + 2\left\langle \widetilde{x}^{k,h} - x^*, \widetilde{x}^{k,h+1} - \widetilde{x}^{k,h}\right\rangle + \left\|\widetilde{x}^{k,h+1} - \widetilde{x}^{k,h}\right\|^2 \\
&\stackrel{(21)}{=} \left\|\widetilde{x}^{k,h} - x^*\right\|^2 - 2\gamma\theta\left\langle \widetilde{x}^{k,h} - x^*, g^{k-1,H^{k-1}}\right\rangle \\
&\quad -2\gamma(1-\theta)\left\langle \widetilde{x}^{k,h} - x^*, \frac{1}{M}\sum_{m=1}^{M}\nabla f_m(x^{k,h},\xi_m^{k,h})\right\rangle \\
&\quad +\gamma^2\left\|\theta g^{k-1,H^{k-1}} + (1-\theta)\frac{1}{M}\sum_{m=1}^{M}\nabla f_m(x^{k,h},\xi_m^{k,h})\right\|^2 \\
&\stackrel{(Jen)}{\leqslant} \left\|\widetilde{x}^{k,h} - x^*\right\|^2 - 2\gamma\theta\left\langle \widetilde{x}^{k,h} - x^*, g^{k-1,H^{k-1}}\right\rangle \\
&\quad -2\gamma(1-\theta)\left\langle \widetilde{x}^{k,h} - x^{k,h}, \frac{1}{M}\sum_{m=1}^{M}\nabla f_m(x^{k,h},\xi_m^{k,h})\right\rangle \\
&\quad -2\gamma(1-\theta)\left\langle x^{k,h} - x^*, \frac{1}{M}\sum_{m=1}^{M}\nabla f_m(x^{k,h},\xi_m^{k,h})\right\rangle \\
&\quad +\gamma^2\theta\left\|g^{k-1,H^{k-1}}\right\|^2 + \gamma^2(1-\theta)\left\|\frac{1}{M}\sum_{m=1}^{M}\nabla f_m(x^{k,h},\xi_m^{k,h})\right\|^2.
\end{aligned}
$$

Taking the expectation over $\xi_m^{k,h}$, we have

$$
\begin{aligned}
\mathbb{E}_{\xi_m^{k,h}}\left\|\widetilde{x}^{k,h+1} - x^*\right\|^2 &\leqslant \mathbb{E}_{\xi_m^{k,h}}\left\|\widetilde{x}^{k,h} - x^*\right\|^2 - 2\gamma\theta\mathbb{E}_{\xi_m^{k,h}}\left\langle \widetilde{x}^{k,h} - x^*, g^{k-1,H^{k-1}}\right\rangle \\
&\quad -2\gamma(1-\theta)\mathbb{E}_{\xi_m^{k,h}}\left\langle \widetilde{x}^{k,h} - x^{k,h}, \frac{1}{M}\sum_{m=1}^{M}\nabla f_m(x^{k,h},\xi_m^{k,h})\right\rangle \\
&\quad -2\gamma(1-\theta)\mathbb{E}_{\xi_m^{k,h}}\left\langle x^{k,h} - x^*, \frac{1}{M}\sum_{m=1}^{M}\nabla f_m(x^{k,h},\xi_m^{k,h})\right\rangle
\end{aligned}
$$

$$+\gamma^2 \theta \mathbb{E}_{\xi_m^{k,h}} \left\| g^{k-1,H^{k-1}} \right\|^2$$

$$+\gamma^2 (1-\theta) \mathbb{E}_{\xi_m^{k,h}} \left\| \frac{1}{M} \sum_{m=1}^M \nabla f_m(x^{k,h}, \xi_m^{k,h}) \right\|^2. \tag{22}$$

Mention that

$$\widetilde{x}^{k,h} \overset{(20)}{=} x^{k,h} - \gamma g^{k,h}$$

$$\overset{\text{Line 12}}{=} x^{k,h} - \gamma \left( g^{k,h-1} + (1-\theta) \sum_{m=1}^M \left( \frac{1}{M} - \widetilde{\pi}_m^{k,h-1} \right) \nabla f(x^{k,h-1}, \xi_m^{k,h-1}) \right),$$

Thus, $\widetilde{x}^{k,h}$ and $\xi_m^{k,h}$ are independent. Analogously, $g^{k-1,H^{k-1}}$ and $\xi_m^{k,h}$ are independent. In this way, equation 22 transforms into

$$\begin{aligned}
\mathbb{E}_{\xi_m^{k,h}} \left\| \widetilde{x}^{k,h+1} - x^* \right\|^2 \quad \leqslant \quad & \left\| \widetilde{x}^{k,h} - x^* \right\|^2 - 2\gamma\theta \left\langle \widetilde{x}^{k,h} - x^*, g^{k-1,H^{k-1}} \right\rangle \\
& - 2\gamma(1-\theta) \left\langle \widetilde{x}^{k,h} - x^{k,h}, \nabla f(x^{k,h}) \right\rangle \\
& - 2\gamma(1-\theta) \left\langle x^{k,h} - x^*, \nabla f(x^{k,h}) \right\rangle \\
& + \gamma^2 \theta \left\| g^{k-1,H^{k-1}} \right\|^2 \\
& + \gamma^2 (1-\theta) \mathbb{E}_{\xi_m^{k,h}} \left\| \frac{1}{M} \sum_{m=1}^M \nabla f_m(x^{k,h}, \xi_m^{k,h}) \right\|^2 \\
\overset{(CS)}{\leqslant} \quad & \left\| \widetilde{x}^{k,h} - x^* \right\|^2 - 2\gamma\theta \left\langle \widetilde{x}^{k,h} - x^*, g^{k-1,H^{k-1}} \right\rangle \\
& - 2\gamma(1-\theta) \left\langle \widetilde{x}^{k,h} - x^{k,h}, \nabla f(x^{k,h}) \right\rangle \\
& - 2\gamma(1-\theta) \left\langle x^{k,h} - x^*, \nabla f(x^{k,h}) \right\rangle \\
& + 2\gamma^2 (1-\theta) \mathbb{E}_{\xi_m^{k,h}} \left\| \frac{1}{M} \sum_{m=1}^M \left( \nabla f_m(x^{k,h}, \xi_m^{k,h}) - \nabla f(x^{k,h}) \right) \right\|^2 \\
& + 2\gamma^2 (1-\theta) \left\| \nabla f(x^{k,h}) \right\|^2 + \gamma^2 \theta \left\| g^{k-1,H^{k-1}} \right\|^2. \tag{23}
\end{aligned}$$

Now we pay attention to the following term:

$$\mathbb{E}_{\xi_m^{k,h}} \left\| \frac{1}{M} \sum_{m=1}^M \left( \nabla f_m(x^{k,h}, \xi_m^{k,h}) - \nabla f_m(x^{k,h}) \right) \right\|^2$$

$$\overset{(i)}{=} \frac{1}{M^2} \sum_{m=1}^M \mathbb{E}_{\xi_m^{k,h}} \left\| \nabla f_m(x^{k,h}, \xi_m^{k,h}) - \nabla f_m(x^{k,h}) \right\|^2$$

$$+ \frac{2}{M^2} \sum_{i \neq j} \left\langle \mathbb{E}_{\xi_i^{k,h}} \left[ \nabla f_i(x^{k,h}, \xi_i^{k,h}) - \nabla f_i(x^{k,h}) \right], \mathbb{E}_{\xi_j^{k,h}} \left[ \nabla f_j(x^{k,h}, \xi_j^{k,h}) - \nabla f_j(x^{k,h}) \right] \right\rangle$$

$$\overset{\text{As. 2.4}}{\leqslant} \frac{1}{M} \sigma^2,$$

where $(i)$ is correct, since $\xi_i^{k,h}$ and $\xi_j^{k,h}$ are independent. Substituting this estimate into equation 23, we have

$$\begin{aligned}
\mathbb{E}_{\xi_m^{k,h}} \left\| \widetilde{x}^{k,h+1} - x^* \right\|^2 \quad \leqslant \quad & \left\| \widetilde{x}^{k,h} - x^* \right\|^2 - 2\gamma\theta \left\langle \widetilde{x}^{k,h} - x^*, g^{k-1,H^{k-1}} \right\rangle \\
& - 2\gamma(1-\theta) \left\langle \widetilde{x}^{k,h} - x^{k,h}, \nabla f(x^{k,h}) \right\rangle \\
& - 2\gamma(1-\theta) \left\langle x^{k,h} - x^*, \nabla f(x^{k,h}) \right\rangle
\end{aligned}$$

$$+2\gamma^2(1-\theta)\left\|\nabla f(x^{k,h})\right\|^2 + \gamma^2\theta\left\|g^{k-1,H^{k-1}}\right\|^2$$

$$+\frac{2\gamma^2(1-\theta)\sigma^2}{M}. \tag{24}$$

Let us estimate scalar products separately.

$$-2\gamma\theta\left\langle\widetilde{x}^{k,h}-x^*, g^{k-1,H^{k-1}}\right\rangle \overset{(Fen)}{\leqslant} \theta\left\|\widetilde{x}^{k,h}-x^*\right\|^2 + \gamma^2\theta\left\|g^{k-1,H^{k-1}}\right\|^2,$$

$$-2\gamma(1-\theta)\left\langle\widetilde{x}^{k,h}-x^{k,h}, \nabla f(x^{k,h})\right\rangle \overset{(Fen)}{\leqslant} (1-\theta)\left\|\widetilde{x}^{k,h}-x^{k,h}\right\|^2$$

$$+\gamma^2(1-\theta)\left\|\nabla f(x^{k,h})\right\|^2$$

$$\overset{(20)}{=} \gamma^2(1-\theta)\left\|g^{k,h}\right\|^2 + \gamma^2(1-\theta)\left\|\nabla f(x^{k,h})\right\|^2,$$

$$-2\gamma(1-\theta)\left\langle x^{k,h}-x^*, \nabla f(x^{k,h})\right\rangle \overset{\text{As. 2.2(b)}}{\leqslant} -\gamma\mu(1-\theta)\left\|x^{k,h}-x^*\right\|^2$$

$$-2\gamma(1-\theta)\left[f(x^{k,h})-f(x^*)\right]$$

$$\overset{(CS)}{\leqslant} -\frac{\gamma\mu(1-\theta)}{2}\left\|\widetilde{x}^{k,h}-x^*\right\|^2$$

$$+\gamma\mu(1-\theta)\left\|x^{k,h}-\widetilde{x}^{k,h}\right\|^2$$

$$-2\gamma(1-\theta)\left[f(x^{k,h})-f(x^*)\right]$$

$$\overset{(20)}{=} -\frac{\gamma\mu(1-\theta)}{2}\left\|\widetilde{x}^{k,h}-x^*\right\|^2$$

$$+\gamma^3\mu(1-\theta)\left\|g^{k,h}\right\|^2$$

$$-2\gamma(1-\theta)\left[f(x^{k,h})-f(x^*)\right].$$

Substituting this estimates into equation 24, we obtain

$$\mathbb{E}_{\xi_m^{kh}}\left\|\widetilde{x}^{k,h+1}-x^*\right\|^2 \leqslant \left\|\widetilde{x}^{k,h}-x^*\right\|^2 + \theta\left\|\widetilde{x}^{k,h}-x^*\right\|^2 + 2\gamma^2\theta\left\|g^{k-1,H^{k-1}}\right\|^2$$

$$+\gamma^2(1-\theta)(1+\gamma\mu)\left\|g^{k,h}\right\|^2 + 3\gamma^2(1-\theta)\left\|\nabla f(x^{k,h})\right\|^2$$

$$-\frac{\gamma\mu(1-\theta)}{2}\left\|\widetilde{x}^{k,h}-x^*\right\|^2 - 2\gamma(1-\theta)\left[f(x^{k,h})-f(x^*)\right]$$

$$+\frac{2\gamma^2(1-\theta)\sigma^2}{M}$$

$$= \left(1-\frac{\gamma\mu(1-\theta)}{2}+\theta\right)\left\|\widetilde{x}^{k,h}-x^*\right\|^2 + 2\gamma^2\theta\left\|g^{k-1,H^{k-1}}\right\|^2$$

$$+\gamma^2(1-\theta)(1+\gamma\mu)\left\|g^{k,h}\right\|^2 + 3\gamma^2(1-\theta)\left\|\nabla f(x^{k,h})\right\|^2$$

$$-2\gamma(1-\theta)\left[f(x^{k,h})-f(x^*)\right] + \frac{2\gamma^2(1-\theta)\sigma^2}{M}. \tag{25}$$

Let us choose $\theta \leqslant \frac{\gamma\mu}{4}$ and $\gamma \leqslant \frac{1}{L}$. Then, $\left(1-\frac{\gamma\mu(1-\theta)}{2}+\theta\right) \leqslant \left(1-\frac{3\gamma\mu}{8}+\frac{\gamma\mu}{4}\right) = \left(1-\frac{\gamma\mu}{8}\right)$. In this way, equation 25 transforms to

$$\mathbb{E}_{\xi_m^{kh}}\left\|\widetilde{x}^{k,h+1}-x^*\right\|^2 \leqslant \left(1-\frac{\gamma\mu}{8}\right)\left\|\widetilde{x}^{k,h}-x^*\right\|^2 + 2\gamma^2\theta\left\|g^{k-1,H^{k-1}}\right\|^2$$

$$+2\gamma^2(1-\theta)\left\|g^{k,h}\right\|^2 + 3\gamma^2(1-\theta)\left\|\nabla f(x^{k,h})\right\|^2$$

$$-2\gamma(1-\theta)\left[f(x^{k,h})-f(x^*)\right] + \frac{2\gamma^2(1-\theta)\sigma^2}{M}. \tag{26}$$

Next we estimate

$$3\gamma^2(1-\theta)\left\|\nabla f(x^{k,h})\right\|^2 \overset{(Lip)}{\leqslant} 6\gamma^2 L(1-\theta)\left[f(x^{k,h}) - f(x^*)\right]$$

and combine with equation 25:

$$
\begin{aligned}
\mathbb{E}_{\xi_m^{k,h}}\left\|\widetilde{x}^{k,h+1} - x^*\right\|^2 \leqslant{} & \left(1 - \frac{\gamma\mu}{8}\right)\left\|\widetilde{x}^{k,h} - x^*\right\|^2 + 2\gamma^2\theta\left\|g^{k-1,H^{k-1}}\right\|^2 \\
& + 2\gamma^2(1-\theta)\left\|g^{k,h}\right\|^2 \\
& - 2\gamma(1-\theta)(1 - 3\gamma L)\left[f(x^{k,h}) - f(x^*)\right] \\
& + \frac{2\gamma^2(1-\theta)\sigma^2}{M}.
\end{aligned}
$$

By choosing $\gamma \leqslant \frac{1}{6L}$ we can simplify as

$$
\begin{aligned}
\mathbb{E}_{\xi_m^{k,h}}\left\|\widetilde{x}^{k,h+1} - x^*\right\|^2 \leqslant{} & \left(1 - \frac{\gamma\mu}{8}\right)\left\|\widetilde{x}^{k,h} - x^*\right\|^2 + 2\gamma^2\theta\left\|g^{k-1,H^{k-1}}\right\|^2 \\
& + 2\gamma^2(1-\theta)\left\|g^{k,h}\right\|^2 - \gamma(1-\theta)\left[f(x^{k,h}) - f(x^*)\right] \\
& + \frac{2\gamma^2(1-\theta)\sigma^2}{M}.
\end{aligned}
$$

Now we put $h = H^k - 1$ and take additional expectations to obtain

$$
\begin{aligned}
\mathbb{E}_{\xi_m^{k-1,0}}\ldots\mathbb{E}_{\xi_m^{k,H^k-1}}\left\|\widetilde{x}^{k,H^k} - x^*\right\|^2 \leqslant{} & \left(1 - \frac{\gamma\mu}{8}\right)\mathbb{E}_{\xi_m^{k-1,0}}\ldots\mathbb{E}_{\xi_m^{k,H^k-1}}\left\|\widetilde{x}^{k,H^k-1} - x^*\right\|^2 \\
& + 2\gamma^2\theta\mathbb{E}_{\xi_m^{k-1,0}}\ldots\mathbb{E}_{\xi_m^{k-1,H^{k-1}-1}}\left\|g^{k-1,H^{k-1}}\right\|^2 \\
& + 2\gamma^2(1-\theta)\mathbb{E}_{\xi_m^{k-1,0}}\ldots\mathbb{E}_{\xi_m^{k,H^k-1}}\left\|g^{k,H^k-1}\right\|^2 \\
& - \gamma(1-\theta)\mathbb{E}_{\xi_m^{k-1,0}}\ldots\mathbb{E}_{\xi_m^{k,H^k-1}}\left[f(x^{k,H^k-1}) - f(x^*)\right] \\
& + \frac{2\gamma^2(1-\theta)\sigma^2}{M}.
\end{aligned}
$$

We take expectation with respect to $H^{k-1}$ and $H^k$, and apply Lemma C.1:

$$
\begin{aligned}
& \mathbb{E}_{H^{k-1}}\mathbb{E}_{H^k}\mathbb{E}_{\xi_m^{k-1,0}}\ldots\mathbb{E}_{\xi_m^{k,H^k-1}}\left\|\widetilde{x}^{k,H^k} - x^*\right\|^2 \\
& \leqslant p\left(1 - \frac{\gamma\mu}{8}\right)\mathbb{E}_{H^{k-1}}\mathbb{E}_{\xi_m^{k-1,0}}\ldots\mathbb{E}_{\xi_m^{k-1,H^{k-1}-1}}\left\|\widetilde{x}^{k,0} - x^*\right\|^2 \\
& \quad + (1-p)\left(1 - \frac{\gamma\mu}{8}\right)\mathbb{E}_{H^{k-1}}\mathbb{E}_{H^k}\mathbb{E}_{\xi_m^{k-1,0}}\ldots\mathbb{E}_{\xi_m^{k,H^k-1}}\left\|\widetilde{x}^{k,H^k} - x^*\right\|^2 \\
& \quad + 2\gamma^2\theta\mathbb{E}_{H^{k-1}}\mathbb{E}_{\xi_m^{k-1,0}}\ldots\mathbb{E}_{\xi_m^{k-1,H^{k-1}-1}}\left\|g^{k-1,H^{k-1}}\right\|^2 \\
& \quad + 2\gamma^2(1-\theta)(1-p)\mathbb{E}_{H^{k-1}}\mathbb{E}_{H^k}\mathbb{E}_{\xi_m^{k-1,0}}\ldots\mathbb{E}_{\xi_m^{k,H^k-1}}\left\|g^{k,H^k}\right\|^2 \\
& \quad + 2\gamma^2(1-\theta)p\mathbb{E}_{H^{k-1}}\mathbb{E}_{\xi_m^{k-1,0}}\ldots\mathbb{E}_{\xi_m^{k-1,H^{k-1}-1}}\underbrace{\left\|g^{k,0}\right\|^2}_{=0} \\
& \quad - \gamma(1-p)(1-\theta)\mathbb{E}_{H^k}\mathbb{E}_{\xi_m^{k-1,0}}\ldots\mathbb{E}_{\xi_m^{k,H^k-1}}\left[f(x^{k,H^k}) - f(x^*)\right] \\
& \quad - \gamma p(1-\theta)\mathbb{E}_{H^{k-1}}\mathbb{E}_{\xi_m^{k-1,0}}\ldots\mathbb{E}_{\xi_m^{k-1,H^{k-1}-1}}\left[f(x^{k,0}) - f(x^*)\right] \\
& \quad + \frac{2\gamma^2(1-\theta)\sigma^2}{M}.
\end{aligned}
$$

We rearrange terms and use that $H^k$ and $\left\{\xi_m^{k-1,h}\right\}_{h=0}^{H^{k-1}-1}$ are independent stochastic values:

$$p\mathbb{E}_{H^{k-1}}\mathbb{E}_{H^k}\mathbb{E}_{\xi_m^{k-1,0}}\dots\mathbb{E}_{\xi_m^{k,H^{k-1}}}\left\|\widetilde{x}^{k,H^k}-x^*\right\|^2$$

$$\leqslant p\left(1-\frac{\gamma\mu}{8}\right)\mathbb{E}_{H^{k-1}}\mathbb{E}_{\xi_m^{k-1,0}}\dots\mathbb{E}_{\xi_m^{k-1,H^{k-1}-1}}\left\|\widetilde{x}^{k,0}-x^*\right\|^2$$

$$+2\gamma^2\theta\mathbb{E}_{H^{k-1}}\mathbb{E}_{\xi_m^{k-1,0}}\dots\mathbb{E}_{\xi_m^{k-1,H^{k-1}-1}}\left\|g^{k-1,H^{k-1}}\right\|^2$$

$$+2\gamma^2(1-p)(1-\theta)\mathbb{E}_{H^{k-1}}\mathbb{E}_{\xi_m^{k-1,0}}\dots\mathbb{E}_{\xi_m^{k-1,H^{k-1}-1}}\mathbb{E}_{H^k}\mathbb{E}_{\xi_m^{k,0}}\dots\mathbb{E}_{\xi_m^{k,H^k-1}}\left\|g^{k,H^k}\right\|^2$$

$$-\gamma(1-p)(1-\theta)\mathbb{E}_{H^{k-1}}\mathbb{E}_{H^k}\mathbb{E}_{\xi_m^{k-1,0}}\dots\mathbb{E}_{\xi_m^{k,H^k}}\left[f(x^{k,H^k})-f(x^*)\right]$$

$$-\gamma p(1-\theta)\mathbb{E}_{H^{k-1}}\mathbb{E}_{\xi_m^{k-1,0}}\dots\mathbb{E}_{\xi_m^{k-1,H^{k-1}-1}}\left[f(x^{k,0})-f(x^*)\right]$$

$$+\frac{2\gamma^2(1-\theta)\sigma^2}{M}.\tag{27}$$

We use Lemma 3.1 to estimate $\left\|g^{k,H^k}\right\|^2$ and $\left\|g^{k-1,H^{k-1}}\right\|^2$. We obtain

$$\mathbb{E}_{H^k}\mathbb{E}_{\xi_m^{k,0}}\dots\mathbb{E}_{\xi_m^{k,H^k-1}}\left\|g^{k,H^k}\right\|^2\leqslant\frac{24(1-\theta)^2\alpha(\delta_1+1)}{p^2}\mathbb{E}_{H^k}\left\|\nabla f(x^{k,H^k})\right\|^2+\frac{48(1-\theta)^2\alpha\delta_2}{p^2}$$

$$+\frac{24(1-\theta)^2\alpha\sigma^2}{Mp^2}.\tag{28}$$

As for $\left\|g^{k-1,H^{k-1}}\right\|^2$, we have

$$\mathbb{E}_{H^{k-1}}\mathbb{E}_{\xi_m^{k-1,0}}\dots\mathbb{E}_{\xi_m^{k-1,H^{k-1}-1}}\left\|g^{k-1,H^{k-1}}\right\|^2$$

$$\leqslant\frac{24(1-\theta)^2\alpha(\delta_1+1)}{p^2}\mathbb{E}_{H^{k-1}}\left\|\nabla f(x^{k-1,H^{k-1}})\right\|^2+\frac{48(1-\theta)^2\alpha\delta_2}{p^2}+\frac{24(1-\theta)^2\alpha\sigma^2}{Mp^2}$$

$$\overset{(CS)}{\leqslant}\frac{48(1-\theta)\alpha(\delta_1+1)}{p^2}\mathbb{E}_{H^{k-1}}\left\|\nabla f(x^{k-1,H^{k-1}})-\nabla f(\widetilde{x}^{k-1,H^{k-1}})\right\|^2$$

$$+\frac{48(1-\theta)^2\alpha(\delta_1+1)}{p^2}\mathbb{E}_{H^{k-1}}\left\|\nabla f(\widetilde{x}^{k-1,H^{k-1}})\right\|^2+\frac{48(1-\theta)^2\alpha\delta_2}{p^2}+\frac{24(1-\theta)^2\alpha\sigma^2}{Mp^2}$$

$$\overset{As.\ 2.1}{\leqslant}\frac{48L^2(1-\theta)^2\alpha(\delta_1+1)}{p^2}\mathbb{E}_{H^{k-1}}\left\|x^{k-1,H^{k-1}}-\widetilde{x}^{k-1,H^{k-1}}\right\|^2$$

$$+\frac{48(1-\theta)^2\alpha(\delta_1+1)}{p^2}\mathbb{E}_{H^{k-1}}\left\|\nabla f(x^{k,0})\right\|^2+\frac{48(1-\theta)^2\alpha\delta_2}{p^2}+\frac{24(1-\theta)^2\alpha\sigma^2}{Mp^2}$$

$$\overset{(20)}{=}\frac{48\gamma^2L^2(1-\theta)^2\alpha(\delta_1+1)}{p^2}\mathbb{E}_{H^{k-1}}\left\|g^{k-1,H^{k-1}}\right\|^2$$

$$+\frac{48(1-\theta)^2\alpha(\delta_1+1)}{p^2}\mathbb{E}_{H^{k-1}}\left\|\nabla f(x^{k,0})\right\|^2+\frac{48(1-\theta)^2\alpha\delta_2}{p^2}+\frac{24(1-\theta)^2\alpha\sigma^2}{Mp^2}.$$

We choose $\gamma\leqslant\frac{p}{96L\sqrt{\alpha}\sqrt{\delta_1+1}}$. Moreover, we take additional expectations and again use that $H^{k-1}$ and $\left\{\xi_m^{k-1,h}\right\}_{h=0}^{H^{k-1}-1}$ are independent stochastic values:

$$\mathbb{E}_{H^{k-1}}\mathbb{E}_{\xi_m^{k-1,0}}\dots\mathbb{E}_{\xi_m^{k-1,H^{k-1}-1}}\left\|g^{k-1,H^{k-1}}\right\|^2$$

$$\leqslant\frac{96(1-\theta)^2\alpha(\delta_1+1)}{p^2}\mathbb{E}_{H^{k-1}}\mathbb{E}_{\xi_m^{k-1,0}}\dots\mathbb{E}_{\xi_m^{k-1,H^{k-1}-1}}\left\|\nabla f(x^{k,0})\right\|^2+\frac{96(1-\theta)^2\alpha\delta_2}{p^2}$$

$$+\frac{48(1-\theta)^2\alpha\sigma^2}{Mp^2}.\tag{29}$$

Applying equation Lip to equation 28, equation 29 and substituting it to equation 27, we get

$$p\mathbb{E}_{H^{k-1}}\mathbb{E}_{H^k}\mathbb{E}_{\xi_m^{k-1,0}}\ldots\mathbb{E}_{\xi_m^{k,H^k-1}}\left\|\widetilde{x}^{k,H^k}-x^*\right\|^2$$

$$\leqslant p\left(1-\frac{\gamma\mu}{8}\right)\mathbb{E}_{\xi_m^{k-1,0}}\ldots\mathbb{E}_{\xi_m^{k-1,H^{k-1}-1}}\left\|\widetilde{x}^{k,0}-x^*\right\|^2$$

$$-\gamma(1-p)(1-\theta)\mathbb{E}_{H^{k-1}}\mathbb{E}_{H^k}\mathbb{E}_{\xi_m^{k-1,0}}\ldots\mathbb{E}_{\xi_m^{k,H^k-1}}\left[f(x^{k,H^k})-f(x^*)\right]$$

$$-\gamma p(1-\theta)\mathbb{E}_{H^{k-1}}\mathbb{E}_{\xi_m^{k-1,0}}\ldots\mathbb{E}_{\xi_m^{k-1,H^{k-1}-1}}\left[f(x^{k,0})-f(x^*)\right]$$

$$+\frac{96\gamma^2 L(1-p)(1-\theta)^3\alpha(\delta_1+1)}{p^2}\mathbb{E}_{H^{k-1}}\mathbb{E}_{\xi_m^{k-1,0}}\ldots\mathbb{E}_{\xi_m^{k-1,H-1}}\mathbb{E}_{H^k}\left[f(x^{k,H^k})-f(x^*)\right]$$

$$+\frac{96\gamma^2(1-p)(1-\theta)^3\alpha\delta_2}{p^2}+\frac{48\gamma^2(1-p)(1-\theta)^3\alpha\sigma^2}{Mp^2}$$

$$+\frac{384\gamma^2 L\theta(1-\theta)^2\alpha(\delta_1+1)}{p^2}\mathbb{E}_{H^{k-1}}\mathbb{E}_{\xi_m^{k-1,0}}\ldots\mathbb{E}_{\xi_m^{k-1,H-1}}\left[f(x^{k,0})-f(x^*)\right]$$

$$+\frac{192\gamma^2\theta(1-\theta)^2\alpha\delta_2}{p^2}+\frac{96\gamma^2\theta(1-\theta)^2\alpha\sigma^2}{Mp^2}+\frac{2\gamma^2(1-\theta)\sigma^2}{M}.$$

We take the full expectation, then use a law of expectation and rearrange terms:

$$p\mathbb{E}\left\|\widetilde{x}^{k,H^k}-x^*\right\|^2 \leqslant p\left(1-\frac{\gamma\mu}{8}\right)\mathbb{E}\left\|\widetilde{x}^{k,0}-x^*\right\|^2$$

$$-\gamma(1-p)(1-\theta)\left(1-\frac{96\gamma L(1-\theta)^2\alpha(\delta_1+1)}{p^2}\right)\cdot$$

$$\cdot\mathbb{E}\left[f(x^{k,H^k})-f(x^*)\right]$$

$$-\gamma p(1-\theta)\left(1-\frac{384\gamma L\theta(1-\theta)\alpha(\delta_1+1)}{p^3}\right)\mathbb{E}\left[f(x^{k,0})-f(x^*)\right]$$

$$+\frac{96\gamma^2(1-p)(1-\theta)^3\alpha\delta_2}{p^2}+\frac{192\gamma^2\theta(1-\theta)^2\alpha\delta_2}{p^2}$$

$$+\frac{48\gamma^2(1-p)(1-\theta)^3\alpha\sigma^2}{Mp^2}+\frac{96\gamma^2\theta(1-\theta)^2\alpha\sigma^2}{Mp^2}$$

$$+\frac{2\gamma^2(1-\theta)\sigma^2}{M}.$$

We choose $\theta\leqslant\frac{p\gamma\mu}{4}$ and $\gamma\leqslant\frac{p^2}{96L\alpha(\delta_1+1)}$. Note that all previous transitions hold even with larger choice of $\theta$ and $\gamma$, consequently this choice is correct. In that way, we obtain

$$\mathbb{E}\left\|\widetilde{x}^{k,H^k}-x^*\right\|^2 \leqslant \left(1-\frac{\gamma\mu}{8}\right)\mathbb{E}\left\|\widetilde{x}^{k,0}-x^*\right\|^2+\frac{96\gamma^2\alpha\delta_2}{p^3}+\frac{48\gamma^3\mu\alpha\delta_2}{p^2}$$

$$+\frac{48\gamma^2\alpha\sigma^2}{Mp^3}+\frac{24\gamma^3\mu\alpha\sigma^2}{Mp^2}+\frac{2\gamma^2\sigma^2}{Mp}.$$

Note that $\widetilde{x}^{k,H^k}=x^{k,H^k}-\gamma g^{k,H^k}=x^{k+1,0}$ and $\widetilde{x}^{k,0}=x^{k,0}$. Thus,

$$\mathbb{E}\left\|x^{k+1,0}-x^*\right\|^2\leqslant\left(1-\frac{\gamma\mu}{8}\right)\mathbb{E}\left\|x^{k,0}-x^*\right\|^2+\frac{\gamma^2\alpha}{p^3}\left(144\delta_2+\frac{74\sigma^2}{M}\right).$$

It remains for us to take into account going into recursion over all epochs and claim the result of the theorem:

$$\mathbb{E}\left\|x^{K,0}-x^*\right\|^2 \leqslant \left(1-\frac{\gamma\mu}{8}\right)^K\left\|x^{0,0}-x^*\right\|^2+\frac{\gamma^2\alpha}{p^3}\left(144\delta_2+\frac{74\sigma^2}{M}\right)\sum_{k=0}^{K}\left(1-\frac{\gamma\mu}{8}\right)^k$$

$$\leqslant \quad \left(1 - \frac{\gamma\mu}{8}\right)^K \left\|x^{0,0} - x^*\right\|^2 + \frac{8\gamma\alpha}{\mu p^3}\left(144\delta_2 + \frac{74\sigma^2}{M}\right).$$

$\square$

**Corollary D.7** (**Corollary 3.6**). *Under conditions of Theorem 3.5 Algorithm 1 with fixed rules* $\widehat{\mathcal{R}} \equiv \widetilde{\mathcal{R}} \equiv \mathcal{R}$ *needs*

$$\widetilde{\mathcal{O}}\left(\left(\frac{M}{C}\right)^2\left(\frac{L}{\mu}\alpha\delta_1\log\left(\frac{1}{\varepsilon}\right) + \frac{M}{C}\frac{\alpha\delta_2}{\mu^2\varepsilon} + \frac{\alpha\sigma^2}{\mu^2 C\varepsilon}\right)\right) \quad epochs\ and$$

$$\widetilde{\mathcal{O}}\left(M\left(\frac{M}{C}\right)^2\left(\frac{L}{\mu}\alpha\delta_1\log\left(\frac{1}{\varepsilon}\right) + \frac{M}{C}\frac{\alpha\delta_2}{\mu^2\varepsilon} + \frac{\alpha\sigma^2}{\mu^2 C\varepsilon}\right)\right) \quad number\ of\ devices\ communications$$

*to reach* $\varepsilon$*-accuracy, where* $\varepsilon^2 = \mathbb{E}\left\|x^{K,0} - x^*\right\|^2$ *and* $C$ *is the number devices participating in each epoch.*

*Proof.* Using the result of Theorem 3.5, we choose

$$\gamma \leqslant \min\left\{\frac{p^2}{96L\alpha(\delta_1+1)}, \frac{8\log\left(\max\left\{2, \frac{\mu^2 Mp^3\left\|x^{0,0}-x^*\right\|^2 K}{4736\alpha\sigma^2}, \frac{\mu^2 p^3\left\|x^{0,0}-x^*\right\|^2 K}{9216\alpha\delta_2}\right\}\right)}{\mu K}\right\}$$

Thus, we need $\widetilde{\mathcal{O}}\left(\frac{L\alpha\delta_1}{\mu p^2}\log\left(\frac{1}{\varepsilon}\right) + \frac{\alpha\delta_2}{\mu^2 p^3\varepsilon} + \frac{\alpha\sigma^2}{\mu^2 Mp^3\varepsilon}\right)$ epochs to reach $\varepsilon$-accuracy, where $\varepsilon^2 = \mathbb{E}\left\|x^{K,0} - x^*\right\|^2$. Since the length of the epoch $H \in \mathrm{Geom}(p)$, Algorithm 1 requires $\widetilde{\mathcal{O}}\left(\frac{L\alpha\delta_1}{\mu p^3}\log\left(\frac{1}{\varepsilon}\right) + \frac{\alpha\delta_2}{\mu^2 p^4\varepsilon} + \frac{\alpha\sigma^2}{\mu^2 Mp^4\varepsilon}\right)$ communication rounds. Next we mention that at each communication round we communicate with $C$ devices, thus, number of communications is $\widetilde{\mathcal{O}}\left(C\left(\frac{L\alpha\delta_1}{\mu p^3}\log\left(\frac{1}{\varepsilon}\right) + \frac{\alpha\delta_2}{\mu^2 p^4\varepsilon} + \frac{\alpha\sigma^2}{\mu^2 Mp^4\varepsilon}\right)\right)$. Taking, $p = \frac{C}{M}$, we have the result of the corollary. The choice of $p$ is motivated by the fact that we perform $\frac{1}{p}C + M$ communications per-epoch, and established $p$ is the minimal, which delivers $\mathcal{O}(M)$ communications at each epoch. This is also the reason for the additional factor $M$ in the estimate on communications. $\square$

**Corollary D.8.** *Under conditions of Theorem 3.5 Algorithm 1 needs*

$$\widetilde{\mathcal{O}}\left(\left(\frac{M}{\min\limits_{k,h}C^{k,h}}\right)^2\left(\frac{L}{\mu}\alpha\delta_1\log\left(\frac{1}{\varepsilon}\right) + \frac{M}{\min\limits_{k,h}C^{k,h}}\frac{\alpha\delta_2}{\mu^2\varepsilon} + \frac{\alpha\sigma^2}{\mu^2\min\limits_{k,h}C^{k,h}\varepsilon}\right)\right) \quad epochs\ and$$

$$\widetilde{\mathcal{O}}\left(M\left(\frac{M}{\min\limits_{k,h}C^{k,h}}\right)^3\left(\frac{L}{\mu}\alpha\delta_1\log\left(\frac{1}{\varepsilon}\right) + \frac{M}{\min\limits_{k,h}C^{k,h}}\frac{\alpha\delta_2}{\mu^2\varepsilon} + \frac{\alpha\sigma^2}{\mu^2\min\limits_{k,h}C^{k,h}\varepsilon}\right)\right)$$

*number of devices communications to reach* $\varepsilon$*-accuracy, where* $\varepsilon^2 = \mathbb{E}\left\|x^{K,0} - x^*\right\|^2$ *and* $C^{k,h}$ *is the number of devices participating in* $k$*-th iteration in* $h$*-th epoch.*

*Proof.* Using the result of Theorem 3.5, we choose

$$\gamma \leqslant \min\left\{\frac{p^2}{96L\alpha(\delta_1+1)}, \frac{8\log\left(\max\left\{2, \frac{\mu^2 Mp^3\left\|x^{0,0}-x^*\right\|^2 K}{4736\alpha\sigma^2}, \frac{\mu^2 p^3\left\|x^{0,0}-x^*\right\|^2 K}{9216\alpha\delta_2}\right\}\right)}{\mu K}\right\}$$

Thus, we need $\widetilde{\mathcal{O}}\left(\frac{L\alpha\delta_1}{\mu p^2}\log\left(\frac{1}{\varepsilon}\right) + \frac{\alpha\delta_2}{\mu^2 p^3\varepsilon} + \frac{\alpha\sigma^2}{\mu^2 Mp^3\varepsilon}\right)$ epochs to reach $\varepsilon$-accuracy, where $\varepsilon^2 = \mathbb{E}\left\|x^{K,0} - x^*\right\|^2$. Since the length of the epoch $H \in \mathrm{Geom}(p)$, Algorithm 1 requires $\widetilde{\mathcal{O}}\left(\frac{L\alpha\delta_1}{\mu p^3}\log\left(\frac{1}{\varepsilon}\right) + \frac{\alpha\delta_2}{\mu^2 p^4\varepsilon} + \frac{\alpha\sigma^2}{\mu^2 Mp^4\varepsilon}\right)$ communication rounds. Next we mention that at each communication round we communicate with $C^{k,h}$ devices, thus, number of communications is $\widetilde{\mathcal{O}}\left(\max\limits_{k,h}C^{k,h}\left(\frac{L\alpha\delta_1}{\mu p^3}\log\left(\frac{1}{\varepsilon}\right) + \frac{\alpha\delta_2}{\mu^2 p^4\varepsilon} + \frac{\alpha\sigma^2}{\mu^2 Mp^4\varepsilon}\right)\right)$. Taking, $p = \frac{\min\limits_{k,h}C^{k,h}}{M}$, we have the result of the

corollary. The choice of $p$ is motivated by the fact that we perform $\frac{1}{p} \max_{k,h} C^{k,h} + M$ communications per-epoch, and established $p$ is the minimal, which delivers $\mathcal{O}\left(M \frac{M}{\min_{k,h} C^{k,h}}\right)$ communications at each epoch while guarantee the epoch is executed (if we take $p = \frac{\max_{k,h} C^{k,h}}{M}$, we can meet $p = 1$). This is also the reason for the additional factor $M \frac{M}{\min_{k,h} C^{k,h}}$ in the estimate on communications. $\square$

*Remark* D.9. Considering fixed rules $\widehat{\mathcal{R}} \equiv \widetilde{\mathcal{R}} \equiv \mathcal{R}$,
we have $\widetilde{\mathcal{O}}\left(M\left(\frac{M}{C}\right)^2 \left(\frac{L}{\mu}\delta_1 \log\left(\frac{1}{\varepsilon}\right) + \frac{M}{C}\frac{\delta_2}{\mu^2\varepsilon} + \frac{\sigma^2}{\mu^2 C\varepsilon}\right)\right)$
and $\widetilde{\mathcal{O}}\left(M^2\left(\frac{M}{C}\right)^2 \left(\frac{L}{\mu}\delta_1 \log\left(\frac{1}{\varepsilon}\right) + \frac{M}{C}\frac{\delta_2}{\mu^2\varepsilon} + \frac{\sigma^2}{\mu^2 C\varepsilon}\right)\right)$ number of devices communications with regularizing parameter $\alpha = 1$ and $\alpha = M$ respectively. Considering various rules, best case with regularizing coefficient $\alpha = 1$ gives us $\widetilde{\mathcal{O}}\left(M\left(\frac{M}{\min_{k,h} C^{k,h}}\right)^3 \left(\frac{L}{\mu}\delta_1 \log\left(\frac{1}{\varepsilon}\right) + \frac{M}{\min_{k,h} C^{k,h}}\frac{\delta_2}{\mu^2\varepsilon} + \frac{\sigma^2}{\mu^2 \min_{k,h} C^{k,h}\varepsilon}\right)\right)$ and worst case $\alpha = M$ gives us $\widetilde{\mathcal{O}}\left(M^2\left(\frac{M}{\min_{k,h} C^{k,h}}\right)^3 \left(\frac{L}{\mu}\delta_1 \log\left(\frac{1}{\varepsilon}\right) + \frac{M}{\min_{k,h} C^{k,h}}\frac{\delta_2}{\mu^2\varepsilon} + \frac{\sigma^2}{\mu^2 \min_{k,h} C^{k,h}\varepsilon}\right)\right)$ number of devices communications.

# E   PROOFS FOR ALGORITHM 2

**Lemma E.1.** *Suppose Assumptions 2.3, 2.4 hold. Then for Algorithm 2 it implies that*

$$\mathbb{E}_{H^k}\mathbb{E}_{\xi_m^{k,0}}\mathbb{E}_{\eta_m^{k,0}}\ldots\mathbb{E}_{\xi_m^{k,H^k-1}}\mathbb{E}_{\eta_m^{k,H^k-1}}\left\|g^{k,H^k}\right\|^2 \leqslant \frac{96(1-\theta)^2\alpha(\delta_1+1)}{p^2 \min_{1\leqslant m\leqslant M} q_m}\mathbb{E}_{H^k}\left\|\nabla f(x^{k,H^k})\right\|^2$$

$$+ \frac{192(1-\theta)^2\alpha\delta_2}{p^2 \min_{1\leqslant m\leqslant M} q_m} + \frac{96(1-\theta)^2\alpha\sigma^2}{p^2 \min_{1\leqslant m\leqslant M} q_m}.$$

*Proof.* Let us start with the following estimate:

$$\left\|g^{k,h+1}\right\|^2 = \left\|g^{k,h} + (1-\theta)\sum_{m=1}^{M}\frac{\eta_m^{k,h}}{q_m}\left(\frac{1}{M} - \hat{\pi}_m^{k,h}\right)\nabla f_m(x^{k,h}, \xi_m^{k,h})\right\|^2$$

$$\overset{(Fen)}{\leqslant} (1+c)\left\|g^{k,h}\right\|^2$$

$$+ \left(1 + \frac{1}{c}\right)(1-\theta)^2\left\|\sum_{m=1}^{M}\frac{\eta_m^{k,h}}{q_m}\left(\frac{1}{M} - \hat{\pi}_m^{k,h}\right)\nabla f_m(x^{k,h}, \xi_m^{k,h})\right\|^2, \quad (30)$$

where $c$ is defined below. Let us estimate the last term and obtain

$$\left\|\sum_{m=1}^{M}\frac{\eta_m^{k,h}}{q_m}\left(\frac{1}{M} - \hat{\pi}_m^{k,h}\right)\nabla f_m(x^{k,h}, \xi_m^{k,h})\right\|^2$$

$$\overset{(CS)}{\leqslant} 2\left\|\sum_{m=1}^{M}\left(\frac{\eta_m^{k,h}}{q_m} - 1\right)\left(\frac{1}{M} - \hat{\pi}_m^{k,h}\right)\nabla f_m(x^{k,h}, \xi_m^{k,h})\right\|^2$$

$$+ 2\left\|\sum_{m=1}^{M}\left(\frac{1}{M} - \hat{\pi}_m^{k,h}\right)\nabla f_m(x^{k,h}, \xi_m^{k,h})\right\|^2$$

$$\overset{(CS)}{\leqslant} 2\sum_{m=1}^{M}\left(\frac{\eta_m^{k,h}}{q_m} - 1\right)^2\left(\frac{1}{M} - \hat{\pi}_m^{k,h}\right)^2\sum_{m=1}^{M}\left\|\nabla f_m(x^{k,h}, \xi_m^{k,h})\right\|^2$$

$$+ 2 \left\| \sum_{m=1}^{M} \left( \frac{1}{M} - \hat{\pi}_m^{k,h} \right) \nabla f_m(x^{k,h}, \xi_m^{k,h}) \right\|^2.$$

We pay attention to the first term. Using $\eta_m^{k,h} \sim \mathcal{B}(q_m)$,

$$\mathbb{E}_{\eta_m^{k,h}} \left( \frac{\eta_m^{k,h}}{q_m} - 1 \right)^2 = \frac{\mathbb{E}_{\eta_m^{k,h}} \left( \eta_m^{k,h} - q_m \right)^2}{(q_m)^2} \leqslant \frac{\sigma_\eta^2}{(q_m)^2} = \frac{1 - q_m}{q_m} \leqslant \frac{1}{q_m}.$$

In that way,

$$\mathbb{E}_{\eta_m^{k,h}} \left\| \sum_{m=1}^{M} \frac{\eta_m^{k,h}}{q_m} \left( \frac{1}{M} - \hat{\pi}_m^{k,h} \right) \nabla f_m(x^{k,h}, \xi_m^{k,h}) \right\|^2$$

$$\leqslant \frac{2}{\min_{1 \leqslant m \leqslant M} q_m} \sum_{m=1}^{M} \left( \frac{1}{M} - \hat{\pi}_m^{k,h} \right)^2 \sum_{m=1}^{M} \left\| \nabla f_m(x^{k,h}, \xi_m^{k,h}) \right\|^2$$

$$+ 2 \left\| \sum_{m=1}^{M} \left( \frac{1}{M} - \hat{\pi}_m^{k,h} \right) \nabla f_m(x^{k,h}, \xi_m^{k,h}) \right\|^2. \tag{31}$$

We obtained an estimate for the second term in Lemma D.1 in equation 7:

$$\mathbb{E}_{\xi_m^{k,h}} \left\| \sum_{m=1}^{M} \left( \frac{1}{M} - \hat{\pi}_m^{k,h} \right) \nabla f_m(x^{k,h}) \right\|^2 \leqslant 4\alpha (\delta_1 + 1) \left\| \nabla f(x^{k,h}) \right\|^2 + 4\alpha\delta_2 + \frac{2\alpha\sigma^2}{M}.$$

Moreover, in equation 6 we found out

$$\sum_{m=1}^{M} \left( \frac{1}{M} - \hat{\pi}_m^{k,h} \right)^2 \leqslant \frac{\alpha}{M}.$$

Combining this estimates with equation 31,

$$\mathbb{E}_{\xi_m^{k,h}} \mathbb{E}_{\eta_m^{k,h}} \left\| \sum_{m=1}^{M} \frac{\eta_m^{k,h}}{q_m} \left( \frac{1}{M} - \hat{\pi}_m^{k,h} \right) \nabla f_m(x^{k,h}, \xi_m^{k,h}) \right\|^2$$

$$\leqslant \frac{2\alpha}{\min_{1 \leqslant m \leqslant M} q_m M} \sum_{m=1}^{M} \mathbb{E}_{\xi_m^{k,h}} \left\| \nabla f_m(x^{k,h}, \xi_m^{k,h}) \right\|^2$$

$$+ 8\alpha(\delta_1 + 1) \left\| \nabla f(x^{k,h}) \right\|^2 + 8\alpha\delta_2 + \frac{4\alpha\sigma^2}{M}$$

$$\stackrel{(CS)}{\leqslant} \frac{4\alpha}{\min_{1 \leqslant m \leqslant M} q_m M} \sum_{m=1}^{M} \mathbb{E}_{\xi_m^{k,h}} \left\| \nabla f_m(x^{k,h}) \right\|^2$$

$$+ \frac{4\alpha}{\min_{1 \leqslant m \leqslant M} q_m M} \sum_{m=1}^{M} \mathbb{E}_{\xi_m^{k,h}} \left\| \nabla f_m(x^{k,h}, \xi_m^{k,h}) - \nabla f_m(x^{k,h}) \right\|^2$$

$$+ 8\alpha(\delta_1 + 1) \left\| \nabla f(x^{k,h}) \right\|^2 + 8\alpha\delta_2 + \frac{4\alpha\sigma^2}{M}$$

$$\stackrel{\text{As. 2.4}}{\leqslant} \frac{4\alpha}{\min_{1 \leqslant m \leqslant M} q_m M} \sum_{m=1}^{M} \left\| \nabla f_m(x^{k,h}) \right\|^2 + \frac{4\alpha\sigma^2}{\min_{1 \leqslant m \leqslant M} q_m}$$

$$+ 8\alpha(\delta_1 + 1) \left\| \nabla f(x^{k,h}) \right\|^2 + 8\alpha\delta_2 + \frac{4\alpha\sigma^2}{M}$$

$$
\overset{(CS)}{\leqslant} \frac{8\alpha}{\min\limits_{1\leqslant m\leqslant M} q_m M} \sum_{m=1}^{M} \left\| \nabla f(x^{k,h}) \right\|^2 + \frac{8\alpha}{\min\limits_{1\leqslant m\leqslant M} q_m M} \sum_{m=1}^{M} \left\| \nabla f_m(x^{k,h}) - \nabla f(x^{k,h}) \right\|^2
$$

$$
+ 8\alpha(\delta_1 + 1) \left\| \nabla f(x^{k,h}) \right\|^2 + 8\alpha\delta_2 + \frac{8\alpha\sigma^2}{\min\limits_{1\leqslant m\leqslant M} q_m}
$$

$$
\overset{\text{As. 2.3}}{\leqslant} \frac{8\alpha(\delta_1 + 1)}{\min\limits_{1\leqslant m\leqslant M} q_m} \left\| \nabla f(x^{k,h}) \right\|^2 + 8\alpha(\delta_1 + 1) \left\| \nabla f(x^{k,h}) \right\|^2 + \frac{8\alpha\delta_2}{\min\limits_{1\leqslant m\leqslant M} q_m}
$$

$$
+ 8\alpha\delta_2 + \frac{8\alpha\sigma^2}{\min\limits_{1\leqslant m\leqslant M} q_m}
$$

$$
\leqslant \frac{16\alpha(\delta_1 + 1)}{\min\limits_{1\leqslant m\leqslant M} q_m} \left\| \nabla f(x^{k,h}) \right\|^2 + \frac{16\alpha\delta_2}{\min\limits_{1\leqslant m\leqslant M} q_m} + \frac{8\alpha\sigma^2}{\min\limits_{1\leqslant m\leqslant M} q_m}.
$$

Substituting this estimate into equation 30, we have

$$
\mathbb{E}_{\xi_m^{k,h}} \mathbb{E}_{\eta_m^{k,h}} \left\| g^{k,h+1} \right\|^2 \leqslant (1+c) \left\| g^{k,h} \right\|^2 + 16 \left(1 + \frac{1}{c}\right)(1-\theta)^2 \frac{\alpha(\delta_1 + 1)}{\min\limits_{1\leqslant m\leqslant M} q_m} \left\| \nabla f(x^{k,h}) \right\|^2
$$

$$
+ 16 \left(1 + \frac{1}{c}\right)(1-\theta)^2 \frac{\alpha\delta_2}{\min\limits_{1\leqslant m\leqslant M} q_m}
$$

$$
+ 8 \left(1 + \frac{1}{c}\right)(1-\theta)^2 \frac{\alpha\sigma^2}{\min\limits_{1\leqslant m\leqslant M} q_m}.
$$

Enrolling a recursion, we get

$$
\mathbb{E}_{\xi_m^{k,0}} \mathbb{E}_{\eta_m^{k,0}} \dots \mathbb{E}_{\xi_m^{k,h}} \mathbb{E}_{\eta_m^{k,h}} \left\| g^{k,h+1} \right\|^2
$$

$$
\leqslant 16 \left(1 + \frac{1}{c}\right)(1-\theta)^2 \frac{\alpha(\delta_1 + 1)}{\min\limits_{1\leqslant m\leqslant M} q_m} \sum_{i=0}^{h}(1+c)^{h-i} \left\| \nabla f(x^{k,i}) \right\|^2
$$

$$
+ 16 \left(1 + \frac{1}{c}\right)(1-\theta)^2 \frac{\alpha\delta_2}{\min\limits_{1\leqslant m\leqslant M} q_m} \sum_{i=0}^{h}(1+c)^{h-i}
$$

$$
+ 8 \left(1 + \frac{1}{c}\right)(1-\theta)^2 \frac{\alpha\sigma^2}{\min\limits_{1\leqslant m\leqslant M} q_m} \sum_{i=0}^{h}(1+c)^{h-i}. \tag{32}
$$

Next, choosing $c = \frac{p}{2}$, taking exception on $H^k$ and applying equation 9, equation 10, equation 11 from Lemma D.1, we obtain the result of the lemma:

$$
\mathbb{E}_{H^k} \mathbb{E}_{\xi_m^{k,0}} \mathbb{E}_{\eta_m^{k,0}} \dots \mathbb{E}_{\xi_m^{k,H^k-1}} \mathbb{E}_{\eta_m^{k,H^k-1}} \left\| g^{k,H^k} \right\|^2 \leqslant \frac{96(1-\theta)^2\alpha(\delta_1 + 1)}{p^2 \min\limits_{1\leqslant m\leqslant M} q_m} \mathbb{E}_{H^k} \left\| \nabla f(x^{k,H^k}) \right\|^2
$$

$$
+ \frac{192(1-\theta)^2\alpha\delta_2}{p^2 \min\limits_{1\leqslant m\leqslant M} q_m} + \frac{96(1-\theta)^2\alpha\sigma^2}{p^2 \min\limits_{1\leqslant m\leqslant M} q_m}.
$$

$\square$

### E.1 PROOF FOR NON-CONVEX SETTING

**Theorem E.2.** *Suppose Assumptions 2.1, 2.2(a), 2.3, 2.4 hold. Then for Algorithm 2 with $\theta \leqslant \frac{\gamma L p^2}{2}$ and $\gamma \leqslant \frac{p \min_{1 \leqslant m \leqslant M} q_m}{768 L \alpha (\delta_1 + 1)}$ it implies that*

$$
\frac{1}{K} \sum_{k=0}^{K-1} \mathbb{E} \left\| \nabla f(x^{k,0}) \right\|^2 \quad \leqslant \quad \frac{16 \left( f(x^{0,0}) - f(x^*) \right)}{\gamma K}
$$

$$
+ \frac{1536 \gamma^2 L^2 \alpha \delta_2}{p^3 \min_{1 \leqslant m \leqslant M} q_m} + \frac{3200 \gamma L \alpha \delta_2}{p \min_{1 \leqslant m \leqslant M} q_m}
$$

$$
+ \frac{768 \gamma^2 L^2 \alpha \sigma^2}{p^3 \min_{1 \leqslant m \leqslant M} q_m} + \frac{1600 \gamma L \alpha \sigma^2}{p \min_{1 \leqslant m \leqslant M} q_m}.
$$

*Proof.* We start with the definition of virtual sequence:

$$
\widetilde{x}^{k,h} = x^{k,h} - \gamma \sum_{m=1}^{M} g_m^{k,h} = x^{k,h} - \gamma g^{k,h}. \tag{33}
$$

It is followed by

$$
\begin{aligned}
\widetilde{x}^{k,h+1} &= x^{k,h+1} - \gamma \sum_{m=1}^{M} g_m^{k,h+1} \\
&= x^{k,h} - \gamma \left[ (1-\theta) \sum_{m=1}^{M} \frac{\eta_m^{k,h}}{q_m} \hat{\pi}_m^{k,h} \nabla f_m(x^{k,h}, \xi_m^{k,h}) + \theta g^{k-1,H^{k-1}} \right] \\
&\quad - \gamma \sum_{m=1}^{M} g_m^{k,h} - \gamma (1-\theta) \sum_{m=1}^{M} \frac{\eta_m^{k,h}}{q_m} \left( \frac{1}{M} - \hat{\pi}_m^{k,h} \right) \nabla f_m(x^{k,h}, \xi_m^{k,h}) \\
&= \widetilde{x}^{k,h} - \gamma \left[ (1-\theta) \frac{1}{M} \sum_{m=1}^{M} \frac{\eta_m^{k,h}}{q_m} \nabla f_m(x^{k,h}, \xi_m^{k,h}) + \theta g^{k-1,H^{k-1}} \right]. \tag{34}
\end{aligned}
$$

Assumption 2.1 implies

$$
\begin{aligned}
f(\widetilde{x}^{k,h+1}) &\leqslant f(\widetilde{x}^{k,h}) + \left\langle \nabla f(\widetilde{x}^{k,h}), \widetilde{x}^{k,h+1} - \widetilde{x}^{k,h} \right\rangle + \frac{L}{2} \left\| \widetilde{x}^{k,h+1} - \widetilde{x}^{k,h} \right\|^2 \\
&\overset{\substack{(34) \\ (Jen)}}{\leqslant} f(\widetilde{x}^{k,h}) - \gamma \theta \left\langle \nabla f(\widetilde{x}^{k,h}), g^{k-1,H^{k-1}} \right\rangle \\
&\quad - \gamma (1-\theta) \left\langle \nabla f(\widetilde{x}^{k,h}), \frac{1}{M} \sum_{m=1}^{M} \frac{\eta_m^{k,h}}{q_m} \nabla f_m(x^{k,h}, \xi_m^{k,h}) \right\rangle \\
&\quad + \frac{\gamma^2 L (1-\theta)}{2} \left\| \frac{1}{M} \sum_{m=1}^{M} \frac{\eta_m^{k,h}}{q_m} \nabla f_m(x^{k,h}, \xi_m^{k,h}) \right\|^2 + \frac{\gamma^2 L \theta}{2} \left\| g^{k-1,H^{k-1}} \right\|^2.
\end{aligned}
$$

Now we use that $\eta_m^{k,h} \sim \mathcal{B}(q_m)$. Consequently, $\mathbb{E} \eta_m^{k,h} = q_m$. Since $\eta_m^{k,h}$ is independent of $x^{k,h}, \widetilde{x}^{k,h}, \xi_m^{k,h}, g^{k-1,H^{k-1}}$, we take the expectation and obtain

$$
\begin{aligned}
\mathbb{E}_{\eta_m^{k,h}} \left[ f(\widetilde{x}^{k,h+1}) \right] &\leqslant f(\widetilde{x}^{k,h}) - \gamma \theta \left\langle \nabla f(\widetilde{x}^{k,h}), g^{k-1,H^{k-1}} \right\rangle \\
&\quad - \gamma (1-\theta) \left\langle \nabla f(\widetilde{x}^{k,h}), \frac{1}{M} \sum_{m=1}^{M} \nabla f_m(x^{k,h}, \xi_m^{k,h}) \right\rangle \\
&\quad + \frac{\gamma^2 L (1-\theta)}{2} \mathbb{E}_{\eta_m^{k,h}} \left\| \frac{1}{M} \sum_{m=1}^{M} \frac{\eta_m^{k,h}}{q_m} \nabla f_m(x^{k,h}, \xi_m^{k,h}) \right\|^2 \\
&\quad + \frac{\gamma^2 L \theta}{2} \left\| g^{k-1,H^{k-1}} \right\|^2.
\end{aligned}
$$

We take the expectation over $\xi_m^{k,h}$. Mention that

$$\widetilde{x}^{k,h} \quad \overset{(33)}{=} \quad x^{k,h} - \gamma g^{k,h}$$

$$\overset{\text{Line 12}}{=} \quad x^{k,h} - \gamma \left( g^{k,h-1} + (1-\theta) \sum_{m=1}^{M} \frac{\eta_m^{k,h-1}}{q_m} \left( \frac{1}{M} - \hat{\pi}_m^{k,h-1} \right) \nabla f(x^{k,h-1}, \xi_m^{k,h-1}) \right).$$

Thus, $\widetilde{x}^{k,h}$ and $\xi_m^{k,h}$ are independent. Analogously, $g^{k-1,H^{k-1}}$ and $\xi_m^{k,h}$ are independent. In this way,

$$
\begin{aligned}
\mathbb{E}_{\xi_m^{k,h}} \mathbb{E}_{\eta_m^{k,h}} \left[ f(\widetilde{x}^{k,h+1}) \right] \quad &\leqslant \quad f(\widetilde{x}^{k,h}) - \gamma\theta \left\langle \nabla f(\widetilde{x}^{k,h}), g^{k-1,H^{k-1}} \right\rangle \\
&\quad - \gamma(1-\theta) \left\langle \nabla f(\widetilde{x}^{k,h}), \nabla f(x^{k,h}) \right\rangle \\
&\quad + \frac{\gamma^2 L(1-\theta)}{2} \mathbb{E}_{\xi_m^{k,h}} \mathbb{E}_{\eta_m^{k,h}} \left\| \frac{1}{M} \sum_{m=1}^{M} \frac{\eta_m^{k,h}}{q_m} \nabla f_m(x^{k,h}, \xi_m^{k,h}) \right\|^2 \\
&\quad + \frac{\gamma^2 L\theta}{2} \left\| g^{k-1,H^{k-1}} \right\|^2 .
\end{aligned}
\tag{35}
$$

Let us consider separately the following term:

$$
\begin{aligned}
\mathbb{E}_{\xi_m^{k,h}} \mathbb{E}_{\eta_m^{k,h}} \left\| \frac{1}{M} \sum_{m=1}^{M} \frac{\eta_m^{k,h}}{q_m} \nabla f_m(x^{k,h}, \xi_m^{k,h}) \right\|^2 & \\
\overset{(CS)}{\leqslant} 2\mathbb{E}_{\xi_m^{k,h}} \mathbb{E}_{\eta_m^{k,h}} & \left\| \frac{1}{M} \sum_{m=1}^{M} \frac{\eta_m^{k,h}}{q_m} \nabla f_m(x^{k,h}, \xi_m^{k,h}) - \frac{1}{M} \sum_{m=1}^{M} \nabla f_m(x^{k,h}, \xi_m^{k,h}) \right\|^2 \\
+ 2\mathbb{E}_{\xi_m^{k,h}} \mathbb{E}_{\eta_m^{k,h}} & \left\| \frac{1}{M} \sum_{m=1}^{M} \nabla f_m(x^{k,h}, \xi_m^{k,h}) \right\|^2 \\
= 2\mathbb{E}_{\xi_m^{k,h}} \mathbb{E}_{\eta_m^{k,h}} & \left\| \frac{1}{M} \sum_{m=1}^{M} \left( \frac{\eta_m^{k,h}}{q_m} - 1 \right) \nabla f_m(x^{k,h}, \xi_m^{k,h}) \right\|^2 \\
+ 2\mathbb{E}_{\xi_m^{k,h}} & \left\| \frac{1}{M} \sum_{m=1}^{M} \nabla f_m(x^{k,h}, \xi_m^{k,h}) \right\|^2 \\
\overset{(CS)}{\leqslant} \frac{2}{M^2} E_{\xi_m^{k,h}} \mathbb{E}_{\eta_m^{k,h}} & \sum_{m=1}^{M} \left( \frac{\eta_m^{k,h}}{q_m} - 1 \right)^2 \sum_{m=1}^{M} \left\| \nabla f_m(x^{k,h}, \xi_m^{k,h}) \right\|^2 \\
+ 2\mathbb{E}_{\xi_m^{k,h}} & \left\| \frac{1}{M} \sum_{m=1}^{M} \nabla f_m(x^{k,h}, \xi_m^{k,h}) \right\|^2 .
\end{aligned}
\tag{36}
$$

We pay attention to the first term. Using $\eta_m^{k,h} \sim \mathcal{B}(q_m)$,

$$
\begin{aligned}
\sum_{m=1}^{M} \mathbb{E}_{\eta_m^{k,h}} \left( \frac{\eta_m^{k,h}}{q_m} - 1 \right)^2 &= \sum_{m=1}^{M} \frac{\mathbb{E}_{\eta_m^{k,h}} \left( \eta_m^{k,h} - q_m \right)^2}{(q_m)^2} \leqslant \sum_{m=1}^{M} \frac{\sigma_\eta^2}{(q_m)^2} \\
&= \sum_{m=1}^{M} \frac{1 - q_m}{q_m} \leqslant \frac{M}{\min\limits_{1 \leqslant m \leqslant M} q_m} .
\end{aligned}
$$

Combining with equation 36,

$$\mathbb{E}_{\xi_m^{k,h}} \mathbb{E}_{\eta_m^{k,h}} \left\| \frac{1}{M} \sum_{m=1}^{M} \frac{\eta_m^{k,h}}{q_m} \nabla f_m(x^{k,h}, \xi_m^{k,h}) \right\|^2$$

$$\leqslant \frac{2}{M \min_{1 \leqslant m \leqslant M} q_m} \sum_{m=1}^{M} \mathbb{E}_{\xi_m^{k,h}} \left\| \nabla f_m(x^{k,h}, \xi_m^{k,h}) \right\|^2 + 2\mathbb{E}_{\xi_m^{k,h}} \left\| \frac{1}{M} \sum_{m=1}^{M} \nabla f_m(x^{k,h}, \xi_m^{k,h}) \right\|^2$$

$$\overset{(CS)}{\leqslant} \frac{4}{M \min_{1 \leqslant m \leqslant M} q_m} \sum_{m=1}^{M} \mathbb{E}_{\xi_m^{k,h}} \left\| \nabla f_m(x^{k,h}, \xi_m^{k,h}) \right\|^2$$

$$\overset{(CS)}{\leqslant} \frac{8}{M \min_{1 \leqslant m \leqslant M} q_m} \sum_{m=1}^{M} \left\| \nabla f_m(x^{k,h}) \right\|^2$$

$$+ \frac{8}{M \min_{1 \leqslant m \leqslant M} q_m} \sum_{m=1}^{M} \mathbb{E}_{\xi_m^{k,h}} \left\| \nabla f_m(x^{k,h}, \xi_m^{k,h}) - \nabla f_m(x^{k,h}) \right\|^2$$

$$\overset{\text{As. 2.4}}{\leqslant} \frac{8}{M \min_{1 \leqslant m \leqslant M} q_m} \sum_{m=1}^{M} \left\| \nabla f_m(x^{k,h}) \right\|^2 + \frac{8\sigma^2}{\min_{1 \leqslant m \leqslant M} q_m}$$

$$\overset{(CS)}{\leqslant} \frac{16}{M \min_{1 \leqslant m \leqslant M} q_m} \sum_{m=1}^{M} \left\| \nabla f(x^{k,h}) \right\|^2$$

$$+ \frac{16}{M \min_{1 \leqslant m \leqslant M} q_m} \sum_{m=1}^{M} \left\| \nabla f_m(x^{k,h}) - \nabla f(x^{k,h}) \right\|^2 + \frac{8\sigma^2}{\min_{1 \leqslant m \leqslant M} q_m}$$

$$\overset{\text{As. 2.3}}{\leqslant} \frac{16(\delta_1 + 1)}{\min_{1 \leqslant m \leqslant M} q_m} \left\| \nabla f(x^{k,h}) \right\|^2 + \frac{16\delta_2}{\min_{1 \leqslant m \leqslant M} q_m} + \frac{8\sigma^2}{\min_{1 \leqslant m \leqslant M} q_m}. \tag{37}$$

We substitute this estimate into equation 35 to obtain

$$\begin{aligned}
\mathbb{E}_{\xi_m^{k,h}} \mathbb{E}_{\eta_m^{k,h}} \left[ f(\widetilde{x}^{k,h+1}) \right] \quad \leqslant \quad & f(\widetilde{x}^{k,h}) - \gamma\theta \left\langle \nabla f(\widetilde{x}^{k,h}), g^{k-1,H^{k-1}} \right\rangle \\
& - \gamma(1-\theta) \left\langle \nabla f(\widetilde{x}^{k,h}), \nabla f(x^{k,h}) \right\rangle \\
& + \frac{8\gamma^2 L(1-\theta)(\delta_1 + 1)}{\min_{1 \leqslant m \leqslant M} q_m} \left\| \nabla f(x^{k,h}) \right\|^2 \\
& + \frac{\gamma^2 L\theta}{2} \left\| g^{k-1,H^{k-1}} \right\|^2 + \frac{8\gamma^2 L(1-\theta)\delta_2}{\min_{1 \leqslant m \leqslant M} q_m} \\
& + \frac{4\gamma^2 L(1-\theta)\sigma^2}{\min_{1 \leqslant m \leqslant M} q_m}. \tag{38}
\end{aligned}$$

Let us estimate the scalar products separately.

$$\begin{aligned}
-\gamma(1-\theta) \left\langle \nabla f(\widetilde{x}^{k,h}), \nabla f(x^{k,h}) \right\rangle \quad = \quad & -\frac{\gamma(1-\theta)}{2} \left\| \nabla f(\widetilde{x}^{k,h}) \right\|^2 - \frac{\gamma(1-\theta)}{2} \left\| \nabla f(x^{k,h}) \right\|^2 \\
& + \frac{\gamma(1-\theta)}{2} \left\| \nabla f(\widetilde{x}^{k,h}) - \nabla f(x^{k,h}) \right\|^2 \\
\overset{\text{As. 2.1}}{\leqslant} \quad & -\frac{\gamma(1-\theta)}{2} \left\| \nabla f(\widetilde{x}^{k,h}) \right\|^2 - \frac{\gamma(1-\theta)}{2} \left\| \nabla f(x^{k,h}) \right\|^2 \\
& + \frac{\gamma L^2(1-\theta)}{2} \left\| \widetilde{x}^{k,h} - x^{k,h} \right\|^2 \\
\overset{(12)}{=} \quad & -\frac{\gamma(1-\theta)}{2} \left\| \nabla f(\widetilde{x}^{k,h}) \right\|^2 - \frac{\gamma(1-\theta)}{2} \left\| \nabla f(x^{k,h}) \right\|^2 \\
& + \frac{\gamma^3 L^2(1-\theta)}{2} \left\| g^{k,h} \right\|^2, \\
-\gamma\theta \left\langle \nabla f(\widetilde{x}^{k,h}), g^{k-1,H^{k-1}} \right\rangle \quad \overset{(Fen)}{\leqslant} \quad & \frac{\gamma\theta}{2} \left\| \nabla f(\widetilde{x}^{k,h}) \right\|^2 + \frac{\gamma\theta}{2} \left\| g^{k-1,H^{k-1}} \right\|^2.
\end{aligned}$$

Combining it with equation 38,

$$
\begin{aligned}
\mathbb{E}_{\xi_m^{k,h}} \mathbb{E}_{\eta_m^{k,h}} \left[ f(\widetilde{x}^{k,h+1}) \right] \quad \leqslant \quad & f(\widetilde{x}^{k,h}) - \frac{\gamma(1-\theta)}{2} \left( 1 - \frac{16\gamma L(\delta_1 + 1)}{\min\limits_{1 \leqslant m \leqslant M} q_m} \right) \left\| \nabla f(x^{k,h}) \right\|^2 \\
& - \frac{\gamma(1-2\theta)}{2} \left\| \nabla f(\widetilde{x}^{k,h}) \right\|^2 + \frac{\gamma^3 L^2 (1-\theta)}{2} \left\| g^{k,h} \right\|^2 \\
& + \frac{\gamma\theta(\gamma L + 1)}{2} \left\| g^{k-1,H^{k-1}} \right\|^2 + \frac{8\gamma^2 L(1-\theta)\delta_2}{\min\limits_{1 \leqslant m \leqslant M} q_m} + \frac{4\gamma^2 L(1-\theta)\sigma^2}{\min\limits_{1 \leqslant m \leqslant M} q_m}.
\end{aligned}
$$

Choosing $\gamma \leqslant \frac{\min_{1 \leqslant m \leqslant M} q_m}{32 L(\delta_1 + 1)}$ and $\theta \leqslant \frac{1}{2}$,

$$
\begin{aligned}
\mathbb{E}_{\xi_m^{k,h}} \mathbb{E}_{\eta_m^{k,h}} \left[ f(\widetilde{x}^{k,h+1}) \right] \quad \leqslant \quad & f(\widetilde{x}^{k,h}) - \frac{\gamma(1-\theta)}{4} \left\| \nabla f(x^{k,h}) \right\|^2 + \frac{\gamma^3 L^2 (1-\theta)}{2} \left\| g^{k,h} \right\|^2 \\
& + \gamma\theta \left\| g^{k-1,H^{k-1}} \right\|^2 + \frac{8\gamma^2 L(1-\theta)\delta_2}{\min\limits_{1 \leqslant m \leqslant M} q_m} + \frac{4\gamma^2 L(1-\theta)\sigma^2}{\min\limits_{1 \leqslant m \leqslant M} q_m}.
\end{aligned}
$$

Now we put $h = H^k - 1$ and take additional expectations.

$$
\begin{aligned}
& \mathbb{E}_{\xi_m^{k-1,0}} \mathbb{E}_{\eta_m^{k-1,0}} \ldots \mathbb{E}_{\xi_m^{k,H^k-1}} \mathbb{E}_{\eta_m^{k,H^k-1}} \left[ f(\widetilde{x}^{k,H^k}) \right] \\
& \leqslant \mathbb{E}_{\xi_m^{k-1,0}} \mathbb{E}_{\eta_m^{k-1,0}} \ldots \mathbb{E}_{\xi_m^{k,H^k-1}} \mathbb{E}_{\eta_m^{k,H^k-1}} \left[ f(\widetilde{x}^{k,H^k-1}) \right] \\
& \quad - \frac{\gamma(1-\theta)}{4} \mathbb{E}_{\xi_m^{k-1,0}} \mathbb{E}_{\eta_m^{k-1,0}} \ldots \mathbb{E}_{\xi_m^{k,H^k-1}} \mathbb{E}_{\eta_m^{k,H^k-1}} \left\| \nabla f(x^{k,H^k-1}) \right\|^2 \\
& \quad + \frac{\gamma^3 L^2 (1-\theta)}{2} \mathbb{E}_{\xi_m^{k-1,0}} \mathbb{E}_{\eta_m^{k-1,0}} \ldots \mathbb{E}_{\xi_m^{k,H^k-1}} \mathbb{E}_{\eta_m^{k,H^k-1}} \left\| g^{k,H^k-1} \right\|^2 \\
& \quad + \gamma\theta \mathbb{E}_{\xi_m^{k-1,0}} \mathbb{E}_{\eta_m^{k-1,0}} \ldots \mathbb{E}_{\xi_m^{k-1,H^{k-1}-1}} \mathbb{E}_{\eta_m^{k-1,H^{k-1}-1}} \left\| g^{k-1,H^{k-1}} \right\|^2 \\
& \quad + \frac{8\gamma^2 L(1-\theta)\delta_2}{\min\limits_{1 \leqslant m \leqslant M} q_m} + \frac{4\gamma^2 L(1-\theta)\sigma^2}{\min\limits_{1 \leqslant m \leqslant M} q_m}.
\end{aligned}
$$

We take expectation with respect to $H^{k-1}$ and $H^k$, and apply Lemma C.1:

$$
\begin{aligned}
& \mathbb{E}_{H^{k-1}} \mathbb{E}_{H^k} \mathbb{E}_{\xi_m^{k-1,0}} \mathbb{E}_{\eta_m^{k-1,0}} \ldots \mathbb{E}_{\xi_m^{k,H^k-1}} \mathbb{E}_{\eta_m^{k,H^k-1}} \left[ f(\widetilde{x}^{k,H^k}) \right] \\
& \leqslant (1-p) \mathbb{E}_{H^{k-1}} \mathbb{E}_{H^k} \mathbb{E}_{\xi_m^{k-1,0}} \mathbb{E}_{\eta_m^{k-1,0}} \ldots \mathbb{E}_{\xi_m^{k,H^k-1}} \mathbb{E}_{\eta_m^{k,H^k-1}} \left[ f(\widetilde{x}^{k,H^k}) \right] \\
& \quad + p \mathbb{E}_{H^{k-1}} \mathbb{E}_{\xi_m^{k-1,0}} \mathbb{E}_{\eta_m^{k-1,0}} \ldots \mathbb{E}_{\xi_m^{k-1,H^{k-1}-1}} \mathbb{E}_{\eta_m^{k-1,H^{k-1}-1}} \left[ f(\widetilde{x}^{k,0}) \right] \\
& \quad - \frac{\gamma(1-\theta)p}{4} \mathbb{E}_{H^{k-1}} \mathbb{E}_{\xi_m^{k-1,0}} \mathbb{E}_{\eta_m^{k-1,0}} \ldots \mathbb{E}_{\xi_m^{k-1,H^{k-1}-1}} \mathbb{E}_{\eta_m^{k-1,H^{k-1}-1}} \left\| \nabla f(x^{k,0}) \right\|^2 \\
& \quad - \frac{\gamma(1-\theta)(1-p)}{4} \mathbb{E}_{H^{k-1}} \mathbb{E}_{H^k} \mathbb{E}_{\xi_m^{k-1,0}} \mathbb{E}_{\eta_m^{k-1,0}} \ldots \mathbb{E}_{\xi_m^{k,H^k-1}} \mathbb{E}_{\eta_m^{k,H^k-1}} \left\| \nabla f(x^{k,H^k}) \right\|^2 \\
& \quad + \frac{\gamma^3 L^2 (1-\theta)p}{2} \mathbb{E}_{H^{k-1}} \mathbb{E}_{\xi_m^{k-1,0}} \mathbb{E}_{\eta_m^{k-1,0}} \ldots \mathbb{E}_{\xi_m^{k-1,H^{k-1}-1}} \mathbb{E}_{\eta_m^{k-1,H^{k-1}-1}} \underbrace{\left\| g^{k,0} \right\|^2}_{=0} \\
& \quad + \frac{\gamma^3 L^2 (1-\theta)(1-p)}{2} \mathbb{E}_{H^{k-1}} \mathbb{E}_{H^k} \mathbb{E}_{\xi_m^{k-1,0}} \mathbb{E}_{\eta_m^{k-1,0}} \ldots \mathbb{E}_{\xi_m^{k,H^k-1}} \mathbb{E}_{\eta_m^{k,H^k-1}} \left\| g^{k,H^k} \right\|^2 \\
& \quad + \gamma\theta \mathbb{E}_{H^{k-1}} \mathbb{E}_{\xi_m^{k-1,0}} \mathbb{E}_{\eta_m^{k-1,0}} \ldots \mathbb{E}_{\xi_m^{k-1,H^{k-1}-1}} \mathbb{E}_{\eta_m^{k-1,H^{k-1}-1}} \left\| g^{k-1,H^{k-1}} \right\|^2 \\
& \quad + \frac{8\gamma^2 L(1-\theta)\delta_2}{\min\limits_{1 \leqslant m \leqslant M} q_m} + \frac{4\gamma^2 L(1-\theta)\sigma^2}{\min\limits_{1 \leqslant m \leqslant M} q_m}.
\end{aligned} \tag{39}
$$

We use Lemma E.1 to estimate $\left\|g^{k,H^k}\right\|^2$ and $\left\|g^{k-1,H^{k-1}}\right\|^2$. We have

$$
\mathbb{E}_{H^k}\mathbb{E}_{\xi_m^{k,0}}\mathbb{E}_{\eta_m^{k,0}}\ldots\mathbb{E}_{\xi_m^{k,H^k-1}}\mathbb{E}_{\eta_m^{k,H^k-1}}\left\|g^{k,H^k}\right\|^2 \leqslant \frac{96(1-\theta)^2\alpha(\delta_1+1)}{p^2\min\limits_{1\leqslant m\leqslant M}q_m}\mathbb{E}_{H^k}\left\|\nabla f(x^{k,H^k})\right\|^2
$$

$$
+\frac{192(1-\theta)^2\alpha\delta_2}{p^2\min\limits_{1\leqslant m\leqslant M}q_m}+\frac{96(1-\theta)^2\alpha\sigma^2}{p^2\min\limits_{1\leqslant m\leqslant M}q_m}. \quad (40)
$$

Next, analogously to equation 19, we choose $\gamma\leqslant\frac{p\cdot\sqrt{\min_{1\leqslant m\leqslant M}q_m}}{384L\sqrt{\alpha}\sqrt{\delta_1+1}}$ and obtain

$$
\mathbb{E}_{H^{k-1}}\mathbb{E}_{\xi_m^{k-1,0}}\mathbb{E}_{\eta_m^{k-1,0}}\ldots\mathbb{E}_{\xi_m^{k-1,H^{k-1}-1}}\mathbb{E}_{\eta_m^{k-1,H^{k-1}-1}}\left\|g^{k-1,H^{k-1}}\right\|^2
$$

$$
\leqslant\frac{384(1-\theta)^2\alpha(\delta_1+1)}{p^2\min\limits_{1\leqslant m\leqslant M}q_m}\mathbb{E}_{H^{k-1}}\mathbb{E}_{\xi_m^{k-1,0}}\mathbb{E}_{\eta_m^{k-1,0}}\ldots\mathbb{E}_{\xi_m^{k-1,H^{k-1}-1}}\mathbb{E}_{\eta_m^{k-1,H^{k-1}-1}}\left\|\nabla f(x^{k,0})\right\|^2
$$

$$
+\frac{384(1-\theta)^2\alpha\delta_2}{p^2\min\limits_{1\leqslant m\leqslant M}q_m}+\frac{192(1-\theta)^2\alpha\sigma^2}{p^2\min\limits_{1\leqslant m\leqslant M}q_m}. \quad (41)
$$

We combine equation 40 and equation 41 with equation 39 and use that $H^{k-1}$ with $\left\{\xi_m^{k-1,h}\right\}_{h=0}^{H^{k-1}-1}$ and $H^k$ with $\left\{\eta_m^{k-1,h}\right\}_{h=0}^{H^{k-1}-1}$ are independent stochastic values. Moreover we take full expectation:

$$
\begin{aligned}
p\mathbb{E}\left[f(\widetilde{x}^{k,H^k})\right] &\leqslant p\mathbb{E}\left[f(\widetilde{x}^{k,0})\right]\\
&-\frac{\gamma(1-\theta)p}{4}\mathbb{E}\left\|\nabla f(x^{k,0})\right\|^2-\frac{\gamma(1-\theta)(1-p)}{4}\mathbb{E}\left\|\nabla f(x^{k,H^k})\right\|^2\\
&+\frac{48\gamma^3L^2(1-\theta)^3(1-p)\alpha(\delta_1+1)}{p^2\min\limits_{1\leqslant m\leqslant M}q_m}\mathbb{E}\left\|\nabla f(x^{k,H^k})\right\|^2\\
&+\frac{192\gamma\theta(1-\theta)^2\alpha(\delta_1+1)}{p^2\min\limits_{1\leqslant m\leqslant M}q_m}\mathbb{E}\left\|\nabla f(x^{k,0})\right\|^2\\
&+\frac{96\gamma^3L^2(1-\theta)^3(1-p)\alpha\delta_2}{p^2\min\limits_{1\leqslant m\leqslant M}q_m}+\frac{384\gamma\theta(1-\theta)^2\alpha\delta_2}{p^2\min\limits_{1\leqslant m\leqslant M}q_m}+\frac{8\gamma^2L(1-\theta)\delta_2}{\min\limits_{1\leqslant m\leqslant M}q_m}\\
&+\frac{48\gamma^3L^2(1-\theta)^3(1-p)\alpha\sigma^2}{p^2\min\limits_{1\leqslant m\leqslant M}q_m}+\frac{192\gamma\theta(1-\theta)^2\alpha\sigma^2}{p^2\min\limits_{1\leqslant m\leqslant M}q_m}+\frac{4\gamma^2L(1-\theta)\sigma^2}{\min\limits_{1\leqslant m\leqslant M}q_m}\\
&= p\mathbb{E}\left[f(\widetilde{x}^{k,0})\right]\\
&-\frac{\gamma(1-\theta)(1-p)}{4}\left(1-\frac{192\gamma^2L^2(1-\theta)^2\alpha(\delta_1+1)}{p^2\min\limits_{1\leqslant m\leqslant M}q_m}\right)\mathbb{E}\left\|\nabla f(x^{k,H^k})\right\|^2\\
&-\frac{\gamma(1-\theta)p}{4}\left(1-\frac{384\theta(1-\theta)^2\alpha(\delta_1+1)}{p^3\min\limits_{1\leqslant m\leqslant M}q_m}\right)\mathbb{E}\left\|\nabla f(x^{k,0})\right\|^2\\
&+\frac{96\gamma^3L^2(1-\theta)^3(1-p)\alpha\delta_2}{p^2\min\limits_{1\leqslant m\leqslant M}q_m}+\frac{768\gamma\theta(1-\theta)^2\alpha\delta_2}{p^2\min\limits_{1\leqslant m\leqslant M}q_m}+\frac{8\gamma^2L(1-\theta)\delta_2}{\min\limits_{1\leqslant m\leqslant M}q_m}\\
&+\frac{48\gamma^3L^2(1-\theta)^3(1-p)\alpha\sigma^2}{p^2\min\limits_{1\leqslant m\leqslant M}q_m}+\frac{192\gamma\theta(1-\theta)^2\alpha\sigma^2}{p^2\min\limits_{1\leqslant m\leqslant M}q_m}+\frac{4\gamma^2L(1-\theta)\sigma^2}{\min\limits_{1\leqslant m\leqslant M}q_m}.
\end{aligned}
$$

We choose $\theta\leqslant\frac{\gamma Lp^2}{2}$ $\gamma\leqslant\frac{p\min_{1\leqslant m\leqslant M}q_m}{768L\alpha(\delta+1)}$. In that way,

$$p\mathbb{E}\left[f(\widetilde{x}^{k,H^k})\right] \;\leqslant\; p\mathbb{E}\left[f(\widetilde{x}^{k,0})\right] - \frac{\gamma(1-\theta)p}{8}\mathbb{E}\left\|\nabla f(x^{k,0})\right\|^2$$
$$+\frac{96\gamma^3 L^2\alpha\delta_2}{p^2\min\limits_{1\leqslant m\leqslant M} q_m} + \frac{192\gamma^2 L\alpha\delta_2}{\min\limits_{1\leqslant m\leqslant M} q_m} + \frac{8\gamma^2 L\delta_2}{\min\limits_{1\leqslant m\leqslant M} q_m}$$
$$+\frac{48\gamma^3 L^2\alpha\sigma^2}{p^2\min\limits_{1\leqslant m\leqslant M} q_m} + \frac{96\gamma^2 L\alpha\sigma^2}{\min\limits_{1\leqslant m\leqslant M} q_m} + \frac{4\gamma^2 L\sigma^2}{\min\limits_{1\leqslant m\leqslant M} q_m},$$

$$\frac{\gamma(1-\theta)}{8}\mathbb{E}\left\|\nabla f(x^{k,0})\right\|^2 \;\leqslant\; \mathbb{E}\left[f(\widetilde{x}^{k,0})\right] - \mathbb{E}\left[f(\widetilde{x}^{k,H^k})\right]$$
$$+\frac{96\gamma^3 L^2\alpha\delta_2}{p^3\min\limits_{1\leqslant m\leqslant M} q_m} + \frac{200\gamma^2 L\alpha\delta_2}{p\min\limits_{1\leqslant m\leqslant M} q_m}$$
$$+\frac{48\gamma^3 L^2\alpha\sigma^2}{p^3\min\limits_{1\leqslant m\leqslant M} q_m} + \frac{100\gamma^2 L\alpha\sigma^2}{p\min\limits_{1\leqslant m\leqslant M} q_m}.$$

Note that $\widetilde{x}^{k,H^k} = x^{k,H^k} - \gamma g^{k,H^k} = x^{k+1,0}$ and $\widetilde{x}^{k,0} = x^{k,0}$. Thus,

$$\frac{\gamma(1-\theta)}{8}\mathbb{E}\left\|\nabla f(x^{k,0})\right\|^2 \;\leqslant\; \mathbb{E}\left[f(x^{k,0})\right] - \mathbb{E}\left[f(x^{k+1,0})\right] + \frac{96\gamma^3 L^2\alpha\delta_2}{p^3\min\limits_{1\leqslant m\leqslant M} q_m} + \frac{200\gamma^2 L\alpha\delta_2}{p\min\limits_{1\leqslant m\leqslant M} q_m}$$
$$+\frac{48\gamma^3 L^2\alpha\sigma^2}{p^3\min\limits_{1\leqslant m\leqslant M} q_m} + \frac{100\gamma^2 L\alpha\sigma^2}{p\min\limits_{1\leqslant m\leqslant M} q_m}.$$

Summing over all iterations, we obtain the result of the theorem:

$$\frac{1}{K}\sum_{k=0}^{K-1}\mathbb{E}\left\|\nabla f(x^{k,0})\right\|^2 \;\leqslant\; \frac{8\left(f(x^{0,0}) - \mathbb{E}\left[f(x^{K,0})\right]\right)}{\gamma(1-\theta)K}$$
$$+\frac{768\gamma^2 L^2\alpha\delta_2}{p^3\min\limits_{1\leqslant m\leqslant M} q_m(1-\theta)} + \frac{1600\gamma L\alpha\delta_2}{p\min\limits_{1\leqslant m\leqslant M} q_m(1-\theta)}$$
$$+\frac{384\gamma^2 L^2\alpha\sigma^2}{p^3\min\limits_{1\leqslant m\leqslant M} q_m(1-\theta)} + \frac{800\gamma L\alpha\sigma^2}{p\min\limits_{1\leqslant m\leqslant M} q_m(1-\theta)}$$
$$\leqslant\; \frac{16\left(f(x^{0,0}) - f(x^*)\right)}{\gamma K}$$
$$+\frac{1536\gamma^2 L^2\alpha\delta_2}{p^3\min\limits_{1\leqslant m\leqslant M} q_m} + \frac{3200\gamma L\alpha\delta_2}{p\min\limits_{1\leqslant m\leqslant M} q_m}$$
$$+\frac{768\gamma^2 L^2\alpha\sigma^2}{p^3\min\limits_{1\leqslant m\leqslant M} q_m} + \frac{1600\gamma L\alpha\sigma^2}{p\min\limits_{1\leqslant m\leqslant M} q_m}.$$

$\square$

**Corollary E.3 (Corollary 3.8).** *Under conditions of Theorem E.2 Algorithm 2 with fixed rules* $\widehat{\mathcal{R}} \equiv \widetilde{\mathcal{R}} \equiv \mathcal{R}$ *needs*

$$\mathcal{O}\left(\frac{M}{C}\frac{1}{\min\limits_{1\leqslant m\leqslant M} q_m}\left(\frac{\Delta L\alpha\delta_1}{\varepsilon^2} + \frac{\Delta L\alpha\delta_2}{\varepsilon^4} + \frac{\Delta L\alpha\sigma^2}{\varepsilon^4}\right)\right)$$ *epochs and*

$$\mathcal{O}\left(M\frac{M}{C}\frac{1}{\min\limits_{1\leqslant m\leqslant M} q_m}\left(\frac{\Delta L\alpha\delta_1}{\varepsilon^2} + \frac{\Delta L\alpha\delta_2}{\varepsilon^4} + \frac{\Delta L\alpha\sigma^2}{\varepsilon^4}\right)\right)$$ *number of devices communications*

*to reach $\varepsilon$-accuracy, where $\varepsilon^2 = \frac{1}{K} \sum_{k=0}^{K-1} \mathbb{E} \left\| \nabla f(x^{k,0}) \right\|^2$, $\Delta = f(x^{0,0}) - f(x^*)$ and $C$ is the number of devices participating in each epoch.*

*Proof.* Proof is analogous to the proof of Corollary D.3. □

**Corollary E.4.** *Under conditions of Theorem E.2 Algorithm 2 needs*

$$\mathcal{O}\left( \frac{M}{\min\limits_{k,h} C^{k,h}} \frac{1}{\min\limits_{1 \leqslant m \leqslant M} q_m} \left( \frac{\Delta L \alpha \delta_1}{\varepsilon^2} + \frac{\Delta L \alpha \delta_2}{\varepsilon^4} + \frac{\Delta L \alpha \sigma^2}{\varepsilon^4} \right) \right) \quad \text{epochs and}$$

$$\mathcal{O}\left( M \left( \frac{M}{\min\limits_{k,h} C^{k,h}} \right)^2 \frac{1}{\min\limits_{1 \leqslant m \leqslant M} q_m} \left( \frac{\Delta L \alpha \delta_1}{\varepsilon^2} + \frac{\Delta L \alpha \delta_2}{\varepsilon^4} + \frac{\Delta L \alpha \sigma^2}{\varepsilon^4} \right) \right)$$

*number of devices communications to reach $\varepsilon$-accuracy, where $\varepsilon^2 = \frac{1}{K} \sum_{k=0}^{K-1} \mathbb{E} \left\| \nabla f(x^{k,0}) \right\|^2$, $\Delta = f(x^{0,0}) - f(x^*)$ and $C^{k,h}$ is the number of devices participating in $k$-th iteration in $h$-th epoch.*

*Proof.* Proof is analogous to the proof of Corollary D.4. □

*Remark* E.5. Considering fixed rules $\widehat{\mathcal{R}} \equiv \widetilde{\mathcal{R}} \equiv \mathcal{R}$,

we have $\mathcal{O}\left( M \frac{M}{C} \frac{1}{\min\limits_{1 \leqslant m \leqslant M} q_m} \left( \frac{\Delta L \delta_1}{\varepsilon^2} + \frac{\Delta L \delta_2}{\varepsilon^4} + \frac{\Delta L \sigma^2}{M \varepsilon^4} \right) \right)$

and $\mathcal{O}\left( M^2 \frac{M}{C} \frac{1}{\min\limits_{1 \leqslant m \leqslant M} q_m} \left( \frac{\Delta L \delta_1}{\varepsilon^2} + \frac{\Delta L \delta_2}{\varepsilon^4} + \frac{\Delta L \sigma^2}{M \varepsilon^4} \right) \right)$ number of devices communications with regularizing parameter $\alpha = 1$ and $\alpha = M$ respectively. Considering various rules, best case with regularizing coefficient $\alpha = 1$ gives us $\mathcal{O}\left( M \left( \frac{M}{\min\limits_{k,h} C^{k,h}} \right)^2 \frac{1}{\min\limits_{1 \leqslant m \leqslant M} q_m} \left( \frac{\Delta L \delta_1}{\varepsilon^2} + \frac{\Delta L \delta_2}{\varepsilon^4} + \frac{\Delta L \sigma^2}{M \varepsilon^4} \right) \right)$

and worst case $\alpha = M$ gives us $\mathcal{O}\left( M^2 \left( \frac{M}{\max\limits_{k,h} C^{k,h}} \right)^2 \frac{1}{\min\limits_{1 \leqslant m \leqslant M} q_m} \left( \frac{\Delta L \delta_1}{\varepsilon^2} + \frac{\Delta L \delta_2}{\varepsilon^4} + \frac{\Delta L \sigma^2}{M \varepsilon^4} \right) \right)$ number of devices communications.

## E.2 PROOF FOR STRONGLY-CONVEX SETTING

**Theorem E.6.** *Suppose Assumptions 2.1, 2.2(b), 2.3, 2.4 hold. Then for Algorithm 2 with $\theta \leqslant \frac{p \gamma \mu}{4}$ and $\gamma \leqslant \frac{p^2 \min_{1 \leqslant m \leqslant M} q_m}{384 L \alpha (\delta_1 + 1)}$ it implies that*

$$\mathbb{E} \left\| x^{K,0} - x^* \right\|^2 \quad \leqslant \quad \left( 1 - \frac{\gamma \mu}{8} \right)^K \mathbb{E} \left\| x^{0,0} - x^* \right\|^2 + \frac{2368 \gamma \alpha}{\mu p^3 \min\limits_{1 \leqslant m \leqslant M} q_m} (2 \delta_2 + \sigma^2).$$

*Proof.* We start with the definition of virtual sequence:

$$\widetilde{x}^{k,h} = x^{k,h} - \gamma \sum_{m=1}^{M} g_m^{k,h} = x^{k,h} - \gamma g^{k,h}. \tag{42}$$

It is followed by

$$\begin{aligned}
\widetilde{x}^{k,h+1} &= x^{k,h+1} - \gamma \sum_{m=1}^{M} g_m^{k,h+1} \\
&= x^{k,h} - \gamma \left[ (1-\theta) \sum_{m=1}^{M} \frac{\eta_m^{k,h}}{q_m} \hat{\pi}_m^{k,h} \nabla f_m(x^{k,h}, \xi_m^{k,h}) + \theta g^{k-1,H^{k-1}} \right] \\
&\quad - \gamma \sum_{m=1}^{M} g_m^{k,h} - \gamma(1-\theta) \sum_{m=1}^{M} \frac{\eta_m^{k,h}}{q_m} \left( \frac{1}{M} - \hat{\pi}_m^{k,h} \right) \nabla f_m(x^{k,h}, \xi_m^{k,h})
\end{aligned}$$

$$= \quad \widetilde{x}^{k,h} - \gamma \left[ (1-\theta) \frac{1}{M} \sum_{m=1}^{M} \frac{\eta_m^{k,h}}{q_m} \nabla f_m(x^{k,h}, \xi_m^{k,h}) + \theta g^{k-1,H^{k-1}} \right]. \tag{43}$$

Next, we use this to write a descent:

$$\begin{aligned}
\left\| \widetilde{x}^{k,h+1} - x^* \right\|^2 &= \quad \left\| \widetilde{x}^{k,h} - x^* \right\|^2 + 2 \left\langle \widetilde{x}^{k,h} - x^*, \widetilde{x}^{k,h+1} - \widetilde{x}^{k,h} \right\rangle + \left\| \widetilde{x}^{k,h+1} - \widetilde{x}^{k,h} \right\|^2 \\
&\overset{(43)}{=} \quad \left\| \widetilde{x}^{k,h} - x^* \right\|^2 - 2\gamma\theta \left\langle \widetilde{x}^{k,h} - x^*, g^{k-1,H^{k-1}} \right\rangle \\
&\quad -2\gamma(1-\theta) \left\langle \widetilde{x}^{k,h} - x^*, \frac{1}{M} \sum_{m=1}^{M} \frac{\eta_m^{k,h}}{q_m} \nabla f_m(x^{k,h}, \xi_m^{k,h}) \right\rangle \\
&\quad +\gamma^2 \left\| \theta g^{k-1,H^{k-1}} + (1-\theta) \frac{1}{M} \sum_{m=1}^{M} \frac{\eta_m^{k,h}}{q_m} \nabla f_m(x^{k,h}, \xi_m^{k,h}) \right\|^2 \\
&\overset{(Jen)}{\leqslant} \quad \left\| \widetilde{x}^{k,h} - x^* \right\|^2 - 2\gamma\theta \left\langle \widetilde{x}^{k,h} - x^*, g^{k-1,H^{k-1}} \right\rangle \\
&\quad -2\gamma(1-\theta) \left\langle \widetilde{x}^{k,h} - x^{k,h}, \frac{1}{M} \sum_{m=1}^{M} \frac{\eta_m^{k,h}}{q_m} \nabla f_m(x^{k,h}, \xi_m^{k,h}) \right\rangle \\
&\quad -2\gamma(1-\theta) \left\langle x^{k,h} - x^*, \frac{1}{M} \sum_{m=1}^{M} \frac{\eta_m^{k,h}}{q_m} \nabla f_m(x^{k,h}, \xi_m^{k,h}) \right\rangle \\
&\quad +\gamma^2\theta \left\| g^{k-1,H^{k-1}} \right\|^2 + \gamma^2(1-\theta) \left\| \frac{1}{M} \sum_{m=1}^{M} \frac{\eta_m^{k,h}}{q_m} \nabla f_m(x^{k,h}, \xi_m^{k,h}) \right\|^2.
\end{aligned}$$

Now we use that $\eta_m^{k,h} \sim \mathcal{B}(q_m)$. Consequently, $\mathbb{E}\eta_m^{k,h} = q_m$. Since $\eta_m^{k,h}$ is independent of $x^{k,h}, \widetilde{x}^{k,h}, \xi_m^{k,h}, g^{k-1,H^{k-1}}$, we take the expectation and obtain

$$\begin{aligned}
\mathbb{E}_{\eta_m^{k,h}} \left\| \widetilde{x}^{k,h+1} - x^* \right\|^2 &\leqslant \quad \left\| \widetilde{x}^{k,h} - x^* \right\|^2 - 2\gamma\theta \left\langle \widetilde{x}^{k,h} - x^*, g^{k-1,H^{k-1}} \right\rangle \\
&\quad -2\gamma(1-\theta) \left\langle \widetilde{x}^{k,h} - x^{k,h}, \frac{1}{M} \sum_{m=1}^{M} \nabla f_m(x^{k,h}, \xi_m^{k,h}) \right\rangle \\
&\quad -2\gamma(1-\theta) \left\langle x^{k,h} - x^*, \frac{1}{M} \sum_{m=1}^{M} \nabla f_m(x^{k,h}, \xi_m^{k,h}) \right\rangle \\
&\quad +\gamma^2\theta \left\| g^{k-1,H^{k-1}} \right\|^2 \\
&\quad +\gamma^2(1-\theta) \mathbb{E}_{\eta_m^{k,h}} \left\| \frac{1}{M} \sum_{m=1}^{M} \frac{\eta_m^{k,h}}{q_m} \nabla f_m(x^{k,h}, \xi_m^{k,h}) \right\|^2.
\end{aligned}$$

Now we take the expectation over $\xi_m^{k,h}$. Mention that

$$\begin{aligned}
\widetilde{x}^{k,h} &\overset{(42)}{=} \quad x^{k,h} - \gamma g^{k,h} \\
&\overset{\text{Line } 12}{=} \quad x^{k,h} - \gamma \left( g^{k,h-1} + (1-\theta) \sum_{m=1}^{M} \frac{\eta_m^{k,h-1}}{q_m} \left( \frac{1}{M} - \hat{\pi}_m^{k,h-1} \right) \nabla f(x^{k,h-1}, \xi_m^{k,h-1}) \right).
\end{aligned}$$

Thus, $\widetilde{x}^{k,h}$ and $\xi_m^{k,h}$ are independent. Analogously, $g^{k-1,H^{k-1}}$ and $\xi_m^{k,h}$ are independent. In this way,

$$\begin{aligned}
\mathbb{E}_{\xi_m^{k,h}} \mathbb{E}_{\eta_m^{k,h}} \left\| \widetilde{x}^{k,h+1} - x^* \right\|^2 &\leqslant \left\| \widetilde{x}^{k,h} - x^* \right\|^2 - 2\gamma\theta \left\langle \widetilde{x}^{k,h} - x^*, g^{k-1,H^{k-1}} \right\rangle \\
&\quad - 2\gamma(1-\theta) \left\langle \widetilde{x}^{k,h} - x^{k,h}, \nabla f(x^{k,h}) \right\rangle \\
&\quad - 2\gamma(1-\theta) \left\langle x^{k,h} - x^*, \nabla f(x^{k,h}) \right\rangle \\
&\quad + \gamma^2\theta \left\| g^{k-1,H^{k-1}} \right\|^2
\end{aligned}$$

$$+ \gamma^2(1-\theta)\mathbb{E}_{\xi_m^{k,h}}\mathbb{E}_{\eta_m^{k,h}}\left\|\frac{1}{M}\sum_{m=1}^{M}\frac{\eta_m^{k,h}}{q_m}\nabla f_m(x^{k,h},\xi_m^{k,h})\right\|^2. \quad (44)$$

Recall we estimated the last term in Theorem E.2 in equation 37:

$$\mathbb{E}_{\xi_m^{k,h}}\mathbb{E}_{\eta_m^{k,h}}\left\|\frac{1}{M}\sum_{m=1}^{M}\frac{\eta_m^{k,h}}{q_m}\nabla f_m(x^{k,h},\xi_m^{k,h})\right\|^2 \leqslant \frac{16(\delta_1+1)}{\min\limits_{1\leqslant m\leqslant M}q_m}\left\|\nabla f(x^{k,h})\right\|^2 + \frac{16\delta_2}{\min\limits_{1\leqslant m\leqslant M}q_m}$$
$$+\frac{8\sigma^2}{\min\limits_{1\leqslant m\leqslant M}q_m}. \quad (45)$$

Now let us estimate scalar products separately.

$$-2\gamma\theta\left\langle\widetilde{x}^{k,h}-x^*,g^{k-1,H^{k-1}}\right\rangle \overset{(Fen)}{\leqslant} \theta\left\|\widetilde{x}^{k,h}-x^*\right\|^2 + \gamma^2\theta\left\|g^{k-1,H^{k-1}}\right\|^2,$$

$$-2\gamma(1-\theta)\left\langle\widetilde{x}^{k,h}-x^{k,h},\nabla f(x^{k,h})\right\rangle \overset{(Fen)}{\leqslant} (1-\theta)\left\|\widetilde{x}^{k,h}-x^{k,h}\right\|^2$$
$$+\gamma^2(1-\theta)\left\|\nabla f(x^{k,h})\right\|^2$$
$$\overset{(20)}{=} \gamma^2(1-\theta)\left\|g^{k,h}\right\|^2 + \gamma^2(1-\theta)\left\|\nabla f(x^{k,h})\right\|^2,$$

$$-2\gamma(1-\theta)\left\langle x^{k,h}-x^*,\nabla f(x^{k,h})\right\rangle \overset{\text{As. 2.2(b)}}{\leqslant} -\gamma\mu(1-\theta)\left\|x^{k,h}-x^*\right\|^2$$
$$-2\gamma(1-\theta)\left[f(x^{k,h})-f(x^*)\right]$$
$$\overset{(CS)}{\leqslant} -\frac{\gamma\mu(1-\theta)}{2}\left\|\widetilde{x}^{k,h}-x^*\right\|^2$$
$$+\gamma\mu(1-\theta)\left\|x^{k,h}-\widetilde{x}^{k,h}\right\|^2$$
$$-2\gamma(1-\theta)\left[f(x^{k,h})-f(x^*)\right]$$
$$\overset{(20)}{=} -\frac{\gamma\mu(1-\theta)}{2}\left\|\widetilde{x}^{k,h}-x^*\right\|^2$$
$$+\gamma^3\mu(1-\theta)\left\|g^{k,h}\right\|^2$$
$$-2\gamma(1-\theta)\left[f(x^{k,h})-f(x^*)\right].$$

Substituting this estimates and equation 45 into equation 44,

$$\mathbb{E}_{\xi_m^{k,h}}\mathbb{E}_{\eta_m^{k,h}}\left\|\widetilde{x}^{k,h+1}-x^*\right\|^2 \leqslant \left(1-\frac{\gamma\mu(1-\theta)}{2}+\theta\right)\left\|\widetilde{x}^{k,h}-x^*\right\|^2 + 2\gamma^2\theta\left\|g^{k-1,H^{k-1}}\right\|^2$$
$$+\gamma^2(1-\theta)(1+\gamma\mu)\left\|g^{k,h}\right\|^2$$
$$+\frac{19\gamma^2(1-\theta)(\delta_1+1)}{\min\limits_{1\leqslant m\leqslant M}q_m}\left\|\nabla f(x^{k,h})\right\|^2$$
$$-2\gamma(1-\theta)\left[f(x^{k,h})-f(x^*)\right]$$
$$+\frac{16\gamma^2(1-\theta)\delta_2}{\min\limits_{1\leqslant m\leqslant M}q_m} + \frac{8\gamma^2(1-\theta)\sigma^2}{\min\limits_{1\leqslant m\leqslant M}q_m}.$$

Let us choose $\theta \leqslant \frac{\gamma\mu}{4}$ and $\gamma \leqslant \frac{1}{L}$. Then, $\left(1-\frac{\gamma\mu(1-\theta)}{2}+\theta\right) \leqslant \left(1-\frac{3\gamma\mu}{8}+\frac{\gamma\mu}{4}\right) = \left(1-\frac{\gamma\mu}{8}\right)$. In this way,

$$\mathbb{E}_{\xi_m^{k,h}}\mathbb{E}_{\eta_m^{k,h}}\left\|\widetilde{x}^{k,h+1}-x^*\right\|^2 \leqslant \left(1-\frac{\gamma\mu}{8}\right)\left\|\widetilde{x}^{k,h}-x^*\right\|^2 + 2\gamma^2\theta\left\|g^{k-1,H^{k-1}}\right\|^2$$
$$+\gamma^2(1-\theta)(1+\gamma\mu)\left\|g^{k,h}\right\|^2$$

$$+ \frac{19\gamma^2(1-\theta)(\delta_1+1)}{\min\limits_{1\leqslant m\leqslant M} q_m} \left\| \nabla f(x^{k,h}) \right\|^2$$

$$-2\gamma(1-\theta)\left[f(x^{k,h}) - f(x^*)\right]$$

$$+ \frac{16\gamma^2(1-\theta)\delta_2}{\min\limits_{1\leqslant m\leqslant M} q_m} + \frac{8\gamma^2(1-\theta)\sigma^2}{\min\limits_{1\leqslant m\leqslant M} q_m}.$$

Next, we estimate

$$\frac{19\gamma^2(1-\theta)(\delta_1+1)}{\min\limits_{1\leqslant m\leqslant M} q_m} \left\| \nabla f(x^{k,h}) \right\|^2 \leqslant \frac{38\gamma^2 L(1-\theta)(\delta_1+1)}{\min\limits_{1\leqslant m\leqslant M} q_m} \left[f(x^{k,h}) - f(x^*)\right].$$

It implies that

$$\begin{aligned}
\mathbb{E}_{\xi_m^{k,h}} \mathbb{E}_{\eta_m^{k,h}} \left\| \widetilde{x}^{k,h+1} - x^* \right\|^2 \quad \leqslant \quad & \left(1 - \frac{\gamma\mu}{8}\right) \left\| \widetilde{x}^{k,h} - x^* \right\|^2 + 2\gamma^2\theta \left\| g^{k-1,H^{k-1}} \right\|^2 \\
& + \gamma^2(1-\theta)(1+\gamma\mu) \left\| g^{k,h} \right\|^2 \\
& - 2\gamma(1-\theta)\left(1 - \frac{19\gamma L(\delta_1+1)}{\min\limits_{1\leqslant m\leqslant M} q_m}\right)\left[f(x^{k,h}) - f(x^*)\right] \\
& + \frac{16\gamma^2(1-\theta)\delta_2}{\min\limits_{1\leqslant m\leqslant M} q_m} + \frac{8\gamma^2(1-\theta)\sigma^2}{\min\limits_{1\leqslant m\leqslant M} q_m}.
\end{aligned}$$

Choosing $\gamma \leqslant \frac{\min_{1\leqslant m\leqslant M} q_m}{38L(\delta_1+1)}$, we can simplify as

$$\begin{aligned}
\mathbb{E}_{\xi_m^{k,h}} \mathbb{E}_{\eta_m^{k,h}} \left\| \widetilde{x}^{k,h+1} - x^* \right\|^2 \quad \leqslant \quad & \left(1 - \frac{\gamma\mu}{8}\right) \left\| \widetilde{x}^{k,h} - x^* \right\|^2 + 2\gamma^2\theta \left\| g^{k-1,H^{k-1}} \right\|^2 \\
& + 2\gamma^2(1-\theta) \left\| g^{k,h} \right\|^2 \\
& - \gamma(1-\theta)\left[f(x^{k,h}) - f(x^*)\right] \\
& + \frac{16\gamma^2(1-\theta)\delta_2}{\min\limits_{1\leqslant m\leqslant M} q_m} + \frac{8\gamma^2(1-\theta)\sigma^2}{\min\limits_{1\leqslant m\leqslant M} q_m}.
\end{aligned}$$

Now we put $h = H^k - 1$ and take additional expectations to obtain

$$\mathbb{E}_{\xi_m^{k-1,0}} \mathbb{E}_{\eta_m^{k-1,0}} \ldots \mathbb{E}_{\xi_m^{k,H^k-1}} \mathbb{E}_{\eta_m^{k,H^k-1}} \left\| \widetilde{x}^{k,H^k} - x^* \right\|^2$$

$$\leqslant \left(1 - \frac{\gamma\mu}{8}\right) \mathbb{E}_{\xi_m^{k-1,0}} \mathbb{E}_{\eta_m^{k-1,0}} \ldots \mathbb{E}_{\xi_m^{k,H^k-1}} \mathbb{E}_{\eta_m^{k,H^k-1}} \left\| \widetilde{x}^{k,H^k-1} - x^* \right\|^2$$

$$+ 2\gamma^2\theta \mathbb{E}_{\xi_m^{k-1,0}} \mathbb{E}_{\eta_m^{k-1,0}} \ldots \mathbb{E}_{\xi_m^{k-1,H^{k-1}-1}} \mathbb{E}_{\eta_m^{k-1,H^{k-1}-1}} \left\| g^{k-1,H^{k-1}} \right\|^2$$

$$+ 2\gamma^2(1-\theta) \mathbb{E}_{\xi_m^{k-1,0}} \mathbb{E}_{\eta_m^{k-1,0}} \ldots \mathbb{E}_{\xi_m^{k,H^k-1}} \mathbb{E}_{\eta_m^{k,H^k-1}} \left\| g^{k,H^k-1} \right\|^2$$

$$- \gamma(1-\theta) \mathbb{E}_{\xi_m^{k-1,0}} \mathbb{E}_{\eta_m^{k-1,0}} \ldots \mathbb{E}_{\xi_m^{k,H^k-1}} \mathbb{E}_{\eta_m^{k,H^k-1}} \left[f(x^{k,H^k-1}) - f(x^*)\right]$$

$$+ \frac{16\gamma^2(1-\theta)\delta_2}{\min\limits_{1\leqslant m\leqslant M} q_m} + \frac{8\gamma^2(1-\theta)\sigma^2}{\min\limits_{1\leqslant m\leqslant M} q_m}.$$

We take expectation with respect to $H^{k-1}$ and $H^k$, and apply Lemma C.1:

$$\mathbb{E}_{H^{k-1}} \mathbb{E}_{H^k} \mathbb{E}_{\xi_m^{k-1,0}} \mathbb{E}_{\eta_m^{k-1,0}} \ldots \mathbb{E}_{\xi_m^{k,H^k-1}} \mathbb{E}_{\eta_m^{k,H^k-1}} \left\| \widetilde{x}^{k,H^k} - x^* \right\|^2$$

$$\leqslant p\left(1 - \frac{\gamma\mu}{8}\right) \mathbb{E}_{H^{k-1}} \mathbb{E}_{\xi_m^{k-1,0}} \mathbb{E}_{\eta_m^{k-1,0}} \ldots \mathbb{E}_{\xi_m^{k-1,H^{k-1}-1}} \mathbb{E}_{\eta_m^{k-1,H^{k-1}-1}} \left\| \widetilde{x}^{k,0} - x^* \right\|^2$$

$$+(1-p)\left(1-\frac{\gamma\mu}{8}\right)\mathbb{E}_{H^{k-1}}\mathbb{E}_{H^k}\mathbb{E}_{\xi_m^{k-1,0}}\mathbb{E}_{\eta_m^{k-1,0}}\ldots\mathbb{E}_{\xi_m^{k,H^k-1}}\mathbb{E}_{\eta_m^{k,H^k-1}}\left\|\widetilde{x}^{k,H^k}-x^*\right\|^2$$

$$+2\gamma^2\theta\mathbb{E}_{H^{k-1}}\mathbb{E}_{\xi_m^{k-1,0}}\mathbb{E}_{\eta_m^{k-1,0}}\ldots\mathbb{E}_{\xi_m^{k-1,H^{k-1}-1}}\mathbb{E}_{\eta_m^{k-1,H^{k-1}-1}}\left\|g^{k-1,H^{k-1}}\right\|^2$$

$$+2\gamma^2(1-\theta)(1-p)\mathbb{E}_{H^{k-1}}\mathbb{E}_{H^k}\mathbb{E}_{\xi_m^{k-1,0}}\mathbb{E}_{\eta_m^{k-1,0}}\ldots\mathbb{E}_{\xi_m^{k,H^k-1}}\mathbb{E}_{\eta_m^{k,H^k-1}}\left\|g^{k,H^k}\right\|^2$$

$$+2\gamma^2(1-\theta)p\mathbb{E}_{H^{k-1}}\mathbb{E}_{\xi_m^{k-1,0}}\mathbb{E}_{\eta_m^{k-1,0}}\ldots\mathbb{E}_{\xi_m^{k-1,H^{k-1}-1}}\mathbb{E}_{\eta_m^{k-1,H^{k-1}-1}}\underbrace{\left\|g^{k,0}\right\|^2}_{=0}$$

$$-\gamma(1-p)(1-\theta)\mathbb{E}_{H^{k-1}}\mathbb{E}_{H^k}\mathbb{E}_{\xi_m^{k-1,0}}\mathbb{E}_{\eta_m^{k-1,0}}\ldots\mathbb{E}_{\xi_m^{k,H^k-1}}\mathbb{E}_{\eta_m^{k,H^k-1}}\left[f(x^{k,H^k})-f(x^*)\right]$$

$$-\gamma p(1-\theta)\mathbb{E}_{H^{k-1}}\mathbb{E}_{\xi_m^{k-1,0}}\mathbb{E}_{\eta_m^{k-1,0}}\ldots\mathbb{E}_{\xi_m^{k-1,H^{k-1}-1}}\mathbb{E}_{\eta_m^{k-1,H^{k-1}-1}}\left[f(x^{k,0})-f(x^*)\right]$$

$$+\frac{16\gamma^2(1-\theta)\delta_2}{\min\limits_{1\leqslant m\leqslant M}q_m}+\frac{8\gamma^2(1-\theta)\sigma^2}{\min\limits_{1\leqslant m\leqslant M}q_m}. \tag{46}$$

We use Lemma E.1 to estimate $\left\|g^{k,H^k}\right\|^2$ and $\left\|g^{k-1,H^{k-1}}\right\|^2$. We have

$$\mathbb{E}_{H^k}\mathbb{E}_{\xi_m^{k,0}}\mathbb{E}_{\eta_m^{k,0}}\ldots\mathbb{E}_{\xi_m^{k,H^k-1}}\mathbb{E}_{\eta_m^{k,H^k-1}}\left\|g^{k,H^k}\right\|^2 \leqslant \frac{96(1-\theta)^2\alpha(\delta_1+1)}{p^2\min\limits_{1\leqslant m\leqslant M}q_m}\mathbb{E}_{H^k}\left\|\nabla f(x^{k,H^k})\right\|^2$$

$$+\frac{192(1-\theta)^2\alpha\delta_2}{p^2\min\limits_{1\leqslant m\leqslant M}q_m}+\frac{96(1-\theta)^2\alpha\sigma^2}{p^2\min\limits_{1\leqslant m\leqslant M}q_m}. \tag{47}$$

Next, analogously to equation 29, we choose $\gamma\leqslant\frac{p\cdot\sqrt{\min_{1\leqslant m\leqslant M}q_m}}{384L\sqrt{\alpha}\sqrt{\delta_1+1}}$ and obtain

$$\mathbb{E}_{H^{k-1}}\mathbb{E}_{\xi_m^{k-1,0}}\mathbb{E}_{\eta_m^{k-1,0}}\ldots\mathbb{E}_{\xi_m^{k-1,H^{k-1}-1}}\mathbb{E}_{\eta_m^{k-1,H^{k-1}-1}}\left\|g^{k-1,H^{k-1}}\right\|^2$$

$$\leqslant\frac{384(1-\theta)^2\alpha(\delta_1+1)}{p^2\min\limits_{1\leqslant m\leqslant M}q_m}\mathbb{E}_{H^{k-1}}\mathbb{E}_{\xi_m^{k-1,0}}\mathbb{E}_{\eta_m^{k-1,0}}\ldots\mathbb{E}_{\xi_m^{k-1,H^{k-1}-1}}\mathbb{E}_{\eta_m^{k-1,H^{k-1}-1}}\left\|\nabla f(x^{k,0})\right\|^2$$

$$+\frac{384(1-\theta)^2\alpha\delta_2}{p^2\min\limits_{1\leqslant m\leqslant M}q_m}+\frac{192(1-\theta)^2\alpha\sigma^2}{p^2\min\limits_{1\leqslant m\leqslant M}q_m}. \tag{48}$$

Now we use equation Lip and that $H^k$ with $\left\{\xi_m^{k-1,h}\right\}_{h=0}^{H^{k-1}-1}$ and $H^{k-1}$ with $\left\{\eta_m^{k-1,h}\right\}_{h=0}^{H^{k-1}-1}$ are independent stochastic values. Moreover, we combine equation 47 and equation 48 with equation 46 and take full expectation.

$$p\mathbb{E}\left\|\widetilde{x}^{k,H^k}-x^*\right\|^2 \leqslant p\left(1-\frac{\gamma\mu}{8}\right)\mathbb{E}\left\|\widetilde{x}^{k,0}-x^*\right\|^2$$

$$-\gamma(1-p)(1-\theta)\mathbb{E}\left[f(x^{k,H^k})-f(x^*)\right]$$

$$-\gamma p(1-\theta)\mathbb{E}\left[f(x^{k,0})-f(x^*)\right]$$

$$+\frac{384\gamma^2L(1-\theta)^3(1-p)\alpha(\delta_1+1)}{p^2\min\limits_{1\leqslant m\leqslant M}q_m}\mathbb{E}\left[f(x^{k,H^k})-f(x^*)\right]$$

$$+\frac{1536\gamma^2L\theta(1-\theta)^2\alpha(\delta_1+1)}{p^2\min\limits_{1\leqslant m\leqslant M}q_m}\mathbb{E}\left[f(x^{k,0})-f(x^*)\right]$$

$$+\frac{384\gamma^2(1-\theta)^3(1-p)\alpha\delta_2}{p^2\min\limits_{1\leqslant m\leqslant M}q_m}+\frac{768\gamma^2\theta(1-\theta)^2\alpha\delta_2}{p^2\min\limits_{1\leqslant m\leqslant M}q_m}+\frac{16\gamma^2(1-\theta)\delta_2}{\min\limits_{1\leqslant m\leqslant M}q_m}$$

$$+\frac{192\gamma^2(1-\theta)^3(1-p)\alpha\sigma^2}{p^2\min\limits_{1\leqslant m\leqslant M}q_m}+\frac{384\gamma^2\theta(1-\theta)^2\alpha\sigma^2}{p^2\min\limits_{1\leqslant m\leqslant M}q_m}+\frac{8\gamma^2(1-\theta)\sigma^2}{\min\limits_{1\leqslant m\leqslant M}q_m}$$

$$\leqslant \quad p\left(1 - \frac{\gamma\mu}{8}\right) \mathbb{E}\left\|\widetilde{x}^{k,0} - x^*\right\|^2$$

$$-\gamma(1-p)(1-\theta)\left(1 - \frac{384\gamma L(1-\theta)^2\alpha(\delta_1+1)}{p^2 \min\limits_{1\leqslant m\leqslant M} q_m}\right)$$

$$\cdot\mathbb{E}\left[f(x^{k,H^k}) - f(x^*)\right]$$

$$-\gamma p(1-\theta)\left(1 - \frac{1536\gamma L\theta(1-\theta)\alpha(\delta_1+1)}{p^3 \min\limits_{1\leqslant m\leqslant M} q_m}\right)\mathbb{E}\left[f(x^{k,0}) - f(x^*)\right]$$

$$+\frac{384\gamma^2(1-\theta)^3(1-p)\alpha\delta_2}{p^2 \min\limits_{1\leqslant m\leqslant M} q_m} + \frac{768\gamma^2\theta(1-\theta)^2\alpha\delta_2}{p^2 \min\limits_{1\leqslant m\leqslant M} q_m} + \frac{16\gamma^2(1-\theta)\delta_2}{\min\limits_{1\leqslant m\leqslant M} q_m}$$

$$+\frac{192\gamma^2(1-\theta)^3(1-p)\alpha\sigma^2}{p^2 \min\limits_{1\leqslant m\leqslant M} q_m} + \frac{384\gamma^2\theta(1-\theta)^2\alpha\sigma^2}{p^2 \min\limits_{1\leqslant m\leqslant M} q_m} + \frac{8\gamma^2(1-\theta)\sigma^2}{\min\limits_{1\leqslant m\leqslant M} q_m}.$$

Choosing $\theta \leqslant \frac{p\gamma\mu}{4}$ and $\gamma \leqslant \frac{p^2 \min_{1\leqslant m\leqslant M} q_m}{384L\alpha(\delta_1+1)}$, we obtain

$$\mathbb{E}\left\|\widetilde{x}^{k,H^k} - x^*\right\|^2 \quad\leqslant\quad \left(1 - \frac{\gamma\mu}{8}\right)\mathbb{E}\left\|\widetilde{x}^{k,0} - x^*\right\|^2 + \frac{296\gamma\alpha}{p^3 \min\limits_{1\leqslant m\leqslant M} q_m}(2\delta_2 + \sigma^2).$$

Note that $\widetilde{x}^{k,H^k} = x^{k,H^k} - \gamma g^{k,H^k} = x^{k+1,0}$ and $\widetilde{x}^{k,0} = x^{k,0}$. Thus,

$$\mathbb{E}\left\|x^{k+1,0} - x^*\right\|^2 \quad\leqslant\quad \left(1 - \frac{\gamma\mu}{8}\right)\mathbb{E}\left\|x^{k,0} - x^*\right\|^2 + \frac{296\gamma^2\alpha}{p^3 \min\limits_{1\leqslant m\leqslant M} q_m}(2\delta_2 + \sigma^2).$$

It remains for us to going into recursion over all epochs and the result of the theorem:

$$\mathbb{E}\left\|x^{K,0} - x^*\right\|^2 \quad\leqslant\quad \left(1 - \frac{\gamma\mu}{8}\right)^K \mathbb{E}\left\|x^{0,0} - x^*\right\|^2 + \frac{296\gamma^2\alpha}{p^3 \min\limits_{1\leqslant m\leqslant M} q_m}(2\delta_2 + \sigma^2)\sum_{k=0}^{K}\left(1 - \frac{\gamma\mu}{8}\right)^k$$

$$\leqslant\quad \left(1 - \frac{\gamma\mu}{8}\right)^K \mathbb{E}\left\|x^{0,0} - x^*\right\|^2 + \frac{2368\gamma\alpha}{\mu p^3 \min\limits_{1\leqslant m\leqslant M} q_m}(2\delta_2 + \sigma^2).$$

$\square$

**Corollary E.7** (**Corollary 3.10**). *Under conditions of Theorem E.6 Algorithm 2 with fixed rules* $\widehat{\mathcal{R}} \equiv \widetilde{\mathcal{R}} \equiv \mathcal{R}$ *needs*

$$\widetilde{\mathcal{O}}\left(\left(\frac{M}{C}\right)^2 \frac{1}{\min\limits_{1\leqslant m\leqslant M} q_m}\left(\frac{L}{\mu}\alpha\delta_1 \log\left(\frac{1}{\varepsilon}\right) + \frac{M}{C}\frac{\alpha\delta_2}{\mu^2\varepsilon} + \frac{M}{C}\frac{\alpha\sigma^2}{\mu^2\varepsilon}\right)\right)$$

*epochs and*

$$\widetilde{\mathcal{O}}\left(M\left(\frac{M}{C}\right)^2 \frac{1}{\min\limits_{1\leqslant m\leqslant M} q_m}\left(\frac{L}{\mu}\alpha\delta_1 \log\left(\frac{1}{\varepsilon}\right) + \frac{M}{C}\frac{\alpha\delta_2}{\mu^2\varepsilon} + \frac{M}{C}\frac{\alpha\sigma^2}{\mu^2\varepsilon}\right)\right)$$

*number of devices communications*

*to reach $\varepsilon$-accuracy, where $\varepsilon^2 = \mathbb{E}\left\|x^{K,0} - x^*\right\|^2$ and $C$ is number of devices participating in each epoch.*

*Proof.* Proof is analogous to the proof of Corollary D.7. $\square$

**Corollary E.8.** *Under conditions of Theorem E.6 Algorithm 2 needs*

$$\widetilde{\mathcal{O}}\left(\left(\frac{M}{\min\limits_{k,h}C^{k,h}}\right)^2 \frac{1}{\min\limits_{1\leqslant m\leqslant M}q_m}\left(\frac{L}{\mu}\alpha\delta_1\log\left(\frac{1}{\varepsilon}\right)+\frac{M}{\min\limits_{k,h}C^{k,h}}\frac{\alpha\delta_2}{\mu^2\varepsilon}+\frac{M}{\min\limits_{k,h}C^{k,h}}\frac{\alpha\sigma^2}{\mu^2\varepsilon}\right)\right)$$

*epochs or*

$$\widetilde{\mathcal{O}}\left(M\left(\frac{M}{\min\limits_{k,h}C^{k,h}}\right)^3 \frac{1}{\min\limits_{1\leqslant m\leqslant M}q_m}\left(\frac{L}{\mu}\alpha\delta_1\log\left(\frac{1}{\varepsilon}\right)+\frac{M}{\min\limits_{k,h}C^{k,h}}\frac{\alpha\delta_2}{\mu^2\varepsilon}+\frac{M}{\min\limits_{k,h}C^{k,h}}\frac{\alpha\sigma^2}{\mu^2\varepsilon}\right)\right)$$

*communications*

*to reach $\varepsilon$-accuracy, where $\varepsilon^2 = \mathbb{E}\left\|x^{K,0}-x^*\right\|^2$ and $C^{k,h}$ is the number of devices participating in $k$-th iteration in $h$-th epoch.*

*Proof.* Proof is analogous to the proof of Corollary D.8. □

*Remark* E.9. Considering fixed rules $\widehat{\mathcal{R}} \equiv \widetilde{\mathcal{R}} \equiv \mathcal{R}$,

we have $\widetilde{\mathcal{O}}\left(M\left(\frac{M}{C}\right)^2 \frac{1}{\min\limits_{1\leqslant m\leqslant M}q_m}\left(\frac{L}{\mu}\delta_1\log\left(\frac{1}{\varepsilon}\right)+\frac{M}{C}\frac{\delta_2}{\mu^2\varepsilon}+\frac{M}{C}\frac{\sigma^2}{\mu^2\varepsilon}\right)\right)$

and $\widetilde{\mathcal{O}}\left(M^2\left(\frac{M}{C}\right)^2 \frac{1}{\min\limits_{1\leqslant m\leqslant M}q_m}\left(\frac{L}{\mu}\delta_1\log\left(\frac{1}{\varepsilon}\right)+\frac{M}{C}\frac{\delta_2}{\mu^2\varepsilon}+\frac{M}{C}\frac{\sigma^2}{\mu^2\varepsilon}\right)\right)$ number of devices communications with regularizing parameter $\alpha = 1$ and $\alpha = M$ respectively. Considering various rules, best case with regularizing coefficient $\alpha = 1$ gives us

$\widetilde{\mathcal{O}}\left(M\left(\frac{M}{\min\limits_{k,h}C^{k,h}}\right)^3 \frac{1}{\min\limits_{1\leqslant m\leqslant M}q_m}\left(\frac{L}{\mu}\delta_1\log\left(\frac{1}{\varepsilon}\right)+\frac{M}{\min\limits_{k,h}C^{k,h}}\frac{\delta_2}{\mu^2\varepsilon}+\frac{M}{\min\limits_{k,h}C^{k,h}}\frac{\sigma^2}{\mu^2\varepsilon}\right)\right)$ and worst case $\alpha =$

$M$ gives us $\widetilde{\mathcal{O}}\left(M^2\left(\frac{M}{\min\limits_{k,h}C^{k,h}}\right)^3 \frac{1}{\min\limits_{1\leqslant m\leqslant M}q_m}\left(\frac{L}{\mu}\delta_1\log\left(\frac{1}{\varepsilon}\right)+\frac{M}{\min\limits_{k,h}C^{k,h}}\frac{\delta_2}{\mu^2\varepsilon}+\frac{M}{\min\limits_{k,h}C^{k,h}}\frac{\sigma^2}{\mu^2\varepsilon}\right)\right)$ number of devices communications.

## THE USE OF LARGE LANGUAGE MODELS (LLMS)

In this work, large language models (LLMs) were used exclusively for spelling edits.

