# OpenReview forum: "A Strategy-Agnostic Framework for Partial Participation in Federated Learning"
_ICLR.cc/2026/Conference — Submitted to ICLR 2026_

### Official Review · Reviewer_V3pL · 2025-10-15

**Soundness:** 2
**Presentation:** 2
**Contribution:** 2
**Rating:** 2
**Confidence:** 4

**Summary:**

This paper proposes PPBC, a strategy-agnostic framework for federated learning with partial client participation, and its variant PPBC+, which handles unavailable or offline clients. The framework introduces a surrogate-gradient accumulation mechanism to compensate for biased or stochastic client sampling and provides convergence guarantees under both non-convex and strongly convex objectives. It also provides theoretical results of the proposed PPBC under nonconvex and strongly convex conditions. Experiments on CIFAR-10 and Food101 with ResNet-18 and FasterViT show that PPBC and PPBC+ outperform baselines across various sampling strategies.

**Strengths:**

1. This paper aims to design a strategy-agnostic PPBC, which integrates different client sampling rules and participation strategies.

2. The authors provided the theoretical analysis under nonconvexity and strong convexity.

3. Experimentally, PPBC shows obvious advantages compared to all baselines. (Need to be verified if the authors could provide more evidence, such as source code. )

**Weaknesses:**

W1. According to Line 20 of Algorithm 1, the proposed PPBC requires all clients to send $g_{m}^{k,H^k}$ to the server.

W2. The Assumption 2.3 does not specify the ranges of the constants $\delta_1$ and $\delta_2$, even though this assumption is commonly used.

W3. In the description of Algorithm 1 (lines 251-253), it says that the number of participating clients can vary. However, according to the definition of $C$ in Corollaries 3.3, 3.5, 3.6 and 3.7, the number of participating clients in each epoch in Algorithms 1 and 2 is fixed. It leads to an inconsistency between the algorithm and theory, which is my big concern about this paper. The case in the theory is just a special case of their proposed algorithm.

W4. In Sections 3.2-3.4, the discussion on theoretical results could be deeper and enriched. For example, it will be helpful for readers to understand the theoretical connection if the authors can specifically compare the theoretical results of partial participation with and without unavailable clients.

W5. The use of a Top-$C$ client selection in Section 4.1 conflicts with the definition of full client participation. Full participation means all clients are sampled in each round. Selecting only the Top-$C$ clients results in partial participation, and therefore, the experimental terminology and settings should be clarified or corrected.

W6. For the experiment results in Tables 1 and 2, the proposed PPBC achieved a significant improvement in test accuracy, compared to the baselines. Based on my personal experience, it is uncommon. Moreover, I did not find any source code in the supplementary materials, which makes it difficult for me to verify the reproducibility of the reported results.

**Minor:**

W7. Line 261: "an ablation studies" -> "ablation studies"

w8. Line 335: "we refuse using" -> "we refuse to use"

**Questions:**

Q1. Can the authors explain the motivation to use the Geometric distribution to generate different numbers of iterations? Can it bring any advantage?

Q2. Can the authors explain how the unavailable clients affect the algorithm in theory?

Q3. In Algorithm 1, the server performs client sampling twice (in lines 5 and 10). Could the authors clarify why two separate sampling steps are necessary? Or what is the motivation behind that?

Q4. Can the authors provide a communication cost analysis or comparison, theoretically or experimentally, of PPBC, PPBC+, and baselines? Since PPBC and PPBC+ need to communicate more than FedAvg, the analysis or comparison will enhance the understanding of the proposed algorithm.

I would consider updating my rate if my concerns can be addressed properly.

---

> ### Author Response · Authors · 2025-11-20
>
> Dear reviewer V3pL!
>
> We appreciate their time and the careful consideration reviewer have given to our work! Below we answer for concerns.
>
> 1. > According to Line 20 of Algorithm 1...
>
> We acknowledge the reviewer’s concern about the potential communication bottleneck. However, it occurs once per epoch. The fundamental principle of partial participation methods is precisely to reduce the frequency of full aggregation. This issue exists in all practical approaches including FedAvg and Scaffold [1].
>
> 2. > The Assumption 2.3 does not specify...
>
> While Assumption 2.3 does not explicitly
> specify the ranges of $\delta_1$ and $\delta_2$, this is standard practice in the
> literature. The assumption is mild and only requires that gradient heterogeneity
> is bounded *on average*, allowing some local gradients to deviate
> significantly as long as others compensate. In highly non-IID settings,
> $\delta_1$ and $\delta_2$ naturally increase, which is fully reflected in our
> convergence bounds (Corollaries 3.3, 3.5, 3.6, 3.7).
>
> We would also like to note that existing partial-participation works rely on
> *much stricter* assumptions, such as bounded stochastic gradients
> [2](ICLR 2020), [3](AISTATS 2022), [4](IEEE 2022) or explicit similarity between individual
> local gradients and the global gradient [5](NeurIPS 2022). Compared to these,
> our assumption is weaker and more realistic. Importantly, our method converges
> even under poor gradient alignment, as supported by both theory and experiments.
>
> 3. > In the description of Algorithm 1, it says that the number of participating...
>
> Thanks to the reviewer for pointing this out. Indeed, the referenced corollaries address the setting in which the client-selection rules are fixed and the number of participating clients remains the same across all epochs and iterations. In contrast, our analysis explicitly covers the more general and practically relevant scenario where both the strategy and the number of participating clients may vary from one iteration to another. Specifically, Corollaries D.4, D.8, E.4, and E.8 in our appendix extend the aforementioned results to this dynamic setting(Remarks D.5, D.9, E.5, E.9), so ensuring that our convergence guarantees remain valid under varying participation patterns.
>
> 4. > In Sections 3.2-3.4, the discussion on theoretical results could be deeper...(W4/Q2)
>
> Thanks to the reviewer for the suggestion to improve the paper! The difference in the convergence bounds of the PPBC and PPBC+ algorithms is entirely natural. Since PPBC+ incorporates the theoretical possibility that clients may drop out during training, the corresponding dropout effect appears in the bound. For each device, we model dropouts using a Bernoulli random variable with parameter $q_m$. As a result, the bound deteriorates by a factor of $\min_m q_m$. We will include this discussion in the final version of the paper.
>
> 5. > The use of a Top-**C** client selection in Section 4.1 conflicts...
>
> Thanks to Reviewer for the clarification. We will revise the text to ensure precise terminology.
> In Section 4.1, all clients are available, but only a subset communicates at each
> iteration to reduce communication cost. In contrast, Section 4.2 considers true partial
> participation, where some clients do not participate in training for an entire epoch.
> We will update the section to clearly distinguish these two settings.
>
> 6. > For the experiment results in Tables 1 and 2, the proposed PPBC ...
>
> We would like to note that the implementation code has already been provided
> through an anonymous GitHub link, as referenced in line 847 of Appendix B.
>
> 7. > W7, W8
>
> Thanks, typo, fixed!
>
> 8. > Can the authors explain the motivation to use the Geometric...
>
> Thanks to the reviewer for the interesting question!
> In our analysis, we use a geometrically distributed random variable as the epoch length in order to bound the norm of the accumulated surrogate gradient within an epoch. We present this theoretical bound as a separate lemmas (Lemmas D.1, E.1). From an algorithmic perspective, the community typically does not differentiate much between using a geometric or a Bernoulli random variable to model the epoch length.
>
> Moreover, in our work we identify a strong coupling between the hyperparameters
> $p$ and $\theta$. By varying $p$, and thus changing the expected epoch length, we can control performance through $\theta$: decreasing  $p$  (increasing the epoch length) reduces the contribution of the accumulated surrogates by lowering $\theta$(Table 6).
> The inverse relationship is observed as well(Table 5). A detailed discussion of this ablation study can be found in Appendix B.
>
> We kindly ask the reviewer to follow our next Official Comment, in which we continue our response.

---

> > ### Comment · Reviewer_V3pL · 2025-11-21
> >
> > I really appreciate the authors' detailed response. However, I still have the following concerns.
> >
> > - (To your point #1) This is my **core concern**. For W1, what I worried is not about communication cost but the full client availability. I know that in Algorithm 2, you provided a PPBC+ supporting partial client participation with unavailable devices. However, it is not real partial participation due to Line 20 in Algorithm 2. Hence, I would like to ask: if some clients are offline or unavailable, which means that they cannot upload and receive anything in a certain period of time, then how do your algorithms 1 and 2 work? Requiring all clients to upload information is usually impractical in reality. Moreover, FedAvg and Scaffold do not require full client availability under partial participation, which is a key difference between them and your proposed method.
> > - (To your point #3) I recommend that the authors move varying participating client case to the main paper because I think it is more important than fixed $C$. Anyway, this is just my personal opinion.
> > - (To your point #6) Thank you for your clarification!
> > - (To your point #8) May I understand it as the use of a geometric distribution for the epoch length is motivated mainly by theoretical proof, rather than by an intrinsic requirement of your proposed algorithm?

---

> ### Author Response · Authors · 2025-11-20
>
> 9. > In Algorithm 1, the server performs client sampling twice...
>
> Thank for the insightful question!
> Our method employs two client-selection strategies: one at the beginning of each epoch and another at the beginning of each iteration. This design serves two purposes. First, it reduces communication overhead by selecting only a subset of clients at each iteration, while the remaining clients accumulate surrogate gradients locally. As a result, full communication with all clients occurs only once per epoch after all iterations are completed whereas during the iterations, communication is performed solely with the selected subset of clients.
>
> 10. > Can the authors provide a communication cost analysis or comparison, theoretically...
>
> a. All theoretical communication costs were presented in Corollaries 3.3, 3.5, 3.6 and 3.7; D.4, D.8, E.4, and E.8 .
>
> b. For the experimental part, we used a fixed number of clients per iteration specifically, 1 client in each iteration. Thus, in our experimental setup, where we performed on average 3 iterations per epoch, the communication complexity of each epoch does not exceed $3+n$, where
> $n=10$ is the total number of clients.
> Consequently, over 100 epochs on CIFAR-10 with ResNet-18, the total communication cost corresponds to transmitting 1300 vectors, each having the dimensionality of the model. For FOOD101 with ViT over 15 epochs, 195 such vectors were transmitted.
> For the PoC[6] method, an additional transmission of losses (13 scalar values per epoch) may occur. In total, compared to standard FedAvg, this results in approximately 23% more transmitted information over the full training run.
>
> c. It is worth noting that, relative to methods that use a fixed set of clients every epoch, our approach significantly outperforms them in terms of accuracy, which compensates for the additional communication overhead that naturally arises in partial-participation settings.
>
> ---
>
> **References**
>
> [1] Karimireddy et al. Scaffold ... **ICML** 2020
>
> [2] Li et al. On the Convergence of FedAvg... **ICLR** 2020
>
> [3] Cho et al. Towards understanding biased client selection in federated learning. **AISTATS**  2022
>
> [4] Luo et al. Tackling system and statistical heterogeneity... **IEEE** 2022
>
> [5] Wang and Ji. A unified analysis of federated learning... **NeurIPS**,  2022
>
> [6] Cho et al. Towards understanding biased client selection in
> federated learning. **AISTATS** 2022.

---

> ### Author Response · Authors · 2025-11-21
>
> Thank to the reviewer for the quick response!
>
> 1. > This is my core concern...
>
> a. Yes, our framework requires a fixed time point at which the global communication occurs. In other words, we do not consider an asynchronous setup. In this sense, our work preserves the partial-participation setting adopted in prior works([1], [2]). We view asynchronous analysis as a fundamentally different problem. Moreover, based on known lower bounds on convergence ([3], Table 1), in an asynchronous setup with heterogeneous data distributions it is effectively necessary to wait for the slowest transmission. This implies that if our algorithm were to wait for the last client at this fixed final communication moment, it would not affect the potential convergence guarantees in such a setup. Therefore, we can make the relaxation of asynchronous regime by a setup in which there exists a time point at which full communication takes place.
>
> b. The reviewer mentioned that FedAvg and SCAFFOLD do not require full aggregation. From a theoretical standpoint, this is indeed correct. However, as noted above, in an asynchronous setup full aggregation becomes necessary to guarantee convergence in the presence of data heterogeneity. Moreover, based on our experimental results (Table 2 and Figure 2), where we compare our algorithm with high client dropout probability against baselines operating with zero dropout probability, our method still demonstrates superior performance. For this reason, we have no doubt that even when considering partial communication, we would continue to show stronger results.
>
> 2. >  I recommend that the authors move varying participating client ...
>
> Thanks, we have edited appropriate corollaries into the main part. The reviewr can see it in the revised version which we have uploaded on the OpenReview.
>
> 3. Thanks!
>
> 4. > May I understand it as the use of a geometric distribution for the epoch...
>
> Yes, mainly by theoretical proof. We believe that we have addressed this question in detail in our previous Official Comment.
>
> ---
>
> **References**
>
> [1]  Wang et al. Tackling the objective ... **NeurIPS** 2020
>
> [2] Cho et al. Towards understanding biased client... **AISTATS** 2022;
>
> [3] Tyurin, Richtarik. Optimal Time Complexities of Parallel Stochastic..., **NeurIPS 2023**

---

> > ### Comment · Reviewer_V3pL · 2025-11-21
> > **About the core concern.**
> >
> > Thank you for your further explanation. I really appreciate your revision. However, my original core concern remains directly unanswered. My question is actually not about asynchronous training. Instead, if some clients are offine, they are unable to receive or send anything. Your algorithm places stringent requirements on all clients (Line 20 in Algorithms 1 and 2), which are far more demanding than conventional methods. In my opinion, this is quite difficult to implement in practice, specially in a large-scale FL system even though as you said your method has stronger results. Hence, based on this key point, I currently cannot update my score. Please feel free to have further discussion with me.

---

> > > ### Author Response · Authors · 2025-12-02
> > >
> > > To address Reviewer’s comments, we conducted additional experiments.
> > > Reviewer’s main concern is the full aggregation step in line 20 of Algorithms 1 and 2.
> > > This step assumes the existence of a moment in time when all clients send the surrogate gradients
> > > accumulated during the epoch. In Algorithm 2, a client may choose not to accumulate a surrogate
> > > in certain iterations within an epoch, which is controlled by the Bernoulli hyperparameter $q_m$.
> > >
> > > In analogy to this mechanism, we introduced an additional mechanism:
> > > at the moment of full aggregation, a client may choose not to send the surrogate it accumulated during the epoch.
> > > This is modeled similarly to PPBC+, using a Bernoulli random variable with a new hyperparameter $q_e$.
> > > In other words, any client may fail to provide its surrogate during the full aggregation step.
> > > Consequently, line 20 of the Algorithm is modified to the following block:
> > >
> > > ```python
> > > generate $η_m^k$ ~ $\mathcal{B}(q_e)$
> > >
> > > if $η_m^k = 1$:
> > >
> > >      send $g_m^{k, H_k}$ to the server
> > > ```
> > >
> > >
> > > We conducted experiments for different values of both $q_m$ and $q_e$, and compared our results
> > > with standard PPBC+ (Algorithm~2) using $q_m = 0.3$, as well as with the baseline FedAvg under
> > > pathological data heterogeneity(**Figure 10** in revised version). As expected, the new algorithm performs worse than PPBC+ with full aggregation,
> > > yet it still consistently outperforms FedAvg.

---

### Official Review · Reviewer_QkH8 · 2025-10-23

**Soundness:** 2
**Presentation:** 2
**Contribution:** 2
**Rating:** 4
**Confidence:** 4

**Summary:**

This paper studies the problem of partial client participation in Federated Learning, where only a subset of clients participates in each communication round. Existing works often assume unbiased sampling or analyze specific strategies (e.g., FedAvg, FOLB, PoC). The authors propose a strategy-agnostic theoretical framework (PPBC) and an extension *PPBC+* for cases with unstable or unavailable clients. The key idea is to introduce gradient surrogates that accumulate information from inactive clients and correct bias caused by partial participation. The paper provides convergence proofs under both convex and non-convex assumptions, without relying on bounded gradient norms. Theoretical rates are claimed to improve over existing works by removing non-vanishing terms. Empirical studies on CIFAR-10 (ResNet18) and Food101 (FasterViT) show better accuracy compared to several baselines, e.g., FedAvg, SCAFFOLD, FedDyn, MOON, and F3AST.

**Strengths:**

1. The paper targets biased partial participation, which is a long-standing issue in federated optimization. This paper provides a unified theoretical analysis that relaxes some restrictive assumptions in prior works.
2. The idea of accumulating “gradient memory” from inactive clients to correct bias resembles error feedback mechanisms, but it is interestingly reinterpreted in the context of non-contractive sampling operators.
3. The derivations are detailed and mathematically consistent, with convergence rates derived for both convex and non-convex settings.

**Weaknesses:**

Despite its rigorous math and structure, the paper has serious issues regarding novelty, clarity, and experimental validation.

### 1. Theoretical novelty is overstated.

- The proposed framework (PPBC) is essentially an adaptation of error-feedback (EF) and compressed SGD ideas (see Stich & Karimireddy 2020; Richtárik et al. 2021) applied to partial participation.
- The "strategy-agnostic" claim is misleading: the algorithm still requires specifying a client sampling rule $R$ and weighting scheme $π$, hence it is not truly agnostic, but merely compatible with multiple heuristics.
- The main theoretical contributions (Theorem 3.2 and 3.4) resemble classical results from distributed optimization and show only incremental extensions to known bounds (O(1/ε²) for nonconvex, O(κ log(1/ε)) for convex).
- Algorithm 1 (PPBC) and Algorithm 2 (PPBC+) largely mimic FedAvg with gradient memory buffers and stochastic masking; there is no fundamentally new optimization idea.

### 2. Lack of meaningful baselines and comparisons.

- The paper compares PPBC mainly to generic optimizers (FedAvg, SCAFFOLD, FedDyn), not to modern partial participation frameworks such as:
  - FedNova, FedProx, FedPAQ, or adaptive sampling methods (Oort, F3AST, FedCS+).
- The fairness of the comparison is questionable: all methods use the same sampling rule $R_k = Top_C(π_k)$, which may favor the proposed algorithm by design.

### 3. Overly dense and verbose presentation.

- The paper reads more like a mathematical report than a conference paper.
  - The Introduction (Section 1) is excessively long (over 3 pages).
  - Many definitions (e.g., Assumptions 2.1–2.4) are standard and could be condensed.
- Overall readability and intuition are limited; it is difficult to see *why* the new framework helps beyond mathematical manipulation.

**Questions:**

I have no question for this paper. Some suggestions are provided for improvement.

1. Clarify what is new compared to error-feedback, EF21, and FedDyn frameworks.
2. Add communication cost and time-to-convergence metrics to demonstrate efficiency.
3. Reduce redundancy in the introduction and add intuitive explanations and diagrams.
4. Include theoretical intuition: why does bias correction help convergence in non-IID FL?

---

> ### Author Response · Authors · 2025-11-20
>
> Dear reviewer QkH8!
>
> We thank the reviewer for the time and evaluating of our work! Below we answer for the weaknesses and questions.
>
> 1. > The proposed framework (PPBC) is essentially an adaptation of...
>
> a. In your response, you mentioned the EF21 algorithm [1].
> Indeed, this is a classic work that received an oral talk at *NeurIPS'21*.
> The update steps of this method are given by:
> $$
> g_m^{k+1} = g_m^{k} + \mathcal{C}\big(\nabla f_m(x^{k}) - g_m^{k}\big),
> $$
> $$
> x^{k+1} = x^{k} - \gamma \frac{1}{M} \sum_{m=1}^{M} g_m^{k+1}.
> $$
>
> However, we would like to point out that EF21 is in some sense the simplification
> of the DIANA method [2]:
> $$
> h_m^{k+1} = h_m^{k} + \alpha \mathcal{C}\big(\nabla f_m(x^{k}) - h_m^{k}\big),
> $$
> $$
> g_m^{k} = h_m^{k} + \frac{1}{M} \sum_{m=1}^{M} \mathcal{C}\big(\nabla f_m(x^{k}) - h_m^{k}\big),
> $$
> $$
> x^{k+1} = x^{k} - \gamma \frac{1}{M} \sum_{m=1}^{M} g_m^{k}.
> $$
>
> Specifically, in addition to the surrogate $g^{k}$, DIANA exploits an extra surrogate
> $h^{k}$, which is updated recursively with a control parameter $\alpha$.
> When $\alpha = 1$, the surrogates $g^{k}$ and $h^{k}$ coincide, and DIANA reduces to EF21.
> Nevertheless, the idea behind DIANA in some sense was not fully novel at the time of
> its publication in 2019 either. It was directly inspired by the variance-reduction
> scheme SEGA [3]:
> $$
> h^{k+1} = h^{k} + e_i^{\top}\big(\nabla f(x^{k}) - h^{k}\big)e_i,
> $$
> $$
> g^{k} = h^{k} + \alpha e_i^{\top}\big(\nabla f(x^{k}) - h^{k}\big)e_i,
> $$
> $$
> x^{k+1} = x^{k} - \gamma g^{k},
> $$
> where $e_i$ is the basis vector.
> In a certain sense, DIANA generalizes the idea of a coordinate compressor
> onto a single device to arbitrary compressors on multiple devices. In turn,
> one can see that SEGA is a transfer of the SAGA technique [4]
> from finite-sum to coordinate settings. The update of SAGA is as follows:
> $$
> h_m^{k+1} = \nabla f_m(x^{k}), \qquad
> h_{i\neq m}^{k+1} = h_{i\neq m}^{k},
> $$
> $$
> g^{k} = \frac{1}{M}\sum_{i=1}^{M} h_{i}^{k} + \big(\nabla f_m(x^{k}) - h_m^{k}\big),
> $$
> $$
> x^{k+1} = x^{k} - \gamma g^{k},
> $$
> where $m$ is uniformly distributed in $\{1,\dots,M\}$.
>
> Thus, although gradient surrogates have been known in the optimization community
> for a long time, this area continues to actively evolve.
> We note that different variants of error feedback continue to appear at top-tier
> conferences [5], [6], [7], [8].
>
> b. Surrogate-based algorithms mentioned by the Reviewer are primarily focused on
> communication compression in distributed computing.
> In this context, there is a fundamental question:
> *can we treat our approach as a method with compressed communications
> and straightforwardly apply ideas from the methods discussed above?*
> The answer is **no**.
>
> The theory behind EF, EF21, and other existing error-feedback methods relies
> on the assumption that the compressor is contractive:
> $$
> \mathbb{E}\| \nabla f(x) - \mathcal{C}(\nabla f(x))\|^{2}
> \le \left(1 - \frac{1}{\beta}\right)\|\nabla f(x)\|^{2}.
> $$
>
> This condition is meaningful only for $1 \le \beta < \infty$.
> Indeed, rates of traditional methods with compression scale as
> $\mathcal{O}(\beta)$.
> However, in our setting with partial participation,
> some devices do not send their gradients at all, which corresponds to compressing
> their outputs to zero, i.e., $\mathcal{C}(x) = 0$.
> This means infinite power of compression ($\beta = \infty$),
> making compression operators no longer contractive and giving convergence
> guarantees as $\mathcal{O}(\infty)$.
> In fact, there is no compressor.
> This reasoning explains why the idea of adapting error feedback to the partial
> participation setting has not emerged in the community yet.
>
> We would also like to note that this reasoning is presented in a more concise form in the paper, in Lines 207–215.
>
> c. The argument in point b shows that the challenges we face are not merely
> technical. Indeed, it is unclear how to design an algorithm capable of handling
> non-contractive compression. Our novel insight is to interpret partial
> participation not as gradient compression but as (time-varying) sampling
> probability vector. Thus, our work does not simply extend existing error-feedback
> methods but rather introduces a new, orthogonal direction.
> In particular, we demonstrate that gradient surrogates have potential beyond the
> traditional gradient compression settings.
>
> We kindly ask the reviewer to follow our next Official Comment, in which we continue our response.

---

> ### Author Response · Authors · 2025-11-20
>
> 2. > The "strategy-agnostic" claim is misleading...
>
> We agree that the term “strategy-agnostic” may be misleading without further clarification. Our intention was not to claim that the algorithm eliminates the need to choose a sampling rule $\mathbf{R}$ or weighting scheme $\pi$. Instead, we use “strategy-agnostic’’ in the sense that the core convergence guarantees and the algorithmic mechanism do not rely on any specific choice of $\mathbf{R}$ or $\pi$.
>
> Our method is compatible with wide range of existing client-sampling heuristics, and, crucially, its convergence behavior remains stable across these choices due to surrogate-gradient bias-compensation mechanism. This property distinguishes our approach from classical partial participation methods, where performance can vary significantly depending on sampling strategy.
>
> Moreover, our Theorems 3.2, 3.4, and Corollaries D.4, D.8, E.4, E.8, are all stated and proved without assuming fixed sampling rules or weighting schemes. These results hold for any valid $\mathbf{R}$ and $\pi$, which formally supports our claim that method is strategy-flexible rather than tied to a specific heuristic.
>
> 3. > The main theoretical contributions resemble...
>
> We emphasize that our work includes detailed comparison with prior papers that attempted to provide unified analysis of partial participation (Section 1.3). We highlight that our theoretical results strictly improve upon assumptions used in concurrent works. In particular, our analysis avoids several unrealistic assumptions, e.g. uniform gradient bound [13-16]. Moreover, unlike works [14-15], our convergence guarantee does not contain any non-vanishing terms.
>
> 4. > Lack of meaningful baselines ...
>
> We respectfully disagree with the Reviewer. In all comparative experiments, we use the original strategies exactly as described in the corresponding papers (see Section 4 of the main text and Section B of the Appendix). Moreover, the majority of these baselines are built upon the FedAvg framework, while our evaluation additionally includes SCAFFOLD [9] (ICML 2020), which further strengthens and broadens the comparison.
>
> Furthermore, our experiments incorporate recent and widely used methods: FedDyn [10] and MOON [11] for PPBC; F3AST [12] for PPBC+. All corresponding results are reported in Figures 1–2, 4–8, clearly demonstrating that our comparisons are comprehensive to original methods.
>
> 5. > The Introduction is excessively...
>
> We would like to note that Section 1 includes both introduction and related work. Consequently, Section 1 contains motivation, overview of existing results and their comparison, as well as our contribution. With this in mind, we believe that the length of the first section is fully justified. To improve readability, we will separate the related work into a standalone section.
>
> 6. > Many definitions ...
>
> We note that we used the standard exposition of basic assumptions that is employed in virtually all theoretical works published at A*-level conferences [1-9].
>
> 7. > Add communication cost ...
>
> We would like to note that both the theoretical and the practical communication complexity were discussed in the Corollaries 3.3, 3.5-3.7.
>
> 8. > Include theoretical intuition: why...
>
> In our work, we present a Bayes correction technique aimed at improving the performance of weighted methods in a partial-participation setting. This correction is beneficial not so much in a specifically defined non-IID scenario, but rather for a wide range of weighting techniques designed for heterogeneous setups.
>
> We have addressed the reviewer’s concerns and are ready to continue the discussion. If there are no further questions, we kindly ask you to consider raising the evaluation score.
>
>
>
> ---
>
> **References**
>
> [1] Richtárik et al. EF21: A new, simpler, theoretically better ... **NIPS** 2021
>
> [2] Mishchenko et al. Distributed Learning with Compressed ... arXiv 2019 **(cited 272 times)**
>
> [3] Hanzely et al. SEGA: Variance reduction via gradient sketching **NIPS** 2018
>
> [4] Defazio et al. SAGA: A fast incremental gradient method ... **NIPS** 2014
>
> [5] Condat et al. EF-BV: A unified theory of error feedback ... **NIPS** 2022
>
> [6] Li et al. Analysis of error feedback in federated non-convex optimization ... **ICML** 2023
>
> [7] Fatkhullin et al. Momentum provably improves... **NIPS** 2023
>
> [8] Islamov et al. "Safe-EF: Error Feedback ... **ICML** 2025
>
> [9] Karimireddy et al. Scaffold ... **ICML** 2020
>
> [10] Bai et al. Optimization of Federated Learning...**IEEE** 2024
>
> [11] Li et al. Model-contrastive federated learning. **IEEE** 2021
>
> [12] Ribero et al.  Federated learning under intermittent client availability... **IEEE** 2022
>
> [13] Li et al. On the convergence of fedavg on non-iid data. arXiv 2019.
>
> [14] Wang and Ji. A unified analysis of federated learning ... **NeurIPS** 2022
>
> [15] Cho et al. Towards understanding biased client... **AISTATS** 2022
>
> [16] Luo et al. Tackling system and statistical heterogeneity ... **IEEE** 2022

---

### Official Review · Reviewer_v8eb · 2025-11-01

**Soundness:** 2
**Presentation:** 2
**Contribution:** 2
**Rating:** 2
**Confidence:** 3

**Summary:**

The paper addresses the fundamental challenge of partial participation (PP) in federated learning, where only a subset of clients participate in each communication round. The authors propose PPBC (Algorithm 1) and its extension (Algorithm 2), two strategy-agnostic optimization frameworks can work with arbitrary client weighting and sampling strategies. The authors provide unified convergence guarantees for both non-convex and strongly-convex objectives, with bounds that avoid previously observed non-vanishing bias terms.

**Strengths:**

It is a general framework that supports a wide range of weighting and sampling rules, biased or unbiased.

**Weaknesses:**

1. It is unclear to me why we need a general framework for partial participation. We know that client participation not only depends on server but also has close relation with system and client itself.

2. The unified theoretical analysis is claimed as one of the major contributions, but I do not see how this generate existing results. I expect a table to show all the comparisons.

3. There are so many flexible client participation works, including [1-5] and more. I am curious about the comparison between the proposed framework and analysis and these existing works.

[1] Wang, Shiqiang, and Mingyue Ji. "A unified analysis of federated learning with arbitrary client participation." Advances in neural information processing systems 35 (2022): 19124-19137.
[2] Cho, Yae Jee, Jianyu Wang, and Gauri Joshi. "Towards understanding biased client selection in federated learning." International Conference on Artificial Intelligence and Statistics. PMLR, 2022.
[3] Wang, Shiqiang, and Mingyue Ji. "A lightweight method for tackling unknown participation statistics in federated averaging." arXiv preprint arXiv:2306.03401 (2023).
[4] Li, Zhe, et al. "FAST: A Lightweight Mechanism Unleashing Arbitrary Client Participation in Federated Learning." IJCAI (2025)
[5] Yang, Haibo, et al. "Anarchic federated learning." International Conference on Machine Learning. PMLR, 2022.

**Questions:**

Please include the theoretical and experimental comparions with existing flexible client participation works in federated learning.

---

> ### Author Response · Authors · 2025-11-20
>
> Dear reviewer v8eb!
>
> We thank the reviewer for thorough analysis of our work! Below, we address concerns
>
> 1. > It is unclear to me why we need a general framework ...
>
> The federated learning community has proposed a large number of client-weighting and client-sampling strategies [1], [4], [5]. In practice, this requires evaluating multiple approaches for a given task, which significantly increases training and tuning costs. This motivates the need for methods that can generalize across sampling and weighting techniques. A unified theoretical framework is therefore **highly desirable**, as it would enable incorporating arbitrary strategies in a principled and theoretically supported manner. We briefly mentioned this motivation in Lines 119–123.
>
> 2. > The unified theoretical analysis is claimed as one...
>
> We would like to emphasize that our work includes a detailed comparison with prior papers that attempted to provide a unified analysis of partial participation (see Section 1.3). We highlight that our theoretical results strictly improve upon the assumptions used in concurrent works. In particular, our analysis avoids several unrealistic assumptions, e.g. the uniform gradient bound [2], [3], [4], [5]. Moreover, unlike works [3], [4], our convergence guarantee **does not contain** any non-vanishing terms.
>
> 3. > I expect a table to show all the comparisons.
>
> We agree that table with rates comparison can improve readability of our work. We include the table in the final version of our work.
>
> 4. > here are so many flexible client participation works...
>
> From our point of view, Weakness 3 repeats Weakness 2. Could the reviewer, please, clarify this concern?
>
> From our perspective, we believe we have addressed all of the reviewer’s questions. We would like to note that both concerns relate to the presentation of the work and were already addressed in the original manuscript. We remain open to further substantive discussion; however, if all of the reviewer’s doubts have been resolved, we respectfully request to raise our score.
>
> ---
>
> **References**
>
> [1] Wang et al. Tackling the objective ... NeurIPS 2020
>
> [2] Li et al. On the convergence of fedavg on non-iid data. arXiv preprint arXiv:1907.02189, 2019b.
>
> [3] Wang and Ji. A unified analysis of federated learning ... **NeurIPS** 2022
>
> [4] Cho et al. Towards understanding biased client selection in federated learning. **AISTATS** 2022
>
> [5]  Luo et al. Tackling system and statistical heterogeneity ... **IEEE** 2022

---

### Official Review · Reviewer_smyZ · 2025-11-01

**Soundness:** 3
**Presentation:** 2
**Contribution:** 3
**Rating:** 6
**Confidence:** 3

**Summary:**

This paper presents a new theoretical and algorithmic framework for federated learning (FL) under partial participation (PP). Unlike previous works, which rely heavily on specific strategies like FedAvg and often require restrictive assumptions (e.g., gradient boundedness or unbiased sampling), the authors propose a strategy-agnostic and theoretically sound framework that supports a wide range of biased sampling and client weighting strategies. They introduce two main algorithms: 1) PPBC: Partial Participation with Bias Correction, which maintains gradient surrogates on inactive clients and introduces a bias-corrected aggregation; and 2) PPBC+: An extension to handle unavailable clients, modeling dropout with Bernoulli trials. Theoretical results show that the framework achieves convergence in both non-convex and strongly convex settings without requiring non-vanishing error terms in theory. Experimental results on CIFAR-10 and FOOD101 show superior performance of the framework over existing baseline algorithms.

**Strengths:**

1. The paper develops tight convergence rates for both non-convex and strongly convex objectives without relying on unnatural assumptions (e.g., bounded gradients or bounded differences), thereby overcoming limitations of prior work.
2. The proposed framework decouples the optimization procedure from client selection and weighting strategies, enabling flexible, plug-and-play adaptation across a wide range of partial participation settings.
2. The algorithm PPBC+ explicitly accounts for unavailable clients, making it more practical and robust for real-world federated learning scenarios.

**Weaknesses:**

1. The experiments report results in terms of communication rounds but do not discuss communication overhead, memory consumption, or wall-clock time, all of which are critical factors in practical federated learning (FL) deployments.
2. All baselines adopt the authors’ fixed Top-C sampling strategy; however, methods like FedDyn and Moon were originally designed with their own adaptive client selection mechanisms. Enforcing a uniform sampling rule may underestimate their true performance, making the comparison potentially unfair.
3. The algorithms PPBC+ and F3AST are omitted from the CIFAR-10 experiments, while PPBC and several other baselines are omitted from the FOOD101 experiments, resulting in an incomplete experimental evaluation that limits cross-setting insights.
4. The terms "full client participation" and "partial client participation" used in Section 4 are potentially confusing. A more precise terminology would be: "partial client participation without unavailable clients/devices" and "partial client participation with unavailable clients/devices."

**Questions:**

In the proposed framework, even clients that are not selected for communication are required to compute and update their gradient surrogates, which may lead to a significant increase in computation cost. Furthermore, beyond the communication at each epoch $k$, additional communication is required at every iteration $h$ within each epoch, introducing additional communication overhead. Could the authors quantitatively analyze these extra computation and communication costs?

---

> ### Author Response · Authors · 2025-11-21
>
> Dear reviewer smyZ!
>
> Thank their for efforts and evaluation of our work! Below we address concerns.
>
> 1. > The experiments report results in terms of communication rounds but do...
>
> a. All theoretical communication costs were presented in Corollaries 3.3, 3.6, 3.8 and 3.10; D.4, D.8, E.4, and E.8 .
>
> b. For the experimental part, we used a fixed number of clients per iteration specifically, 1 client in each iteration. Thus, in our experimental setup, where we performed on average 3 iterations per epoch, the communication complexity of each epoch does not exceed $3+n$, where
> $n=10$ is the total number of clients.
> Consequently, over 100 epochs on CIFAR-10 with ResNet-18, the total communication cost corresponds to transmitting 1300 vectors, each having the dimensionality of the model. For FOOD101 with ViT over 15 epochs, 195 such vectors were transmitted.
> For the PoC[1] method, an additional transmission of losses (13 scalar values per epoch) may occur. In total, compared to standard FedAvg, this results in approximately 23% more transmitted information over the full training run.
>
> c. It is worth noting that, relative to methods that use a fixed set of clients every epoch, our approach significantly outperforms them in terms of accuracy, which compensates for the additional communication overhead that naturally arises in partial-participation settings.
>
> 2. > All baselines adopt the authors’ fixed Top-C sampling strategy...
>
> In all comparative experiments, we use the original strategies exactly as described in the corresponding papers (see Section 4 of the main text and Section B of the Appendix).
>
> 3. > The algorithms PPBC+ and F3AST are omitted from the CIFAR-10 experiments...
>
> We thank the reviewer for the thorough examination of our work. We will certainly include additional experiments to cover both datasets with both algorithms.
>
> 4.  > The terms "full client participation" and "partial client participation" used in Section 4 are potentially confusing...
>
> Thanks for the clarification. We will revise the text to ensure precise terminology.
> In Section 4.1, all clients are available, but only a subset communicates at each
> iteration to reduce communication cost. In contrast, Section 4.2 considers true partial
> participation, where some clients do not participate in training for an entire epoch.
> We will update the section to clearly distinguish these two settings.
>
> 5. > In the proposed framework, even clients that are not selected for communication are required to...
>
> Thanks to the reviewer for the very interesting question!
> In our work, we do not address the issue of local updates. While we acknowledge that this is an important component of federated learning methods, we believe that incorporating local updates into our analysis deserves a separate dedicated study.
> Regarding communication, our approach is nearly equivalent to FedAvg in terms of the number of communication operations, as demonstrated in the first part of this response. It is also important to note that, in the formulation of Algorithm 1, we focus specifically on communication complexity and do not analyze the local computational complexity on each device, assuming that every device is capable of participating in every epoch.
> If this assumption does not hold, we can instead rely on Algorithm 2, in which by setting the participation probability sufficiently low a device may choose not to participate in a given epoch and thus avoid accumulating the surrogate gradient.
>
> If there are any remaining questions regarding our work, we would be happy to continue the discussion.
>
> ---
>
> **References**
>
> [1] Cho et al. Towards understanding biased client selection in federated learning **AISTATS** 2022.

---

### Meta-Review · Area_Chair_mdLt · 2026-01-08

**Summary:**

* Algorithmic Novelty and Incremental Contribution: Multiple reviewers (QkH8, v8eb) questioned the novelty of the framework. Specifically, QkH8 argued that the Partial Participation with Bias Correction (PPBC) algorithm is essentially an adaptation of existing error-feedback (EF) and compressed SGD techniques applied to the federated learning context.

* Practicality and Communication Overhead: Reviewers (smyZ, V3pL, QkH8) raised concerns about the efficiency of the method. A major point of contention was the requirement for full client participation at certain communication points (Line 20 of the algorithms), which V3pL argued is unrealistic in large-scale, real-world federated learning systems where clients are frequently offline.


* Experimental Fairness and Completeness: Reviewers noted that the experimental evaluation was initially incomplete (omitting certain algorithms from specific datasets) and that enforcing a uniform sampling strategy across all baselines might have unfairly favored the proposed method.

**Reviewer Concerns:**

Addressed Concerns

* Communication Analysis: The authors provided a quantitative analysis showing that while their method transmits more information than FedAvg, the overhead is approximately 23%, which they argue is compensated for by higher accuracy.

* Experimental Scope: The authors committed to including missing baselines (PPBC+ and F3AST) and providing results for both CIFAR-10 and FOOD101 in the revised version.


* Theoretical Clarity: The authors clarified the distinction between full and partial participation terminology and explained that their analysis avoids unrealistic assumptions like uniform gradient bounds.


* Theoretical Justification of Epoch Length: The authors explained that the use of a Geometric distribution for epoch lengths was primarily for theoretical proof consistency to bound surrogate gradient norms.


Outstanding Concerns

* The Full Aggregation Requirement: Reviewer V3pL remained unconvinced by the authors' defense of the full aggregation step. Even after the authors introduced a mechanism to model failures during aggregation, the reviewer maintained that the requirement is "far more demanding than conventional methods" and difficult to implement in practice.


* Fundamental Novelty: While the authors argued that partial participation is a non-contractive compression setting (distinguishing it from standard error-feedback), the core criticism from Reviewer QkH8 remains a subjective but significant concern.

**Reviewer Scores:**

* smyZ: 	The authors addressed the communication overhead and terminology concerns, likely stabilizing this marginal accept.

* v8eb:	The authors provided a detailed motivation for a general framework and clarified the theoretical improvements, but the reviewer's skepticism about novelty likely prevents a high score.


* QkH8:	Mixed. The authors' technical explanation of why partial participation differs from traditional compression (non-contractive vs. contractive) can be convincing to the reviewer.


* V3pL: Despite extensive rebuttals and new experiments, V3pL explicitly stated they could not update their score due to the "stringent requirements" of the aggregation step.

---

### Decision · Program_Chairs · 2026-01-26

Reject